# Semi-Parametric Contextual Pricing with General Smoothness

**Yuxuan Han**[*]
New York University
yh6061@stern.nyu.edu

**Xiaocong Xu**[*]
University of Southern California
xuxiaoco@marshall.usc.edu

**Yuxiao Wen, Yanjun Han, Ilan Lobel, Zhengyuan Zhou**
New York University
{yw3210,yanjunhan}@nyu.edu,{ilobel,zz26}@stern.nyu.edu

## Abstract

We study the contextual pricing problem, where in each round a seller observes a context, sets a price, and receives a binary purchase signal. We adopt a semi-parametric model in which the demand follows a linear parametric form composed with an unknown link function from a $\beta$-Hölder class. Prior work established regret rates of $\tilde{\mathcal{O}}(T^{2/3})$ for $\beta = 1$ and $\tilde{\mathcal{O}}(T^{3/5})$ for $\beta = 2$. Under a uni-modality condition, we propose a unified algorithm that combines the stationary subroutine of Wang & Chen (2025) with local polynomial regression, achieving the general rate $\tilde{\mathcal{O}}(T^{\frac{\beta+1}{2\beta+1}})$ for all $\beta \geq 1$. This recovers and strengthens existing results, while also addressing a gap in the prior analysis for $\beta = 2$. Our analysis develops tighter semi-parametric confidence regions, removes derivative lower bound assumptions from earlier work, and offers a sharper exploration–exploitation trade-off. These insights not only extend theoretical guarantees to general $\beta$ but also improve practical performance by reducing the need for long forced-exploration phases.

## 1 Introduction

Dynamic pricing addresses a central problem in revenue management, where a seller repeatedly interacts with users by offering personalized prices for the same product and collecting revenue from the resulting sales (Cournot, 1927; Den Boer, 2015). Across these interactions, users exhibit heterogeneous demand or private valuations, and the seller faces uncertainty in how demand responds to the offered prices. This demand function effectively captures the market's valuation of the product. Consequently, the seller must learn the demand in real time while simultaneously aiming to maximize revenue, which gives rise to the fundamental exploration–exploitation tradeoff in dynamic pricing (Kleinberg & Leighton, 2003).

Recently, contextual dynamic pricing has gained significant traction in online retail, driven by the widespread availability of user-specific and contextual information (Cohen et al., 2020; Wang et al., 2021; Chen & Gallego, 2021; Luo et al., 2024; Wang & Chen, 2025). Modern platforms can conveniently access rich side information—such as a user's account profile, browsing and purchase history, or relevant environmental factors—before deciding on a price. Incorporating such contextual signals enables sellers to move beyond static or aggregate demand models and tailor prices to individual users or market segments. To achieve this, firms need to find algorithms that learn how demand depends jointly on both price and context and determine an optimal personalized price when a context is revealed.

Among various formulations for capturing the contextual dependence, the formulation with *linear utility model* and *non-parametric noise* is receiving increasing attention due to its flexibility compared with fully parametric model (Javanmard & Nazerzadeh, 2019) and simplicity between fully non-parametric models (Chen & Gallego, 2021). In this formulation, each user is associated with a

---

[*]Equal contribution, alphabetical order.

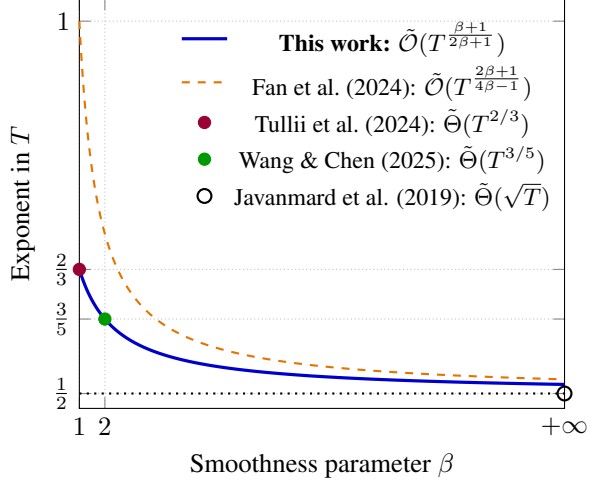

Figure 1: Exponent in $T$ vs. smoothness $\beta$.

| Assumptions | |
| --- | --- |
| (A) Strong uni-modality | (B) Context density LB |
| (C) $\Sigma \succ 0$ | (D) $g'(\cdot) < 0$ |
| **Result** | **Condition** |
| Tullii et al. (2024)($\beta = 1$) | None |
| Fan et al. (2024)*($\beta \geq 1$) | (B)(C)(D) |
| Wang & Chen (2025)($\beta = 2$) | (A)(C)†(D) |
| **Our work**($\beta \geq 1$) | (A)(C)† |

Table 1: Comparison of assumptions in semi- and non-parametric demand models. The strong uni-modality condition is given in Assumption 3, and $\Sigma = \mathbb{E}[c_t c_t^\top]$.
* Fan et al. (2024) does not require (A), but instead assumes another shape condition on $g(\cdot)$.
† In both Wang & Chen (2025) and our work, condition (C) is imposed only during the initial exploration period of length $\tilde{\mathcal{O}}(T^{\frac{\beta+1}{2\beta+1}})$.

context vector $c_t \in \mathbb{R}^d$ upon arrival and derives a private utility $u_t = c_t^\top \theta_* + \xi_t$, with $\theta_* \in \mathbb{R}^d$ being an unknown parameter and $\xi_t$ random noise. After offering a price $p_t$, the seller receives revenue feedback $p_t \mathbf{1}\{p_t \leq u_t\}$ from the user, which indicates whether a purchase is made (i.e., revenue is generated) or not. If denoting the tail distribution function $\mathbb{P}(\xi_t \geq z)$ by $g(z)$, the expected demand then reduces to $D(p) = g(c_t^\top \theta_* - p)$, this corresponds to the *semi-parametric* formulation (Ichimura, 1993; Hristache et al., 2001; Dalalyan et al., 2008).

As in the statistical estimation literature, the regularity of $g(\cdot)$ affects the difficulty of demand identification and thus decision making. Previous works have extensively studied the semi-parametric pricing problem under different levels of regularity of $g$. For instance, Tullii et al. (2024) and Wang & Chen (2025) establish regret bound $\tilde{\mathcal{O}}(T^{\frac{2}{3}})$ and $\tilde{\mathcal{O}}(T^{\frac{3}{5}})$ respectively under first and second-order smoothness assumptions. In contrast, characterizing the general $\beta$-smooth regime for $\beta \in [1, +\infty)$ plays an important role in understanding how regularity influences demand estimation and how the non-parametric regime interpolates to the parametric rates (Hu et al., 2020). To the best of our knowledge, the only prior work that attempts to provide such a unified treatment is Fan et al. (2024), which establishes an $\tilde{\mathcal{O}}(T^{\frac{2\beta+1}{4\beta-1}})$ regret bound. However, this result does not recover the $\tilde{\mathcal{O}}(T^{\frac{3}{5}})$ rate under the conditions of Wang & Chen (2025) and even degenerates to linear regret when $\beta = 1$, leaving room for further improvement.

## OUR CONTRIBUTIONS

Motivated by this gap, we explore the semi-parametric pricing setting in this work and provide the improved regret bounds, we summarize our contributions as the following:

**Improved Regret Bound for $\beta \geq 1$ Regime.** Under *strong uni-modality* (Assumption 3) as in Wang & Chen (2025); Chen & Gallego (2021), we establish a regret upper bound of $\tilde{\mathcal{O}}(T^{\frac{\beta+1}{2\beta+1}})$ for all $\beta \geq 1$. For comparison, under uni-modality together with additional regularity, Fan et al. (2024) obtain $\tilde{\mathcal{O}}(T^{\frac{2\beta+1}{4\beta-1}})$; under these distinct assumptions, our bound achieves a smaller exponent in $T$. Our result matches the optimal contextual guarantees for $\beta = 1$ Luo et al. (2024); Tullii et al. (2024) and $\beta = 2$ Wang & Chen (2025), and it interpolates to the parametric rate $\tilde{\mathcal{O}}(\sqrt{T})$ as $\beta \to \infty$. Moreover, it coincides with the tight non-contextual bound $\tilde{\Theta}(T^{\frac{\beta+1}{2\beta+1}})$ for general $\beta$ established in Wang et al. (2021). Hence, under strong uni-modality, semi-parametric contextual pricing is provably no harder than its non-contextual counterpart.

**Unified Algorithm and Analysis.** The proposed regret bound is achieved by developing a unified joint estimation procedure and confidence bound analysis for the parametric and non-parametric parts via local polynomial regression for $\beta \geq 1$. In particular, when $\beta = 1$, the resulting confidence bound applies directly to yield $\tilde{\mathcal{O}}(T^{2/3})$ regret via the optimistic principle without strong uni-modality,

matching Tullii et al. (2024). When $\beta \geq 2$, a finer control of the parametric estimation error is required to exploit higher-order smoothness. We combine our procedure with elements of Wang & Chen (2025) to obtain the general bound, extending their algorithmic design from the case $\beta = 2$.

**Improved Confidence Bound Analysis.** Our confidence bound analysis generalizes Tullii et al. (2024); Wang & Chen (2025). In particular, when $\beta \geq 2$, we encounter the same challenge of leveraging higher-order smoothness as in Wang & Chen (2025). While we adopt the idea of constrained least squares from their work, extending it to general $\beta$ requires substantially more than a straightforward calculation. First, the analysis in Wang & Chen (2025) heavily relies on a linear-time *local exploration* schedule. As we detail in Section 6, this works under uni-modality when $\beta = 2$, but leads to a degenerate regret rate as $\beta$ increases. Second, although Wang & Chen (2025) pioneered the finite-sample analysis of constrained least squares, the complex dependency beyond martingale structure in such a joint estimation procedure prevents the direct application of standard concentration inequalities such as Azuma–Hoeffding. In their proof, this dependence is overlooked and therefore cannot yield the claimed result[1]. These challenges motivate our improved analysis, which bypasses the dependence issue and significantly shortens the exploration period. As an additional contribution, our analysis also removes the *strictly increasing CDF* condition listed in Table 1, which has been assumed in prior smooth semi-parametric settings Fan et al. (2024); Wang & Chen (2025), thereby broadening the applicability of the theory.

**Notation.** For $n \in \mathbb{Z}_+$, we let $[n] := \{1, \ldots, n\}$ and denote by $\|\cdot\|$ the $\ell_2$ norm. For a positive definite matrix $A \in \mathbb{R}^{n \times n}$ and $u \in \mathbb{R}^n$, write $\|u\|_A := \sqrt{u^\top A u}$. For a matrix $A$, $\|A\|_F$ is the Frobenius norm; $\lambda_{\min}(A)$ and $\lambda_{\max}(A)$ are its extremal eigenvalues; and $A \succeq B$ means $A - B$ is positive semidefinite. For $a \in \mathbb{R}$, $\lfloor a \rfloor$ is the greatest integer $< a$. We use $a \lesssim b$ or $a = \mathcal{O}(b)$ to mean there exists $C > 0$ such that $|a| \leq C|b|$; $a \gtrsim b$ or $a = \Omega(b)$ to mean there exists $c > 0$ such that $|a| \geq c|b|$; and $a \asymp b$ or $a = \Theta(b)$ to mean there exist $c_1, C_1 > 0$ such that $c_1|b| \leq |a| \leq C_1|b|$, all those constants may only depend on $\beta$. We also use $\tilde{\mathcal{O}}, \tilde{\Omega}, \tilde{\Theta}$ to hide polylogarithmic factors.

## 2 PROBLEM FORMULATION

**Dynamic Pricing with Linear Valuations.** At each period $t \in [T]$, a new customer arrives with observable feature vector $\mathsf{c}_t \in \mathbb{R}^d$ drawn i.i.d. from an unknown distribution $P_{\mathcal{C}}$ and generates underlying valuation as $u_t = \mathsf{c}_t^\top \theta_* + \xi_t$ for some $\theta_* \in \mathbb{R}^d$ and $\xi_t$ i.i.d. drawn from a distribution $P_\Xi$ with CDF $F_\Xi$. After observing $\mathsf{c}_t$, the seller posts a price $p_t \in [0, p_{\max}]$ and observes the binary purchase feedback $y_t = \mathbf{1}\{u_t \geq p_t\}$ and the corresponding revenue $p_t y_t$. If we denote the tail function $g(r) := 1 - F_\Xi(r)$, the conditional revenue is then given by

$$R(\mathsf{c}_t, p_t) := p_t \mathbb{E}[y_t \mid \mathsf{c}_t, p_t] = p_t \mathbb{P}(\xi_t \geq p_t - \mathsf{c}_t^\top \theta_* \mid \mathsf{c}_t, p_t) = p_t g(p_t - \mathsf{c}_t^\top \theta_*).$$

The goal of the seller, without knowing $\theta_*$ and $P_\Xi$, is to determine an adaptive policy $\pi$ for posting prices $p_t$ to maximize the cumulative revenue $\mathbb{E}[\sum_{t=1}^T R(\mathsf{c}_t, p_t)]$. The performance of the policy is evaluated by the cumulative revenue gap relative to the optimal policy:

$$\text{Regret}(T) := \mathbb{E}\Big[ \sum_{t=1}^T \max_p R(\mathsf{c}_t, p) - R(\mathsf{c}_t, p_t) \Big]. \tag{1}$$

For compactness we will use an augmented vector form: define $x_t := (\mathsf{c}_t^\top, p_t)^\top \in \mathbb{R}^{d+1}$ and $\theta_0 := (-\theta_*^\top, 1)^\top \in \mathbb{R}^{d+1}$, so that $x_t^\top \theta_0 = p_t - \mathsf{c}_t^\top \theta_*$ and $\mathbb{E}[y_t \mid \mathsf{c}_t, p_t] = g(x_t^\top \theta_0)$.

**Smoothness Condition and Assumptions.** For the parametric part of the model, we make the following boundedness assumption:

**Assumption 1.** *There exist constants $C_\mathsf{c}, C_\theta > 0$ such that $\|\mathsf{c}_t\| \leq C_\mathsf{c}$ almost surely and $\|\theta_*\| \leq C_\theta$. Let $V := C_\mathsf{c} C_\theta + p_{\max}$. Furthermore, the noise distribution $P_\Xi$ is supported on $[-V, V]$.*

Under Assumption 1, the value-price gap $x_t^\top \theta_0 = \mathsf{c}_t^\top \theta_* - p_t$ lies in $[-V, V]$ for all $t$, so $g$ is only evaluated on a compact interval.

---

[1] **We note that, in the revised version of Wang & Chen (2025) after our submission, they independently addressed this issue using an argument different from ours.**

For the non-parametric part, we make the following assumption on $g(\cdot)$, which is equivalent to making an assumption on $F_\Xi$ due to the relation $g(\cdot) = 1 - F_\Xi(\cdot)$.

**Assumption 2** ($\beta$-Hölder smoothness of $g$). *There exist constants $L_g > 0$ and $\beta > 0$ such that $g : [-V, V] \mapsto [0, 1]$ is $\lfloor \beta \rfloor$ times continuously differentiable and, for all $u, u' \in [-V, V]$,*

$$\left| g(u') - \sum_{k=0}^{\lfloor \beta \rfloor} \frac{(u' - u)^k}{k!} g^{(k)}(u) \right| \leq L_g |u' - u|^\beta.$$

This is a standard notation for describing smoothness in nonparametric estimation (Györfi et al., 2002; Tsybakov, 2008). It unifies previous assumptions in the sense that $\beta = 1$ corresponds to the Lipschitz setting studied in Tullii et al. (2024); Luo et al. (2024), and $\beta = 2$ corresponds to the "2nd-order smooth" setting studied in Wang & Chen (2025); Luo et al. (2022). In this work, we provide a unified treatment for all $\beta \geq 1$.

For the revenue function, we make the following assumption.

**Assumption 3** (Strong uni-modality). *For any $|u| \leq C_c C_\theta$, under the shorthand notation $r(u, p) := pg(p - u)$, the maximizer $p^\star(u) := argmax_p r(u, p)$ is unique and lies in the strict interior $(0, p_{\max})$. Moreover, there exist constants $0 < \sigma_r \leq L_r < \infty$ such that for all $|u| \leq C_c C_\theta$ and all $p \in [0, p_{\max}]$,*

$$\frac{\sigma_r}{2} |p - p^\star(u)|^2 \leq r(u, p^\star(u)) - r(u, p) \leq \frac{L_r}{2} |p - p^\star(u)|^2.$$

Assumption 3 says that, given any valuation $u$ fixed, the revenue function $r(u, \cdot)$ is locally strongly convex around its maximizer. Such conditions has appeared in various pricing models (Broder & Rusmevichientong, 2012; Wang et al., 2014; Chen & Gallego, 2021; Wang & Chen, 2025).

It is worth to note that while broadly-accepted, the strong uni-modality is tend to be believed as a relative strong assumption in non-contextual pricing setting, in sense that a $\tilde{\mathcal{O}}(\sqrt{T})$ regret can be achieved even under the Lipschitz condition ($\beta = 1$) of $F_\Xi$ (Kleinberg & Leighton, 2003; Wang et al., 2021). In sharp contrast, for the contextual setting, the $\Omega(T^{3/5})$ regret lower bound under the $\beta = 2$ and Assumption 3 developed in Wang & Chen (2025) shows a clear separation between the contextual and non-contextual cases, indicating that even under Assumption 3, the pricing problem is not overly simplified in our setting.

**Organization.** In the remaining contexts, we divide our presentation into four parts. In Section 3, we recall an initialization guarantee and discuss the potential trade-off incurred for general $\beta$. In Section 4, we describe a joint estimation procedure for semi-parametric estimation and its statistical guarantee. Finally, we combine these two procedures with a policy improvement oracle introduced in Section 5 to present the complete algorithm and its regret guarantee in Section 6.

## 3 INITIAL EXPLORATION PHASE

In the initial exploration, phase, our goal is to obtain a *pilot estimator* $\bar{\theta} \in \mathbb{R}^d$ such that $\|\bar{\theta} - \theta_*\|_2 \leq \eta$ for certain error level $\eta$.

As discussed in Wang & Chen (2025), such a pilot estimator may be obtained through initial access to offline data. However, suitable offline data is not always available. In this section, we recall an initial exploration guarantee under the diverse context distribution assumption from Fan et al. (2024):

**Assumption 4.** *There exists some positive constant $c_{\min}$ so that $\mathbb{E}[(c_t^\top, 1)^\top (c_t^\top, 1)] \succeq \frac{c_{\min}}{d} I$.*

Under Assumption 4, Fan et al. (2024) showed that by posting uniform exploration prices $p_t \sim \text{Unif}[0, V]$ for $\tilde{\mathcal{O}}(\eta^{-2})$ rounds, one can obtain the desired estimator through a suitable parametric estimation procedure. The procedure is summarized in Algorithm 1, and we present the main result below; its proof is provided in Appendix B for completeness.

**Lemma 5.** *Suppose Assumption 1 and 4 hold. Fix $\delta \in (0, 1)$. Algorithm 1 with $\mathcal{O}\big(\eta^{-2} d^3 \log(1/\delta)\big)$ running time can output a parametric estimator $\bar{\theta}$ with $\|\bar{\theta} - \theta_*\| \leq \eta$ with probability at least $1 - \delta$.*

In particular, every price posted in the exploration phase incurs a constant sub-optimality gap, so the total regret from exploration scales as $\tilde{\mathcal{O}}(\eta^{-2})$. To match the desired $\tilde{\mathcal{O}}(T^{\frac{\beta+1}{2\beta+1}})$ regret bound under

---

**Algorithm 1** Pilot estimation (adapted from Fan et al. (2024))

---

1: **Input:** Running time t;
2: **for** the next t time periods **do**
3:      Offer price $p_t \sim \text{Unif}[0, p_{\max}]$ and let $\{\mathcal{D}(t) \equiv \{c_t, p_t, y_t\}_{t \in [t]}\}$ be the collected data;
4: Compute $\bar{\theta} \leftarrow \arg\min_{\theta \in \mathbb{R}^d} \; t^{-1} \sum_{t \in [t]} (p_{\max} y_t - c_t^\top \theta)^2$;
5: **return** $\bar{\theta}$

---

the $\beta$-Hölder condition, we require $\eta^{-2} \asymp T^{\frac{\beta+1}{2\beta+1}}$, thus the pilot estimator accuracy must be restricted to $\mathcal{O}(T^{-\frac{\beta+1}{4\beta+2}})$. When $\beta = 1$, as discussed later in Section 4.1, with $\eta \asymp T^{-1/3}$, even a linear level perturbation bound $\mathcal{O}(T\eta)$ is sufficient to achieve the optimal $\tilde{\mathcal{O}}(T^{2/3})$ regret. As $\beta$ increases, the required regret rate decreases while the pilot error increases, calling for a sharper approach to exploiting smoothness information—this is precisely the challenge we attempt to address in this work. **For convenience, we fix the level of $\eta$ for a given $\beta$ throughout the remaining discussion:** When discussing the setting with a specific $\beta \geq 1$, we set the corresponding pilot estimation error to $\eta = T^{-\frac{\beta+1}{4\beta+2}}$. As noted in the earlier discussion, this choice is the sharpest possible rate under Lemma 5 without violating the desired $\mathcal{O}(T^{\frac{\beta+1}{2\beta+1}})$ regret.

Finally, we note that Assumption 4 is used only to obtain the pilot estimator and is therefore needed only during the initial exploration period of length $\tilde{\mathcal{O}}(T^{\frac{\beta+1}{2\beta+1}})$. While careful readers may find the phrase "needed only" unusual since our setting assumes i.i.d. contexts, here we allow the context distribution to be two-phased: after the initial exploration period, the contexts may follow any distribution that satisfies Assumption 1, not necessarily Assumption 4.

## 4    SEMI-PARAMETRIC ESTIMATION SUB-ROUTINE

### 4.1    PILOTED LOCAL POLYNOMIAL REGRESSION

Given a pilot estimate $\bar{\theta}_0 := (-\bar{\theta}^\top, 1)^\top$ of $\theta_0$, we introduce a piloted local polynomial regression subroutine, with an input dataset $\mathcal{D}$ collected through a sub-routine conducted during a sub-interval of the total horizon $[T]$, as the following.

In Algorithm 2. The input dataset $\mathcal{D}$ is first binned into different intervals $I_j, j \in [M]$ based on the pilot estimator $\bar{\theta}_0$, then a local non-parametric estimation is performed for every candidate parameter $\theta$ over interval $I_j$ to obtain $\hat{g}_j(\cdot \mid \theta)$. Let $n_j := |\mathcal{T}_j|$. Now we present the following *deterministic* estimation error guarantee of $\hat{g}_j(\cdot \mid \theta)$ without any requirement on the input dataset $\mathcal{D}$ :

**Proposition 6.** *Fix* $j \in [M]$ *and* $h \geq T^{-\frac{1}{2\beta+1}}$. *Under Assumptions 1 and 2, for any $\theta$ with* $\|\theta - \bar{\theta}_0\| \leq \eta$ *and any $x$ with* $\|x\| \leq C_c + p_{\max}$ *and* $x^\top \bar{\theta}_0 \in I_j$, *if* $\Lambda_j(\theta)$ *invertible then*

$$\hat{g}_j(x \mid \theta) - g(x^\top \theta_0) = v_j(x, \theta)^\top \delta_j(\theta) + U_j(x, \theta)^\top \sum_{t \in \mathcal{T}_j} \big( \underbrace{y_t - g(x_t^\top \theta_0)}_{:=\varepsilon_t} \big) \Lambda_j^{-1}(\theta) U_j(x_t, \theta)$$

$$+ \mathcal{O}\big( h^\beta (1 + \sqrt{n_j} \|U_j(x, \theta)\|_{\Lambda_j^{-1}(\theta)}) \big),$$

*where, with* $X_j(x, \theta) := \big( (x - \bar{x}_j)^\top, \ldots, \ell((x - \bar{x}_j)^\top \theta)^{\ell-1} \cdot (x - \bar{x}_j)^\top \big)^\top \in \mathbb{R}^\ell$ , *we define*

$$v_j(x, \theta) := X_j(x, \theta) - U_j(x, \theta)^\top \Lambda_j^{-1}(\theta) \sum_{i \in \mathcal{T}_j} U_j(x_i, \theta) X_j(x_i, \theta) \in \mathbb{R}^\ell,$$

$$\delta_j(\theta) := \big( g'(\bar{x}_j^\top \theta_0)(\theta - \theta_0)^\top, \ldots, \frac{g^{(\ell)}(\bar{x}_j^\top \theta_0)}{\ell!}(\theta - \theta_0)^\top \big)^\top \in \mathbb{R}^\ell.$$

In Proposition 6, the estimation error is decomposed into three terms. The first term, which arises from the mismatch between the pilot estimator $\bar{\theta}_0$ and the underlying truth $\theta_0$, creates a central difficulty in the analysis for general[3] $\beta > 1$, as discussed in Remark 7. Due to the $\theta - \theta_0$ term

---

[2]the existence of such $\bar{x}_j$ is straightforwardly ensured by $\mathcal{T}_j \neq \emptyset$.
[3]Note that when $\beta = 1$, the $v_j(x, \theta)^\top \delta_j(\theta)$ term does not appear since $\ell = 0$.

---

**Algorithm 2** Piloted Local Polynomial Regression

---

1: **Inputs:** pilot estimator $\bar{\theta}_0$ with $\|\bar{\theta}_0 - \theta_0\| \leq \eta$; smoothness $\beta \geq 1$, dataset $\mathcal{D} := \{\mathsf{c}_i, p_i, y_i\}_{i=1}^n$, precision of partition $h$.
2: **Initialization:** partition $[-V, V]$ into $M = \lceil 1/h \rceil$ equal bins $I_1, \ldots, I_M$. For $j \in [M]$, set

$$\mathcal{T}_j := \left\{ i \in [n] : x_i = (\mathsf{c}_i^\top, p_i)^\top, \ x_i^\top \bar{\theta}_0 \in I_j \right\}$$

and the polynomial degree $\ell = \lfloor \beta \rfloor$.
3: **for** $j = 1, \ldots, M$ **do**
4:      **if** $\mathcal{T}_j \neq \emptyset$ **then**
5:          Pick arbitrary $^2 \bar{x}_j \in I_j$ with $\|\bar{x}_j\| \leq C_{\mathsf{c}} + p_{\max}$.
6:          For any $\theta$ with $\|\theta - \bar{\theta}_0\| \leq \eta$ and any $x$ with $x^\top \bar{\theta}_0 \in I_j$, define

$$\Delta_j(x, \theta) := (x - \bar{x}_j)^\top \theta, \qquad U_j(x, \theta) := \left(1, \Delta_j(x, \theta), \ldots, \Delta_j(x, \theta)^\ell\right)^\top,$$

     and the Gram matrix $\Lambda_j(\theta) := \sum_{i \in \mathcal{T}_j} U_j(x_i, \theta) U_j(x_i, \theta)^\top$. Set the local estimator

$$\widehat{g}_j(x \mid \theta) := \begin{cases} U_j(x, \theta)^\top \Lambda_j(\theta)^{-1} \sum_{i \in \mathcal{T}_j} y_i U_j(x_i, \theta), & \text{if } \Lambda_j(\theta) \text{ is invertible,} \\ 0, & \text{else.} \end{cases}$$

7:      **else**
8:          Set $\widehat{g}_j(\cdot \mid \theta) \equiv 0$ for all $\theta$.
9: Output $\{\widehat{g}_j(\cdot \mid \theta)\}_{j \in [M], \|\theta - \bar{\theta}_0\| \leq \eta}$.

---

appearing in $\boldsymbol{\delta}_j$, we can only obtain an $\mathcal{O}(\eta)$ upper bound on $|\boldsymbol{v}_j(x, \theta)^\top \boldsymbol{\delta}_j(\theta)|$ in general. On the other hand, carrying such an $\mathcal{O}(\eta)$ bound yields an overall rate of $\mathcal{O}(T\eta) = \mathcal{O}(T^{\frac{3\beta+1}{4\beta+2}})$—far above the desired $\mathcal{O}(T^{\frac{\beta+1}{2\beta+1}})$ result.

This suboptimal $\mathcal{O}(\eta)$-order term in Proposition 6 is the main motivation for using a refined estimator of $\theta_0$ beyond the initial pilot estimation, leading to the *constrained least-squares estimator* for refining the parametric estimates in Section 4.2.

**Remark 7** (An $\mathcal{O}(T^{\frac{\beta+1}{2\beta+1}})$ regret via Proposition 6 without the first term)**.** *Since Proposition 6 is quite general and requires no assumption on how $\mathcal{D}$ is collected, in Appendix J we show that, when combining Algorithm 2 with an Upper-Confidence-Bound–based algorithm design, an $\tilde{\mathcal{O}}(T^{\frac{\beta+1}{2\beta+1}})$ regret can be achieved if the right-hand side of Proposition 6 does not contain the $\boldsymbol{v}_j(x, \theta)^\top \boldsymbol{\delta}_j(\theta)$ term. While the omission of this first term is in general impossible, this discussion mainly illustrates how the problem can be simplified without it.*

*We also note that there are two special cases where such an omission can rigorously hold. First, when $\beta = 1$, we have $\ell = 0$, and this analysis recovers the $\tilde{\mathcal{O}}(T^{2/3})$ rate in Tullii et al. (2024)[4], which is minimax optimal. Second, in the non-contextual setting studied in Wang et al. (2021), where $\mathsf{c} \equiv \mathbf{0}$ and $x = (\mathbf{0}, p)$ only depends on price, one can show that $\boldsymbol{v}_j((\mathbf{0}, p), \theta)^\top \boldsymbol{\delta}_j(\theta) \equiv 0$.*

*In this case, our discussion recovers the general $\mathcal{O}(T^{\frac{\beta+1}{2\beta+1}})$ regret in Wang et al. (2021), which is also minimax-optimal. We also note that throughout this discussion we do not need the strong uni-modality condition in Assumption 3, and in the second setting we do not need the diversity condition in Assumption 4 for exploration. This matches the minimal assumptions used in prior work.*

**Notation for Convenience:** While Algorithm 2 is described with flexible precision $h \geq T^{-\frac{1}{2\beta+1}}$ for generality, throughout the main text we by default set $h = n^{-\frac{1}{2\beta+1}}$ when inputted $|\mathcal{D}| = n$.

---

[4]It is worth noting that the algorithm in Tullii et al. (2024) can work even in an adversarial context setting with adaptive initial exploration, as we discussed in Appendix J.

## 4.2 Constrained Least Squared Refinement

Using the local fits $\widehat{g}_j(\cdot \mid \theta)$ from Algorithm 2, we refine the parametric estimate via a constrained least–squares (LSE) subroutine, a standard device in semi-parametric estimation, cf. Härdle et al. (1993); Wang & Chen (2025). For each $j \in [M]$ (with $\mathcal{T}_j$ defined in Section 4.1), define

$$\widehat{\theta}_j \in \underset{\|\theta - \bar{\theta}_0\| \leq \eta}{\arg\min} \sum_{i \in \mathcal{T}_j} (y_i - \widehat{g}_j(x_i \mid \theta))^2. \tag{2}$$

We have the following statistical guarantee for such constrained LSE under additional *conditional independence* assumption on $\mathcal{D}$:

**Proposition 8.** *Fix any $\delta \in (0,1)$. Suppose Assumptions 1,2 hold and $\{y_i\}_{i=1}^n$ are mutually independent conditioned on $\{x_i\}_{i=1}^n$. Then under the condition that $\Lambda_j(\theta)$ is invertible and $\zeta \asymp n^{\frac{\beta+1}{2\beta+1}} = \Omega(d^7 \log^{7/2}(1/\delta)\sqrt{n})$, we have with probability at least $1 - \mathcal{O}(n\delta)$, uniformly for all $x$ with $x^\top \bar{\theta}_0 \in I_j$ and $j \in [M]$, the solution $\widehat{\theta}_j$ to (2) satisfies*

$$\left|\widehat{g}_j(x \mid \widehat{\theta}_j) - g(x^\top \bar{\theta}_0)\right| \lesssim \mathrm{Err}_j(x) + n^{-\frac{\beta}{2\beta+1}}, \quad \forall j \in [M] \text{ and } \forall x \text{ such that } x^\top \bar{\theta}_0 \in I_j. \tag{3}$$

*Where $\mathrm{Err}_j(x) := \left(\sqrt{d \log(1/\delta)} + \sqrt{n_j} n^{-\frac{\beta}{2\beta+1}}\right) \cdot \left(\|\boldsymbol{v}_j(x, \widehat{\theta}_j)\|_{(\Sigma_j(\widehat{\theta}_j) + \zeta I)^{-1}} + \|U_j(x, \widehat{\theta}_j)\|_{\Lambda_j^{-1}(\widehat{\theta}_j)}\right)$ and $\Sigma_j(\theta) := \sum_{i \in \mathcal{T}_j} \boldsymbol{v}_j(x_i, \theta) \boldsymbol{v}_j(x_i, \theta)^\top$.*

Proposition 8 describes an error bound on the glued estimator

$$\widehat{g}(x) := \sum_{j \in [M]} \mathbf{1}\{x^\top \bar{\theta}_0 \in I_j\} \, \widehat{g}_j(x \mid \widehat{\theta}_j),$$

which relies on a characterization for the parametric minimizer (2). A key difficulty is the dependence: during the analysis of the constrained least squared estimator, the all samples in $\mathcal{T}_j$ are used compute $\widehat{\theta}_j$ and $\widehat{g}_j(\cdot \mid \cdot)$, this together with the non-linearity introduced in the squared loss, creating a complicated dependency structure. With the unified local polynomial approach, a key observation in our analysis is that such complicated joint-lease squared form can be reduced to the concentration analysis of a quadratic form involving observation noises, which then can be tackled via the standard Hanson-Wright inequality.

To see why Proposition 8 refines Proposition 6 and yields improved regret, we argue in aggregate rather than pointwise. The right-hand side of Proposition 8 has a *self-normalized* vector form, which implies the following bound under suitable distributional assumptions on $x$ and $\mathcal{D}$:

**Theorem 9.** *Fix $j \in [M]$. Assume in additional to Proposition 8 that $\mathcal{T}_j$ allows a disjoint decomposition $\mathcal{T}_j = \mathcal{T}_j^{\mathrm{ra}} \cup \mathcal{T}_j^{\mathrm{ro}}$ with:*
*i) Samples from $\mathcal{T}_j^{\mathrm{ra}}$ are i.i.d. from a stationary distribution $Q_j$ and $|\mathcal{T}_j^{\mathrm{ra}}| \geq \lceil n_j/2 \rceil$,*
*ii) For any $\theta$ with $\|\theta - \bar{\theta}_0\| \leq \eta$, $\lambda_{\min}(H\Lambda_j^{\mathrm{ro}}(\theta)H) \gtrsim \sqrt{n_j}$ with $H = \mathrm{diag}(1, M^{-1}, \ldots, M^{-\ell}) \in \mathbb{R}^{(\ell+1) \times (\ell+1)}$ and $\Lambda_j^{\mathrm{ro}}(\theta) := \sum_{t \in \mathcal{T}_j^{\mathrm{ro}}} U_j(x_t, \theta) U_j(x_t, \theta)^\top$.*
*Then it holds with probability at least $1 - \delta$ that*

$$\mathbb{E}_{x \sim Q_j}[\mathrm{Err}_j(x)] \lesssim d^4 \log^2(1/\delta) \left(n_j^{-\frac{1}{2}} + n^{-\frac{\beta}{2\beta+1}}\right). \tag{4}$$

Theorem 9 states that under the distribution collecting (a subset of) $\mathcal{D}$, one attains a parametric rate in $n_j$ and a $\beta$-dependent non-parametric rate in $n$ that matches the usual minimax-optimal rate under $\beta$-Hölder smoothness (Tsybakov, 2008). In particular, when $n$ is linear in $T$, which is the scenario in our subsequent regret analysis, the second term of (4) scales as $\eta^{\frac{2\beta}{\beta+1}}$, improving the linear dependency $\eta$ rate as we discussed in Section 4.1.

**Remark 10.** *In Theorem 9, we assume an $\Omega(\sqrt{n_j})$ eigenvalue lower bound condition on normalized version of $\Lambda_j(\theta)$, mainly to carry out the perturbation analysis involving the inverse of the empirical matrix. By contrast, Lemma EC.11 of Wang & Chen (2025) states a similar result for $\beta = 2$, but under the stronger condition $\Lambda_j(\theta) \succeq n_j I$. As we note in the subsequent Remark 12, relaxing the eigenvalue condition is key to extending the analysis to $\beta \geq 2$ while maintaining the $T^{\frac{\beta+1}{2\beta+1}}$ regret.*

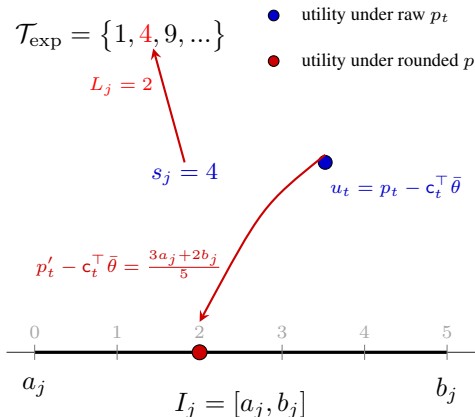

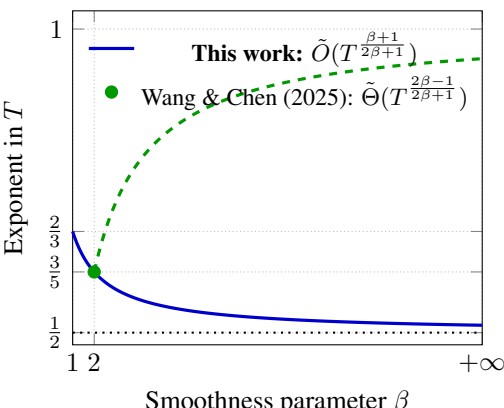

Figure 2: **Forced exploration via price rounding.** At the $s_j$-th time that a raw price $p_t$ determined utility is piloted to interval $I_j$. If $s_j$ is the $L_j$-th element in $\mathcal{T}_{\exp}$, the raw price will be rounded so that the piloted utility lies at the $(L_j \bmod \lfloor\beta\rfloor)$-th equi-partition points of $I_j$. This figure illustrates the case $s_j = 4, \beta = 6$, which corresponds to $L_j = 2$-nd element of $\mathcal{T}_{\exp}$.

Figure 3: **Total Regret incurred by linear versus sub-linear times of local exploration.** The total cost of the local exploration operation (lines 11–13 of Algorithm 3) is plotted. At each bin $j$ and epoch $\tau$, the total exploration time of Wang & Chen (2025) is $\Theta(n_{\tau,j})$, whereas ours is $\Theta(\sqrt{n_{\tau,j}})$, resulting in $\Theta(T^{\frac{2\beta-1}{2\beta+1}})$ regret and $\Theta(T^{\frac{\beta+1}{2\beta+1}})$ regret, respectively, as discussed in Remark 12.

## 5 HANDLING POLICY-INDUCED DISTRIBUTION SHIFT

Theorem 9 is stated under a stationarity assumption: the distribution of $x$ used to evaluate the expected gap matches the distribution of $\mathcal{D}$ used to fit the joint estimator. This is generally hard to use for regret-minimizing policies, which update adaptively and thus induce nonstationary distributions. To address this, we adopt the *distribution-shift* subroutine from the recent advance of Wang & Chen (2025) to design an epoch-wise algorithm, where the distribution mismatch between epochs is well controlled, so that Theorem 9 can be applied to the regret analysis.

Similar to Section 3, since the design of the algorithm is fully credits to Wang & Chen (2025), we present only the key properties needed for our application here and defer the full algorithm to Appendix C for completeness.

**Proposition 11** (Wang & Chen (2025)). *Suppose Assumptions 1 and 3 hold. Consider a stochastic policy $\Pi$ containing all conditional uniform stochastic policies:*

$$\Pi := \big\{\pi : \mathcal{C} \to \Delta([0, p_{\max}]) \mid \pi(\mathsf{c}) \sim \text{Unif}[\underline{\pi}(\mathsf{c}), \overline{\pi}(\mathsf{c})] \text{ for some } \underline{\pi}(\mathsf{c}) \leq \overline{\pi}(\mathsf{c}), \forall c \in \mathcal{C}\big\}.$$

*Then there exists an policy improvement oracle $\mathcal{A}$ (see Algorithm 4 in Appendix C for details), so that with any input tuple $\pi \in \Pi, \widehat{g}(\cdot) : \mathcal{C} \times [0, p_{\max}] \to \mathbb{R}, \text{CB}(\cdot) : \mathcal{C} \times [0, p_{\max}] \to \mathbb{R}$ satisfying*

*(S1) $p^\star(\mathsf{c}^\top\theta_*) \in \text{Supp}(\pi(\mathsf{c}))$ for all $\mathsf{c} \in \mathcal{C}$;*
*(S2) $|\widehat{g}(x) - g(x^\top\theta_0)| \leq \text{CB}(\mathsf{c}, p)$ for all $x = (\mathsf{c}^\top, p)^\top$ with $\mathsf{c} \in \mathcal{C}$ and $p \in [0, p_{\max}]$.*

*Its output $\pi' = \mathcal{A}(\pi, \widehat{g}, \text{CB}) \in \Pi$ satisfies:*

*(i) $p^\star(\mathsf{c}^\top\theta_*) \in \text{Supp}(\pi'(\mathsf{c}))$ for all $\mathsf{c} \in \mathcal{C}$;*
*(ii) $\mathbb{E}_{\mathsf{c}\sim P_{\mathcal{C}}, p\sim\pi'(\mathsf{c})}[R(\mathsf{c}, p^\star(\mathsf{c}^\top\theta_*)) - R(\mathsf{c}, p)] \leq \frac{1}{4}\mathbb{E}_{\mathsf{c}\sim P_{\mathcal{C}}, p\sim\pi(\mathsf{c})}[R(\mathsf{c}, p^\star(\mathsf{c}^\top\theta_*)) - R(\mathsf{c}, p)] + \frac{18L_r^3}{\sigma_r^2}\mathbb{E}_{\mathsf{c}\sim P_{\mathcal{C}}, p\sim\pi(\mathsf{c})}[\text{CB}(\mathsf{c}, p)].$*

Proposition 11 guarantees the existence of a policy improvement oracle $\mathcal{A}$ so that its output policy improves upon the input policy in the sense that it discounts the regret of the input policy by a factor of $1/4$, and adding an expectation of the confidence bounds evaluated under the *input policy's* distribution. This makes it possible to apply Theorem 9 for our regret analysis.

## 6 THE LPSP ALGORITHM AND REGRET RESULTS

In this section, we put all components introduced from Section 3 to 5 into an epoch-wise design to present our main algorithm in Algorithm 3. In Algorithm 3, after an initial phase for calculating the pilot estimator $\bar{\theta}_0$, the algorithm then enters an epoch-wise[5] phase to balance exploration and exploitation. At each epoch $\tau$, the algorithm posts prices based on a fixed stochastic policy $\pi^{(\tau-1)}$ determined by the previous epoch, with a portion of prices rounded for exploration. With such design, Theorem 9 can be applied to analyze the regret incurred by unrounded prices based on Proposition 11, and the key is to ensure the conditions in Theorem 9 holds, which relies on the *localized exploration procedure* we introduced in line 11-13 (see also Figure 2), as detailed below:

**Localized Exploration.** The goal of the localized exploration procedure in lines 11–13 is to construct the $\mathcal{T}_j^{\text{ro}}$ part so that condition ii) in Theorem 9 is satisfied. This procedure plays a key role on keep the design matrix of local polynomial regression well-conditioned even without diverse context assumption in Assumption 4. To see how this works, we provide a high-level analysis for $\bar{\theta}_0$ and leave the full details to Appendix G. Note that the normalized matrix $H\Lambda_{\tau,j}^{\text{ro}}(\bar{\theta}_0)H$ admits a Vandermonde decomposition $H\Lambda_{\tau,j}^{\text{ro}}(\bar{\theta}_0)H = Z_{\tau,j}Z_{\tau,j}^\top$ with

$$Z_{\tau,j} := \left[(1, \Delta_j(x_t, \bar{\theta}_0)/h, \ldots, (\Delta_j(x_t, \bar{\theta}_0)/h)^\ell)^\top : t \in \mathcal{T}_{\tau,j}^{\text{ro}}\right] \in \mathbb{R}^{(\ell+1)\times(L_j-1)}.$$

The lower bound on its singular values depends on the separation between $\Delta_j(x_t, \bar{\theta}_0)$ for different $t$ (Gautschi, 1963). The equi-partition rounding procedure then creates constant-level separations, which ensures that $\lambda_{\min}(H\Lambda_{\tau,j}^{\text{ro}}(\bar{\theta}_0)H) = \sigma_{\min}^2(Z_{\tau,j}) \gtrsim \lfloor L_j/\beta \rfloor$. Moreover, a basic calculation based on the definition of $\mathcal{T}_{\exp}$ yields that $L_j = \Theta(\sqrt{n_{\tau,j}})$, which leads to the eigenvalue lower bound $c_0\sqrt{n_{\tau,j}}$ for $H\Lambda_{\tau,j}(\bar{\theta}_0)H$, provided that $n_{\tau,j}$ exceeds a constant depending only on $c_0, \beta$.

**Remark 12** (Cost of Localized Exploration). *From our exploration schedule, we have the total exploration step at the $\tau$-th epoch is given by $\mathcal{O}(\sum_{j\in[M_\tau]}\sqrt{n_{\tau,j}}) = \mathcal{O}(\sqrt{N_\tau M_\tau}) = \mathcal{O}(N_\tau^{\frac{\beta+1}{2\beta+1}})$. Since Algorithm 3 stops after $\mathcal{O}(\log T)$ epochs, the total exploration steps amount to $\tilde{\mathcal{O}}(T^{\frac{\beta+1}{2\beta+1}})$. This leads to the $\tilde{\mathcal{O}}(T^{\frac{\beta+1}{2\beta+1}})$ total costs. In contrast, as discussed in Remark 10, Wang & Chen (2025) requires a linear-in-T number of exploration steps to satisfy their eigenvalue lower bound conditions. They use uni-modality to control the local exploration cost, which results in a per-epoch exploration regret of $\mathcal{O}(N_\tau M_\tau^{-2}) = \mathcal{O}(N_\tau^{\frac{2\beta-1}{2\beta+1}})$ leading to a total exploration cost of $\mathcal{O}(T^{\frac{2\beta-1}{2\beta+1}})$. While this rate matches their $\mathcal{O}(T^{3/5})$ regret when $\beta = 2$, it deteriorates when $\beta > 2$. This shows that shortening the local exploration length to $\sqrt{N_\tau M_\tau}$ is crucial for attaining the desired $\tilde{\mathcal{O}}(T^{\frac{\beta+1}{2\beta+1}})$ regret for general $\beta$, we illustrate the comparison of total exploration costs in Figure 3.*

We provide the implementation details of Algorithm 3 in Appendix K.1 and claim the regret guarantee of Algorithm 3 as the following:

**Theorem 13.** *Suppose Assumptions 1-4 hold for some $\beta \geq 1$, Algorithm 3 with hyper-parameters $c_0, N_0, C_1$ larger than some constant depending on $\beta$ satisfies*

$$\text{Regret}(T) \lesssim d^4 \log^{5/2}(T)T^{\frac{\beta+1}{2\beta+1}} + \text{Poly}(d^\beta, \log T).$$

**On the dependency on $d$.** Our regret bound stated in Theorem 13 has $d^4$ dependency in the leading order term and $\text{Poly}(d^\beta)$ dependency in the second order term.

The source of the $d^4$ term in the leading order is the in-distribution prediction error result in Theorem 9, which is the consequence on using self-normalized argument for bounding the $\mathbb{E}[\|\boldsymbol{v}_j(x, \widehat{\theta}_j)\|_{\Sigma_j^{-1}}]$ and the union bound taken over $x$ and $\theta$. On the other hand, the additive $\text{Poly}(d^\beta, \log T)$ term is a technical by-product of the covariance regularization used in our analysis. Specifically, to invoke Proposition 8, we require the matrix regularization level to satisfy $\zeta \gtrsim d^7 \log^{7/2}(T)\sqrt{T}$, so that the regularized empirical covariance dominates its population analogue, as required in Lemma 18.

---

[5]**Epoch-wise convention.** Throughout epoch $\tau$, we use the same constructions as in the subroutines but computed from the epoch-$\tau$ dataset $\mathcal{D}_\tau$ and partition $\{I_j\}_{j\in[M_\tau]}$: for any quantity $\mathcal{Q}_j(\cdot)$ defined earlier, we write $\mathcal{Q}_{\tau,j}(\cdot)$ for its epoch-$\tau$ version (e.g., $\Lambda_{\tau,j}, \widehat{\theta}_{\tau,j}, \text{CB}_{\tau,j}, \mathcal{T}_{\tau,j}$).

---

**Algorithm 3** Local Polynomial regression-based Semi-parametric Pricing(LPSP) Algorithm

---

1: **Inputs:** Smoothness parameter $\beta$, total time horizon $T$, hyer-parameter $c_0, N_0, C_1$.
2: **Initialization:** $\pi^{(0)}(\mathsf{c}) \leftarrow \mathrm{Unif}[0, p_{\max}]$ for all $\mathsf{c}$, pilot error level $\eta = T^{-\frac{\beta+1}{4\beta+2}}$ and exploration length $\mathsf{t}_\beta$ specified in Lemma 5. Epoch length schedule $N_\tau = 2^\tau N_0, \tau \geq 1$.
3:    // Initialization Phase as described in Section 3
4: Estimate the pilot estimator $\bar\theta$ and $\bar\theta_0 := (-\bar\theta^\top, 1)^\top$ using $\mathsf{t}_\beta$ time-steps through Algorithm 1.
5: **for** $\tau = 1, 2, \ldots$ until meets $t > T$ **do**
6:    // Decision Making & Data Collection
7:    **initialize** $\mathcal{D}_\tau := \emptyset, \mathcal{T}_{\exp} := \{k^2 : k \geq 1\}$, partition $[-V, V]$ into $M_\tau := \lceil N_\tau^{\frac{1}{2\beta+1}} \rceil$ equal bins $I_1, \ldots, I_{M_\tau}$, set $L_j = 1, s_j = 0, \forall j \in [M_\tau]$.
8:    **for** $s = 1, \ldots, N_\tau$ **do**
9:      Meets the $t$-th customer with context $\mathsf{c}_t$ and sample $p_t \sim \pi^{(\tau-1)}(\mathsf{c}_t)$.
10:      Compute $u_t := p_t - \mathsf{c}_t^\top \bar\theta$ and compute $j_t$ so that $u_t \in I_{j_t}, s_{j_t} \leftarrow s_{j_t} + 1$.
11:      **if** $s_{j_t}$ is the $L_j$-th element in $\mathcal{T}_{\exp}$ **then**
12:        $p_t \leftarrow p'_t$ with $p'_t - \mathsf{c}_t^\top \bar\theta$ is the $(L_j \bmod \lfloor\beta\rfloor)$-th $\lfloor\beta\rfloor$-equi-partition point of $I_j$.
13:        $L_j \leftarrow L_j + 1$
14:      Present $p_t$ to the customer and receive feedback $y_t$. Add $(\mathsf{c}_t, p_t, y_t)$ to $\mathcal{D}_\tau$.
15:      $t \leftarrow t + 1$
16:    Compute $\mathcal{T}_{\tau,j} := \{t \in \mathcal{D}_\tau : x_t^\top \bar\theta_0 \in I_j\}$
17:    // Joint Estimation Phase as described in Section 4
18:    Obtain joint estimators $\{\widehat{g}_{\tau,j}(\cdot \mid \widehat\theta_{\tau,j})\}_{j \in [M_\tau]}$ using $\mathcal{D}_\tau$ with Algorithm 2 and (2).
19:    Compute the glued estimator $\widehat{g}_\tau(x) := \sum_{j \in [M_\tau]} \mathbf{1}\{x^\top \bar\theta_0 \in I_j\}\widehat{g}_{\tau,j}(x \mid \widehat\theta_{\tau,j})$ and the glued confidence bound

$$\mathrm{CB}_\tau(x) := \sum_{j \in [M_\tau]} \mathbf{1}\{x^\top \bar\theta_0 \in I_j\} \cdot \begin{cases} C_1\left(\mathrm{Err}_{\tau,j}(x) + N_\tau^{-\frac{\beta}{2\beta+1}}\right) & \text{if } H\Lambda_{\tau,j}(\widehat\theta_{\tau,j})H \succeq c_0\sqrt{N_{\tau,j}}I, \\ 1 & \text{otherwise.} \end{cases}$$

   with $\mathrm{Err}_j$ defined as in right-hand-side of (3) and $N_{\tau,j} = |\mathcal{T}_{\tau,j}|$.
20:    // Policy Improvement via $\mathcal{A}$ described in Section 5
21:    Update $\pi^{(\tau)} \leftarrow \mathcal{A}(\pi^{(\tau-1)}, \widehat{g}_\tau, \mathrm{CB}_\tau)$

---

Since our algorithm ties $\zeta$ to the pilot accuracy via $\zeta \asymp \eta^{-2} \asymp T^{\frac{\beta+1}{2\beta+1}}$, the above condition may fail for $T^{\frac{1}{4\beta+2}} \lesssim d^7$, resulting in a finite burn-in phase of $T$ whose contribution is summarized by the $\mathrm{Poly}(d^\beta, \log T)$ term.

We would note that, despite the heavy $d$-dependency is included due to artificial reasons as explained above, during the running of our algorithm only a $\mathcal{O}(\sqrt{d})$-level confidence radius and a $(d^3/c_{\min})$-level initialization period used, this may leads to much better empirical performance regarding $d$, as provided in our simple simulation in Appendix K.2. We believe that more careful analysis can either improve the leading-order $d^4$-dependency or remove this burn-in without worsening the polynomial dependence on $d$ in the leading term, and we leave this refinement as an interesting future direction.

Finally, we would note that there are several future directions opened by our result, including:

**Removing the Strong Uni-modality Assumption 3.** While strong uni-modality does not drastically simplify the contextual pricing problem, we believe that the final step in this line of work will eventually match our regret upper bound without relying on this condition. In our analysis and algorithm design, the only part requiring Assumption 3 is the stationary subroutine we called from Wang & Chen (2025). Technically, we believe we have already moved a bit forward from Wang & Chen (2025) in the forced-exploration by removing the need for strong uni-modality in theirs argument via sharper analysis, which might be a foundation of fully removing this condition.

**Achieving Adaptivity on the Smoothness Parameter $\beta$.** Another promising direction building on our work is to study adaptivity to the smoothness parameter $\beta$. Following the progression seen in non-parametric bandits and pricing, where adaptive methods (Gur et al., 2022; Ye & Jiang, 2024; Gong & Zhang, 2025) build on earlier non-adaptive algorithms (Hu et al., 2020; Wang et al., 2021), we believe similar adaptivity can be achieved in our setting under additional self-similarity assumptions.

## ACKNOWLEDGMENTS

This work is supported by NSF (CCF-2312205, ECCS-2419564), ONR-13983263 and 2027 New York University Center for Global Economy and Business grant.

## DETAILS OF LLM USAGE

In writing this paper, the LLM was applied to polish our sentences and correct potential typos. In the experimental section (Appendix K.2), we also used an LLM to help organize the code structure and implement the benchmark algorithms.

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

## A    OTHER RELATED WORKS

**Dynamic Pricing.** There are extensive studies in dynamic pricing (Kleinberg & Leighton, 2003; Den Boer, 2015; Wang et al., 2014; Filippi et al., 2010; Broder & Rusmevichientong, 2012; Qiang & Bayati, 2016; Cohen et al., 2020). In the contextual setting with a linear demand model, $\tilde{\mathcal{O}}(\sqrt{T})$ regret can be obtained when the noise distribution is either fully known (Filippi et al., 2010; Ban & Keskin, 2021; Qiang & Bayati, 2016; Broder & Rusmevichientong, 2012) or assumed to belong to a parametric family (Javanmard & Nazerzadeh, 2019). The semi-parametric setting considered in our work and in Fan et al. (2024); Tullii et al. (2024); Bracale et al. (2025); Wang & Chen (2025); Luo et al. (2022; 2024); Xu & Wang (2022) generalizes this framework by allowing the noise distribution to be fully unknown. Among these works, apart from those listed in Table 1, Bracale et al. (2025); Luo et al. (2022) consider the $\beta = 1$ case and achieve $\tilde{\mathcal{O}}(T^{3/4})$ regret, while Luo et al. (2024) achieves $\tilde{\mathcal{O}}(T^{2/3})$ regret but additionally assumes an online estimation oracle. Finally, there also exist works that consider pricing with fully non-parametric demand (Chen & Gallego, 2021; Tullii et al., 2024; Javanmard et al., 2020) or other additional structures (Bu et al., 2020; Allouah et al., 2023; Keskin & Zeevi, 2014; Miao & Chao, 2021), which are beyond our scope.

**Semi-Parametric Regression and Single-Index Models.** Our setting is closely connected to semi-parametric single–index models, where an unknown low-dimensional index is coupled with a non-parametric link (Powell et al., 1989; Härdle et al., 1993; Ichimura, 1993). Classical work establishes root-$n$ estimation of the index under regularity and recovers the link via one-dimensional smoothing (Klein & Spady, 1993; Ichimura, 1993; Carroll et al., 1997). Foundational kernel procedures— Nadaraya–Watson and local polynomial regression—underpin these analyses, with well-understood uniform convergence and optimal-rate properties (Nadaraya, 1964; Watson, 1964; Stone, 1982). The literature also covers binary responses and generalized or partially linear single-index structures (Klein & Spady, 1993; Carroll et al., 1997), setting with discrete or irregular covariates (Horowitz & Härdle, 1996), and single–index coefficient models under strong mixing (Xia & Li, 1999). Comprehensive expositions and survey treatments can be found in (Györfi et al., 2002; Ruppert et al., 2003; Tsybakov, 2008; Horowitz, 2009). Despite this extensive theory, many results assume smooth design densities and emphasize asymptotics, assumptions that need not hold in contextual pricing where prices are policy–driven and the induced design can be irregular; hence the classical guarantees are informative but not directly applicable without further adaptation.

## B    PILOT ESTIMATION

In this section we introduce a simple pilot estimation stage under a mild diversity assumption on covariates $\lambda_{\min}(\mathbb{E}[\mathsf{c}_1 \mathsf{c}_1^\top]) \geq c_0/d$. We estimate $\theta_*$ by least squares. This procedure appeared in Fan et al. (2024); we include it here for completeness.

**Theorem 14.** *Let $\bar{\theta}$ be the output of Algorithm 1. Suppose that $\lambda_{\min}(\mathbb{E}\mathsf{c}_1\mathsf{c}_1^\top) \geq c_0/d$ for some $c_0 > 0$. Then there exists some constant $C_0 > 0$ such that for $\mathsf{t} \geq C_0 d$, the following holds with probability at least $1 - C_0\delta - 2e^{-\mathsf{t}/C_0 d^2}$:*

$$\|\bar{\theta} - \theta_*\| \leq C_0 \sqrt{\frac{d^3 \log(1/\delta)}{\mathsf{t}}}.$$

**Remark 15.** *For any target error level $\eta$, we may choose $\mathsf{t} = \tilde{\Theta}(d^3/\eta^2)$ to guarantee that $\|\bar{\theta}_0 - \theta_0\| \leq \eta$, which results in a regret of order $\tilde{\mathcal{O}}(d^3/\eta^2)$.*

*Proof of Theorem 14.* Let $H \equiv p_{\max}$ and for any $\theta \in \mathbb{R}^d$

$$\mathcal{L}(\theta) \equiv \frac{1}{\mathsf{t}} \sum_{t \in [\mathsf{t}]} (Hy_t - \mathsf{c}_t^\top \theta)^2.$$

We may compute the gradient and Hessian of $\mathcal{L}(\theta)$ as follows:

$$\nabla_\theta \mathcal{L}(\theta) = \frac{2}{\mathsf{t}} \sum_{t \in [\mathsf{t}]} (\mathsf{c}_t^\top \theta - Hy_t) \cdot \mathsf{c}_t \in \mathbb{R}^d,$$

$$\nabla_\theta^2 \mathcal{L}(\theta) = \frac{2}{t} \sum_{t \in [t]} c_t c_t^\top \in \mathbb{R}^{d \times d}.$$

A second order expansion yields that for some $\tilde{\theta}$ lying between $\bar{\theta}$ and $\theta_*$,

$$0 \geq \mathcal{L}(\bar{\theta}) - \mathcal{L}(\theta_*) = \langle \nabla_\theta \mathcal{L}(\theta_*), \bar{\theta} - \theta_* \rangle + \frac{1}{2} \langle \bar{\theta} - \theta_*, \nabla_\theta^2 \mathcal{L}(\tilde{\theta})(\bar{\theta} - \theta_*) \rangle$$

$$= \langle \nabla_\theta \mathcal{L}(\theta_*), \bar{\theta} - \theta_* \rangle + \frac{1}{t} \langle \bar{\theta} - \theta_*, \sum_{t \in [t]} c_t c_t^\top (\bar{\theta} - \theta_*) \rangle.$$

This implies that

$$\lambda_{\min}\left( \frac{1}{t} \sum_{t \in [t]} c_t c_t^\top \right) \cdot \|\bar{\theta} - \theta_*\|^2 \leq \frac{1}{t} \langle \bar{\theta} - \theta_*, \sum_{t \in [t]} c_t c_t^\top (\bar{\theta} - \theta_*) \rangle \leq \langle \nabla_\theta \mathcal{L}(\theta_*), \theta_* - \bar{\theta} \rangle$$

$$\leq \sqrt{d} \|\nabla_\theta \mathcal{L}(\theta_*)\|_\infty \cdot \|\bar{\theta} - \theta_*\|.$$

**Lower bounding** $\lambda_{\min}\left( \frac{1}{t} \sum_{t \in [t]} c_t c_t^\top \right)$. By the matrix concentration in (Vershynin, 2010, remark 5.40), there exists some constant $c_1 > 0$ such that for $t \geq c_1^{-1} d$, we have with probability at least $1 - 2e^{-c_1 t/d^2}$,

$$\lambda_{\min}\left( \frac{1}{t} \sum_{t \in [t]} c_t c_t^\top \right) \geq \lambda_{\min}\left( \mathbb{E} c_1 c_1^\top \right) - \left\| \frac{1}{t} \sum_{t \in [t]} c_t c_t^\top - \mathbb{E} c_1 c_1^\top \right\| \geq \frac{c_1}{d}.$$

**Upper bounding** $\|\nabla_\theta \mathcal{L}(\theta_0)\|_\infty$. Note that for any $t \in [t]$ and $i \in [d]$, we have $\left| (c_t^\top \theta_0 - H y_t) \cdot c_{t,i} \right| \leq 1$ and

$$\mathbb{E}(c_t^\top \theta_* - H y_t) \cdot c_{t,i}$$
$$= \mathbb{E}\left[ c_t^\top \theta_* \cdot c_{t,i} - H\mathbb{E}[y_t \mid c_t] \cdot c_{t,i} \right]$$
$$= \mathbb{E}\left[ c_t^\top \theta_* \cdot c_{t,i} - H\mathbb{E}[\mathbf{1}\{p_t \leq c_t^\top \theta_* + \xi_t\} \mid c_t] \cdot c_{t,i} \right]$$
$$= \mathbb{E}\left[ c_t^\top \theta_* \cdot c_{t,i} - \mathbb{E}[c_t^\top \theta_* + \xi_t \mid c_t] \cdot c_{t,i} \right] = 0.$$

Therefore, by applying Hoeffding's inequality and a union bound argument, there exists some constant $C_1 > 0$ such that for $t \geq C_1 d$, we have with probability at least $1 - C_1 \delta$ that

$$\|\nabla_\theta \mathcal{L}(\theta_*)\|_\infty \leq C_1 \sqrt{\frac{\log(1/\delta)}{t}}.$$

Combining the above estimates, we have for $t \geq (c_1^{-1} \vee C_1) d$,

$$\|\bar{\theta} - \theta_*\| \leq \frac{C_1}{c_1} \sqrt{\frac{d^3 \log(1/\delta)}{t}}$$

holds with probability at least $1 - C_1 \delta - 2e^{-c_1 t/d^2}$. The claim follows by adjusting constants. $\qquad \square$

## C  DISTRIBUTION SHIFT SUBROUTINE

For completeness, we include Algorithm 4 (adapted from Wang & Chen (2025)). Given a prior range $[\underline{\pi}(c), \overline{\pi}(c)]$, an estimator $\widehat{g}$, and an envelop $\Delta$, the subroutine returns a renewed range $[\underline{\pi}'(c), \overline{\pi}'(c)]$ for uniform pricing.

## D  PROOF OF PROPOSITION 6

Recall that for any $x$ with $x^\top \bar{\theta}_0 \in I_j$,

$$\widehat{g}_j(x \mid \theta) := U_j(x, \theta)^\top \Lambda_j^{-1}(\theta) \sum_{i \in \mathcal{T}_j} y_i U_j(x_i, \theta). \tag{5}$$

---

**Algorithm 4** A subroutine for handling distribution shift (adapted from Wang & Chen (2025))

---

1: **Inputs:** $\{(\underline{\pi}(\mathsf{c}), \overline{\pi}(\mathsf{c}))\}_{\mathsf{c}\in\mathcal{C}}$, $\Delta(\cdot,\cdot)$, $\widehat{g}(\cdot,\cdot)$, $K = 12L_r/\sigma_r$, $\kappa = \sqrt{L_r/\sigma_r}$
  ▷ Input parameters: prior policy that offers price $p \sim \mathrm{Unif}([\underline{\pi}(\mathsf{c}), \overline{\pi}(\mathsf{c})])$, error quantification $\Delta(\cdot,\cdot)$, estimated model $\widehat{g} : \mathcal{C} \times [0, p_{\max}] \to [0, 1]$.
2: **for every** $\mathsf{c} \in \mathcal{C}$ **do**
3:     Partition $J = [\underline{\pi}(\mathsf{c}), \overline{\pi}(\mathsf{c})]$ into $K$ intervals of equal lengths, denoted as $J_1, \ldots, J_K$; write $|J_k|$ for length and $J_k = [\underline{p}(k), \overline{p}(k)]$.
4:     **for** $k = 1, 2, \ldots, K$ **do**
  ▷ Estimated average reward $\widehat{r}(J_k)$ and its uncertainty quantification $\Delta(J_k)$
5:         $\widehat{r}(J_k) \leftarrow |J_k|^{-1} \int_{J_k} p\widehat{g}(\mathsf{c}, p)\mathrm{d}p; \quad \Delta(J_k) \leftarrow |J_k|^{-1} \int_{J_k} \Delta(\mathsf{c}, p)\mathrm{d}p.$
  ▷ Find the optimal price for context $c$ together with its uncertainty $\widehat{\Delta}$
6:         $\widehat{k} \leftarrow \arg\max_{k\in[K]} \widehat{r}(J_k); \quad \widehat{\Delta} \leftarrow \kappa\sqrt{|J_{\widehat{k}}|^2 + \max_{k\in[K]} \Delta(J_k)^2}.$
  ▷ Update the pricing range for context $c$, by stretching out $\widehat{\Delta}$ from the price interval $J_{\widehat{k}}$
7:     $[\underline{\pi}'(\mathsf{c}), \overline{\pi}'(\mathsf{c})] \leftarrow [\underline{p}(\widehat{k}) - \widehat{\Delta}, \ \overline{p}(\widehat{k}) + \widehat{\Delta}] \cap [0, p_{\max}].$
8: **return** $\{\underline{\pi}'(\mathsf{c}), \overline{\pi}'(\mathsf{c})\}_{\mathsf{c}\in\mathcal{C}}$          ▷ renewed policy is $p \sim \mathrm{Unif}([\underline{\pi}'(\mathsf{c}), \overline{\pi}'(\mathsf{c})])$

---

and $\boldsymbol{X}_j(x,\theta) := \left((x - \bar{x}_j)^\top, \ldots, \lfloor\beta\rfloor((x - \bar{x}_j)^\top\theta)^{\lfloor\beta\rfloor-1} \cdot (x - \bar{x}_j)^\top\right)^\top$. Recall also that $\boldsymbol{v}_j(x,\theta) := \boldsymbol{X}_j(x,\theta) - U_j(x,\theta)^\top\Lambda_j^{-1}(\theta)\sum_{i\in\mathcal{T}_j} U_j(x_i,\theta) \cdot \boldsymbol{X}_j(x_i,\theta)$ and $\Sigma_j(\theta) := \sum_{i\in\mathcal{T}_j} \boldsymbol{v}_j(x_i,\theta)\boldsymbol{v}_j(x_i,\theta)^\top$. For any $\theta$ with $\|\theta - \bar{\theta}_0\| \leq \eta$, we set

$$\boldsymbol{\delta}_j(\theta) := \left(g'(\bar{x}_j^\top\theta_0)(\theta - \theta_0)^\top, \ldots, \frac{g^{(\ell)}(\bar{x}_j^\top\theta_0)}{\ell!}(\theta - \theta_0)^\top\right)^\top.$$

Let $h = n^{-\frac{1}{2\beta+1}}$.

We first decompose the error as follows: for any $x$ such that $x^\top\bar{\theta}_0 \in I_j$,

$$\widehat{g}_j(x \mid \theta) - g(x^\top\theta_0) = \underbrace{\widehat{g}_j(x \mid \theta) - \widehat{g}_j(x \mid \theta_0)}_{:= \mathcal{I}_1} + \underbrace{\widehat{g}_j(x \mid \theta_0) - g(x^\top\theta_0)}_{:= \mathcal{I}_2}. \tag{6}$$

**Estimating $\mathcal{I}_1$.** We begin by expressing the response as $y_t = g(x_t^\top\theta_0) + \varepsilon_t$, where $\mathbb{E}[\varepsilon_t \mid x_t] = 0$. Then based on the closed form (5), we can further decompose $\mathcal{I}_1$ as follows:

$$\mathcal{I}_1 = \underbrace{\left\{U_j(x,\theta)^\top\sum_{t\in\mathcal{T}_j} g(x_t^\top\theta_0)\Lambda_j^{-1}(\theta)U_j(x_t,\theta) - U_j(x,\theta_0)^\top\sum_{t\in\mathcal{T}_j} g(x_t^\top\theta_0)\Lambda_j^{-1}(\theta_0)U_j(x_t,\theta_0)\right\}}_{:= \mathcal{I}_{11}}$$

$$+ \underbrace{\left\{U_j(x,\theta)^\top\sum_{t\in\mathcal{T}_j} \varepsilon_t\Lambda_j^{-1}(\theta)U_j(x_t,\theta) - \color{red}{U_j(x,\theta_0)^\top\sum_{t\in\mathcal{T}_j} \varepsilon_t\Lambda_j^{-1}(\theta_0)U_j(x_t,\theta_0)}\right\}}_{:= \mathcal{I}_{12}}.$$

Consider the $\lfloor\beta\rfloor$-order expansion of $g(\cdot)$ at $x^\top\theta$ for each $t$, we have by $\beta$-Hölder continuity and $\eta \leq h$,

$$g(x_t^\top\theta) = D_j^\top U_j(x_t,\theta) + \xi_t$$

for some $\xi_t = \mathcal{O}(h^\beta)$. Then we have

$$\mathcal{I}_{11} = U_j(x,\theta)^\top\sum_{t\in\mathcal{T}_j} \Lambda_j^{-1}(\theta)U_j(x_t,\theta)U_j(x_t,\theta_0)^\top D_j - U_j(x,\theta_0)^\top\sum_{t\in\mathcal{T}_j} \Lambda_j^{-1}(\theta_0)U_j(x_t,\theta_0)U_j(x_t,\theta_0)^\top D_j$$

$$+ U_j(x,\theta)^\top\sum_{t\in\mathcal{T}_j} \xi_t\Lambda_j^{-1}(\theta)U_j(x_t,\theta) - U_j(x,\theta_0)^\top\sum_{t\in\mathcal{T}_j} \xi_t\Lambda_j^{-1}(\theta_0)U_j(x_t,\theta_0)$$

$$\overset{\text{(i)}}{=} U_j(x,\theta)^\top\sum_{t\in\mathcal{T}_j} \Lambda_j^{-1}(\theta)U_j(x_t,\theta)U_j(x_t,\theta_0)^\top D_j - U_j(x,\theta_0)^\top\sum_{t\in\mathcal{T}_j} \Lambda_j^{-1}(\theta)U_j(x_t,\theta)U_j(x_t,\theta)^\top D_j$$

$$+ U_j(x,\theta)^\top \sum_{t\in\mathcal{T}_j} \xi_t \Lambda_j^{-1}(\theta) U_j(x_t,\theta) - U_j(x,\theta_0)^\top \sum_{t\in\mathcal{T}_j} \xi_t \Lambda_j^{-1}(\theta_0) U_j(x_t,\theta_0)$$

$$= U_j(x,\theta)^\top \sum_{t\in\mathcal{T}_j} \Lambda_j^{-1}(\theta) U_j(x_t,\theta)(U_j(x_t,\theta_0) - U_j(x_t,\theta))^\top D_j$$

$$+ (U_j(x,\theta) - U_j(x,\theta_0))^\top \underbrace{\sum_{t\in\mathcal{T}_j} \Lambda_j^{-1}(\theta) U_j(x_t,\theta) U_j(x_t,\theta)^\top}_{=I} D_j$$

$$+ U_j(x,\theta)^\top \sum_{t\in\mathcal{T}_j} \xi_t \Lambda_j^{-1}(\theta) U_j(x_t,\theta) - U_j(x,\theta_0)^\top \sum_{t\in\mathcal{T}_j} \xi_t \Lambda_j^{-1}(\theta_0) U_j(x_t,\theta_0). \tag{7}$$

Here, in (i) we have used the identity $\sum_{t\in\mathcal{T}_j} U_j(x_t,\theta) U_j(x_t,\theta)^\top = \Lambda_j(\theta)$ for all $\theta$. Now noticing that for any $1 \le s \le \lfloor \beta \rfloor$,

$$\Delta_j^s(x,\theta_0) - \Delta_j^s(x,\theta) = ((x-\bar{x}_j)^\top \theta_0)^s - ((x-\bar{x}_j)^\top \theta)^s$$
$$= s((x-\bar{x}_j)^\top \theta)^{s-1} \cdot (x-\bar{x}_j)^\top (\theta_0 - \theta) + \mathcal{O}(\eta^2),$$

we have

$$\left(U_j(x,\theta) - U_j(x,\theta_0)\right)^\top D_j$$

$$= \left(\Delta_j(x,\theta) - \Delta_j(x,\theta_0) \quad \cdots \quad \Delta_j^{\lfloor\beta\rfloor}(x,\theta) - \Delta_j^{\lfloor\beta\rfloor}(x,\theta_0)\right) \begin{pmatrix} g'(\bar{x}_j^\top \theta_0) \\ \vdots \\ \frac{1}{\lfloor\beta\rfloor!} g^{(\lfloor\beta\rfloor)}(\bar{x}_j^\top \theta_0) \end{pmatrix}$$

$$= \underbrace{\left((x-\bar{x}_j)^\top \quad \cdots \quad \lfloor\beta\rfloor((x-\bar{x}_j)^\top\theta)^{\lfloor\beta\rfloor-1} \cdot (x-\bar{x}_j)^\top\right)}_{=\boldsymbol{X}_j^\top(x,\theta)} \underbrace{\begin{pmatrix} g'(\bar{x}_j^\top \theta_0)(\theta-\theta_0) \\ \vdots \\ \frac{g^{(\lfloor\beta\rfloor)}(\bar{x}_j^\top \theta_0)}{\lfloor\beta\rfloor!}(\theta-\theta_0) \end{pmatrix}}_{=\boldsymbol{\delta}_j(\theta)} + \mathcal{O}(\eta^2),$$

we can further writing $\mathcal{I}_{11}$ as

$$\mathcal{I}_{11} = \boldsymbol{X}_j(x,\theta)^\top \boldsymbol{\delta}_j(\theta) + \mathcal{O}(\eta^2) - U_j(x,\theta)^\top \Lambda_j^{-1}(\theta) \sum_{t\in\mathcal{T}_j} U_j(x_t,\theta)\left(\boldsymbol{X}_j(x_t,\theta)^\top \boldsymbol{\delta}_j(\theta) + \mathcal{O}(\eta^2)\right)$$

$$= \underbrace{\left(\boldsymbol{X}_j(x,\theta) - U_j(x,\theta)^\top \Lambda_j^{-1}(\theta) \sum_{t\in\mathcal{T}_j} U_j(x_t,\theta) \cdot \boldsymbol{X}_j(x_t,\theta)\right)^\top}_{\boldsymbol{v}_j(x,\theta)} \boldsymbol{\delta}_j(\theta)$$

$$+ \mathcal{O}\left(\eta^2(1 + \sqrt{n_j}\|U_j(x,\theta)\|_{\Lambda_j^{-1}(\theta)})\right),$$

where we have used the Cauchy-Schwartz's inequality:

$$\left| \sum_{t\in\mathcal{T}_j} \mathcal{O}(\eta^2 U_j(x,\theta)^\top \Lambda_j^{-1}(\theta) U_j(x_t,\theta)) \right|$$

$$\lesssim \eta^2 \left( n_j U_j(x,\theta)^\top \Lambda_j^{-1}(\theta) \underbrace{\sum_{t\in\mathcal{T}_j} U_j(x_t,\theta) U_j(x_t,\theta)^\top}_{=\Lambda_j(\theta)} \Lambda_j^{-1}(\theta) U_j(x,\theta) \right)^{1/2}.$$

This completes the estimation for $\mathcal{I}_1$.

**Estimating $\mathcal{I}_2$.** For $\mathcal{I}_2$, we have

$$\mathcal{I}_2 = U_j(x,\theta_0)^\top \sum_{t\in\mathcal{T}_j} y_t \Lambda_j^{-1}(\theta_0) U_j(x_t,\theta_0) - g(x^\top \theta_0)$$

$$= \left\{ U_j(x,\theta_0)^\top \sum_{t\in\mathcal{T}_j} (y_t - g(x_t^\top \theta_0)) \Lambda_j^{-1}(\theta_0) U_j(x_t,\theta_0) \right\}$$

$$+ \left\{ U_j(x,\theta_0)^\top \sum_{t\in\mathcal{T}_j} g(x_t^\top\theta_0)\Lambda_j^{-1}(\theta_0)U_j(x_t,\theta_0) - g(x^\top\theta_0) \right\}$$

$$= \left\{ U_j(x,\theta_0)^\top \sum_{t\in\mathcal{T}_j} \varepsilon_t\Lambda_j^{-1}(\theta_0)U_j(x_t,\theta_0) \right\}$$

$$+ \left\{ U_j(x,\theta_0)^\top \sum_{t\in\mathcal{T}_j} g(x_t^\top\theta_0)\Lambda_j^{-1}(\theta_0)U_j(x_t,\theta_0) - g(x^\top\theta_0) \right\}$$

$$= \left\{ U_j(x,\theta_0)^\top \sum_{t\in\mathcal{T}_j} \varepsilon_t\Lambda_j^{-1}(\theta_0)U_j(x_t,\theta_0) \right\}$$

$$+ \left\{ U_j(x,\theta_0)^\top \sum_{t\in\mathcal{T}_j} \Lambda_j^{-1}(\theta_0)U_j(x_t,\theta_0)U_j(x_t,\theta)^\top D_j - g(x^\top\theta_0) \right\}$$

$$+ U_j(x,\theta_0)^\top \sum_{t\in\mathcal{T}_j} \xi_t\Lambda_j^{-1}(\theta_0)U_j(x_t,\theta_0)$$

$$= \textcolor{red}{U_j(x,\theta_0)^\top \sum_{t\in\mathcal{T}_j} \varepsilon_t\Lambda_j^{-1}(\theta_0)U_j(x_t,\theta_0)} + \textcolor{blue}{U_j(x,\theta_0)^\top \sum_{t\in\mathcal{T}_j} \xi_t\Lambda_j^{-1}(\theta_0)U_j(x_t,\theta_0)}$$

$$+ \underbrace{\left\{ U_j(x,\theta_0)^\top D_j - g(x^\top\theta_0) \right\}}_{=\mathcal{O}(h^\beta)}.$$

This completes the estimation for $\mathcal{I}_2$.

Combining two expansions and canceling red colored terms and blue colored terms, we have

$$\widehat{g}_j(x \mid \theta) - g(x^\top\theta_0) = \boldsymbol{v}_j(x,\theta)^\top\boldsymbol{\delta}_j(\theta) + U_j(x,\theta)^\top \sum_{t\in\mathcal{T}_j} \varepsilon_t\Lambda_j^{-1}(\theta)U_j(x_t,\theta)$$

$$+ U_j(x,\theta)^\top \sum_{t\in\mathcal{T}_j} \xi_t\Lambda_j^{-1}(\theta)U_j(x_t,\theta) + \mathcal{O}\big((h^\beta + \eta^2)(1 + \sqrt{n_j}\|U_j(x,\theta)\|_{\Lambda_j^{-1}(\theta)})\big).$$

Finally, by Cauchy Schwartz's inequality, we have the third term can be further bounded by

$$\Big| U_j(x,\theta)^\top \sum_{t\in\mathcal{T}_j} \xi_t\Lambda_j^{-1}(\theta)U_j(x_t,\theta) \Big| \lesssim \sqrt{h^{2\beta}n_j}\|U_j(x,\theta)\|_{\Lambda_j^{-1}(\theta)},$$

This concludes that proof. $\qquad\square$

## E    PROOF OF PROPOSITION 8

The proof of Proposition 8 relies on the following two lemmas, whose proofs are deferred to Appendix H. Recall that $h = n^{-\frac{1}{2\beta+1}}$.

**Lemma 16.** *For any $\theta$ with $\|\theta - \bar{\theta}_0\| \le \eta$, it holds that*

$$\widehat{g}_j(x \mid \theta) - g(x^\top\theta_0) = \boldsymbol{v}_j(x,\theta)^\top\boldsymbol{\delta}_j(\theta) + U_j(x,\theta)^\top \sum_{t\in\mathcal{T}_j} \varepsilon_t\Lambda_j^{-1}(\theta)U_j(x_t,\theta)$$

$$+ \mathcal{O}\big(h^\beta(1 + \sqrt{n_j}\|U_j(x,\theta)\|_{\Lambda_j^{-1}(\theta)})\big)$$

*uniformly for all $x$ with $x^\top\bar{\theta}_0 \in I_j$ and $j \in [M]$.*

**Lemma 17.** *For any $\theta$ with $\|\theta - \bar{\theta}_0\| \le \eta$, it holds with probability at least $1 - \mathcal{O}(n_j\delta)$ that*

$$\mathcal{L}_j(\theta) = \sum_{t\in\mathcal{T}_j} \big(y_t - \widehat{g}_j(x_t \mid \theta)\big)^2 = \sum_{t\in\mathcal{T}_j} \varepsilon_t^2 + \Theta\Big(\boldsymbol{\delta}_j(\theta)^\top\Sigma_j(\theta)\boldsymbol{\delta}_j(\theta)\Big)$$

$$+ \mathcal{O}\Big(\sqrt{\log(1/\delta)} \cdot \Big(\sqrt{\boldsymbol{\delta}_j(\theta)^\top\Sigma_j(\theta)\boldsymbol{\delta}_j(\theta) + n_jh^{2\beta}}\Big) + \log(1/\delta) + n_jh^{2\beta}\Big).$$

We now prove Proposition 8.

*Proof of Proposition 8.* In the proof below, we consider $x$ such that $x^\top \overline{\theta}_0 \in I_j$. Recall that $\widehat{\theta}_j = \arg\min_{\|\theta - \overline{\theta}_0\| \le \eta} \mathcal{L}_j(\theta)$ and we write $\widehat{\theta} \equiv \widehat{\theta}_j$ for simplicity.

It follows from Lemma 17 and a standard $\varepsilon$-net argument over $\theta$—leading to an multiplicative $d$ factor before $\log(1/\delta)$—that

$$
\mathcal{L}_j(\widehat{\theta}) = \sum_{t \in \mathcal{T}_j} \varepsilon_t^2 + \Theta\Big(\boldsymbol{\delta}_j(\widehat{\theta})^\top \Sigma_j(\widehat{\theta}) \boldsymbol{\delta}_j(\widehat{\theta})\Big)
$$
$$
+ \mathcal{O}\Big(\sqrt{d\log(1/\delta)} \cdot \Big(\sqrt{\boldsymbol{\delta}_j(\widehat{\theta})^\top \Sigma_j(\widehat{\theta}) \boldsymbol{\delta}_j(\widehat{\theta}) + n_j h^{2\beta}}\Big) + d\log(1/\delta) + n_j h^{2\beta}\Big)
$$

holds with probability at least $1 - \mathcal{O}(\delta \log n)$. As $\mathcal{L}_j(\widehat{\theta}) \le \mathcal{L}_j(\theta_0)$, we can derive that

$$
\boldsymbol{\delta}_j(\widehat{\theta})^\top \Sigma_j(\widehat{\theta}) \boldsymbol{\delta}_j(\widehat{\theta}) + n_j h^{2\beta}
$$
$$
\lesssim \sqrt{d\log(1/\delta)} \cdot \sqrt{\boldsymbol{\delta}_j(\widehat{\theta})^\top \Sigma_j(\widehat{\theta}) \boldsymbol{\delta}_j(\widehat{\theta}) + n_j h^{2\beta}} + d\log(1/\delta) + n_j h^{2\beta}.
$$

Using the facts that $a^2 \lesssim ba + c \implies a \lesssim b + \sqrt{c}, \forall a, b, c \ge 0$ and $\zeta\|\boldsymbol{\delta}_j(\widehat{\theta})\|^2 = \mathcal{O}(1)$, we obtain

$$
\sqrt{\boldsymbol{\delta}_j(\widehat{\theta})^\top \Sigma_j(\widehat{\theta}) \boldsymbol{\delta}_j(\widehat{\theta}) + n_j h^{2\beta}} \lesssim \sqrt{d\log(1/\delta)} + \Big(d\log(1/\delta) + n_j h^{2\beta}\Big)^{1/2}
$$
$$
\implies \sqrt{\boldsymbol{\delta}_j(\widehat{\theta})^\top (\Sigma_j(\widehat{\theta}) + \zeta I) \boldsymbol{\delta}_j(\widehat{\theta})} \lesssim \sqrt{d\log(1/\delta)} + \sqrt{n_j} h^\beta. \tag{8}
$$

On the other hand, by Lemma 16, we have

$$
\Big|(\widehat{g}_j(x \mid \widehat{\theta}) - g(x^\top \theta_0))\Big| \le \underbrace{\Big|\boldsymbol{v}_j(x, \widehat{\theta})^\top \boldsymbol{\delta}_j(\widehat{\theta})\Big|}_{\mathcal{Y}_1} + \underbrace{\Big|U_j(x, \widehat{\theta})^\top \sum_{t \in \mathcal{T}_j} \varepsilon_t \Lambda_j^{-1}(\widehat{\theta}) U_j(x_t, \widehat{\theta})\Big|}_{\mathcal{Y}_2}
$$
$$
+ \mathcal{O}\Big(h^\beta(1 + \sqrt{n_j} \|U_j(x, \widehat{\theta})\|_{\Lambda_j^{-1}(\widehat{\theta})})\Big).
$$

*Term $\mathcal{Y}_1$:* Using Cauchy-Schwarz together with (8) yields that

$$
\mathcal{Y}_1 \lesssim \Big(\sqrt{d\log(1/\delta)} + \sqrt{n_j} h^\beta\Big) \cdot \sqrt{\boldsymbol{v}_j(x, \widehat{\theta})^\top (\Sigma_j(\widehat{\theta}) + \zeta I)^{-1} \boldsymbol{v}_j(x, \widehat{\theta})}.
$$

*Term $\mathcal{Y}_2$:* Applying Hoeffding's inequality with an $\varepsilon$-net argument, we have with probability at least $1 - \mathcal{O}(\delta)$ that

$$
\mathcal{Y}_2 \lesssim \sqrt{d\log(1/\delta) U_j(x, \widehat{\theta})^\top \Lambda_j^{-1}(\widehat{\theta}) U_j(x, \widehat{\theta})}.
$$

Combining the estimates for $\mathcal{Y}_1$ and $\mathcal{Y}_2$ concludes the proof. $\qquad \square$

## F    PROOF OF THEOREM 9

For any $j \in [M]$, let $\mathcal{T}_j^{\mathsf{ra}}$ be the index set that collects the samples that are sampled i.i.d. from a stationary distribution $Q_j$ and $\mathcal{T}_j^{\mathsf{ro}} := \mathcal{T}_j \setminus \mathcal{T}_j^{\mathsf{ra}}$. Let $n_j^{\mathsf{ra}} = |\mathcal{T}_j^{\mathsf{ra}}|$ and $n_j^{\mathsf{ro}} = |\mathcal{T}_j^{\mathsf{ro}}|$. Then we have $\mathcal{T}_j = \mathcal{T}_j^{\mathsf{ra}} \cup \mathcal{T}_j^{\mathsf{ro}}$ and $n_j = n_j^{\mathsf{ra}} + n_j^{\mathsf{ro}}$. The *population* level quantities are defined as

$$
V_j(\theta) := \mathbb{E}_{z \sim Q_j} [U_j(z, \theta) \boldsymbol{X}_j(z, \theta)] + \frac{1}{n_j^{\mathsf{ra}}} \sum_{t \in \mathcal{T}_j^{\mathsf{ro}}} U_j(x_t, \theta) U_j(x_t, \theta)^\top,
$$

$$
\bar{\Lambda}_j(\theta) := \mathbb{E}_{z \sim Q_j} [U_j(z, \theta) U_j(z, \theta)^\top] + \frac{1}{n_j^{\mathsf{ra}}} \sum_{t \in \mathcal{T}_j^{\mathsf{ro}}} U_j(x_t, \theta) U_j(x_t, \theta)^\top
$$

$$
\bar{\boldsymbol{v}}_j(x, \theta) := \boldsymbol{X}_j(x, \theta) - U_j^\top(x, \theta) \bar{\Lambda}_j^{-1}(\theta) V_j(\theta), \quad \bar{\Sigma}_j(\theta) := \sum_{t \in \mathcal{T}_j} \bar{\boldsymbol{v}}_j(x_t, \theta) \bar{\boldsymbol{v}}_j(x_t, \theta)^\top.
$$

**Lemma 18.** *Assume the same conditions as in Theorem 9. It holds with probability at least $1 - \mathcal{O}(\delta)$ that uniformly for all $x$ such that $x^\top \bar{\theta}_0 \in I_j$ and $\theta$ such that $\|\theta - \bar{\theta}_0\| \leq \eta$,*

*(i)* $\|\boldsymbol{v}_j(x, \theta) - \bar{\boldsymbol{v}}_j(x, \theta)\|_2 \lesssim d^{7/2} \log^{3/2}(1/\delta) \cdot \left( n_j^{-1/4} \|U_j(x, \theta)\|_{\bar{\Lambda}_j^{-1}(\theta)} + n_j^{1/4} \|U_j(x, \theta)\|_{\Lambda_j^{-1}(\theta)} \right).$

*(ii) Moreover, if $\zeta \asymp \eta^{-2} = \Omega(d^7 \log^{7/2}(1/\delta)\sqrt{n_j})$, we have*

$$\boldsymbol{v}_j(x, \theta)^\top (\Sigma_j(\theta) + \zeta I)^{-1} \boldsymbol{v}_j(x, \theta) \lesssim \bar{\boldsymbol{v}}_j(x, \theta)^\top (\bar{\Sigma}_j(\theta) + \zeta I)^{-1} \bar{\boldsymbol{v}}_j(x, \theta)$$
$$+ d^7 \log^3(1/\delta) \cdot (n_j^{-1/2} \zeta^{-1} \|U_j(x, \theta)\|_{\bar{\Lambda}_j^{-1}(\theta)}^2 + n_j^{1/2} \zeta^{-1} \|U_j(x, \theta)\|_{\Lambda_j^{-1}(\theta)}^2).$$

The proof of Lemma 18 is deferred to Appendix I.1. Now we prove Theorem 9.

*Proof of Theorem 9.* Recall that $h = n^{-\frac{1}{2\beta+1}}$ and

$$\mathrm{Err}_j(x) := \left( \sqrt{d\log(1/\delta)} + \sqrt{n_j} h^\beta \right) \cdot \left( \|\boldsymbol{v}_j(x, \widehat{\theta}_j)\|_{(\Sigma_j(\widehat{\theta}_j) + \zeta I)^{-1}} + \|U_j(x, \widehat{\theta}_j)\|_{\Lambda_j^{-1}(\widehat{\theta}_j)} \right)$$

With Lemma 18 and the fact $\zeta \gtrsim n_j^{1/2}$, we can now define the population-level confidence bound as

$$\overline{\mathrm{Err}}_j(x) := \left( \sqrt{d\log(1/\delta)} + \sqrt{n_j} h^\beta \right) \cdot \left( \|\bar{\boldsymbol{v}}_j(x, \widehat{\theta}_j)\|_{(\bar{\Sigma}_j(\widehat{\theta}_j) + \zeta I)^{-1}} + \|U_j(x, \widehat{\theta}_j)\|_{\Lambda_j^{-1}(\widehat{\theta}_j)} \right)$$
$$+ d^{7/2} \log^{3/2}(1/\delta) \cdot \left( n_j^{-1} \|U_j(x, \widehat{\theta}_j)\|_{\bar{\Lambda}_j^{-1}(\widehat{\theta}_j)}^2 + (1 + n_j h^{2\beta}) \cdot \|U_j(x, \widehat{\theta}_j)\|_{\Lambda_j^{-1}(\widehat{\theta}_j)}^2 \right)^{1/2},$$

It follows that for any $x \sim Q_j$, if $\zeta \asymp \eta^{-2} = \Omega(d^7 \log^{7/2}(1/\delta)\sqrt{n_j})$,

$$\mathbb{E}_{x \sim Q_j}[\mathrm{Err}_j(x)] \lesssim \mathbb{E}_{x \sim Q_j}[\overline{\mathrm{Err}}_j(x)].$$

Let

$$\mathrm{Err}_{j,1}(x) := \sqrt{\bar{\boldsymbol{v}}_j(x, \widehat{\theta}_j)^\top \bar{\Sigma}_j^{-1}(\widehat{\theta}_j) \bar{\boldsymbol{v}}_j(x, \widehat{\theta}_j)},$$
$$\mathrm{Err}_{j,2}(x) := \sqrt{n_j^{-1} \|U_j(x, \widehat{\theta}_j)\|_{\bar{\Lambda}_j^{-1}(\widehat{\theta}_j)}^2 + (1 + n_j h^{2\beta}) \cdot \|U_j(x, \widehat{\theta}_j)\|_{\Lambda_j^{-1}(\widehat{\theta}_j)}^2}.$$

We then have

$$\mathbb{E}_{x \sim Q_j}[\overline{\mathrm{Err}}_j(x)] = \left( \sqrt{d\log(1/\delta)} + \sqrt{n_{\tau,j}} h^\beta \right) \mathbb{E}_{x \sim Q_j}[\mathrm{Err}_{j,1}(x)]$$
$$+ d^{7/2} \log^{3/2}(1/\delta) \mathbb{E}_{x \sim Q_j}[\mathrm{Err}_{j,2}(x)].$$

**Bounding $\mathbb{E}_{x \sim Q_j}[\overline{\mathrm{Err}}_{j,1}(x)]$.** As in proof of Lemma 18, we decompose $\bar{\Sigma}_j$ into

$$\bar{\Sigma}_j(\widehat{\theta}) = \sum_{t \in \mathcal{T}_j^{\mathrm{ro}}} \bar{\boldsymbol{v}}_j(x_t, \widehat{\theta}_j) \bar{\boldsymbol{v}}_j(x_t, \widehat{\theta}_j)^\top + \underbrace{\sum_{t \in \mathcal{T}_j^{\mathrm{ra}}} \bar{\boldsymbol{v}}_j(x_t, \widehat{\theta}_j) \bar{\boldsymbol{v}}_j(x_t, \widehat{\theta}_j)^\top + \zeta I}_{:=\bar{\Sigma}_j^{\mathrm{ra}}(\widehat{\theta}_j)} \succeq \bar{\Sigma}_j^{\mathrm{ra}}(\widehat{\theta}_j).$$

Then by Jensen's inequality,

$$\mathbb{E}_{x \sim Q_j}[\overline{\mathrm{Err}}_{j,1}(x)] \leq \sqrt{\mathbb{E}_{x \sim Q_j}[\bar{\boldsymbol{v}}_j(x, \widehat{\theta}_j)^\top \bar{\Sigma}_j^{-1}(\widehat{\theta}_j) \bar{\boldsymbol{v}}_j(x, \widehat{\theta}_j)]}$$

$$\leq \sqrt{\mathbb{E}_{x \sim Q_j}[\bar{\boldsymbol{v}}_j(x, \widehat{\theta}_j)^\top (\bar{\Sigma}_j^{\mathrm{ra}})^{-1}(\widehat{\theta}_j) \bar{\boldsymbol{v}}_j(x, \widehat{\theta}_j)]}$$

$$= \left( \underbrace{\left\langle (\bar{\Sigma}_j^{\mathrm{ra}})^{-1}(\widehat{\theta}_j), \mathbb{E}_{x \sim Q_j}[\bar{\boldsymbol{v}}_j(x, \widehat{\theta}_j) \bar{\boldsymbol{v}}_j(x, \widehat{\theta}_j)^\top] - \frac{1}{n_j^{\mathrm{ra}}} (\bar{\Sigma}_j^{\mathrm{ra}}(\widehat{\theta}_j) - \zeta I) \right\rangle}_{=_{(i)} \mathcal{O}(\sqrt{d\log(1/\delta)/n_j})} + \underbrace{\frac{1}{n_j^{\mathrm{ra}}} \left\langle (\bar{\Sigma}_j^{\mathrm{ra}})^{-1}(\widehat{\theta}_j), (\bar{\Sigma}_j^{\mathrm{ra}}(\widehat{\theta}_j) - \zeta I) \right\rangle}_{=\mathcal{O}(1/n_j^{\mathrm{ra}})} \right)^{1/2}$$

$$\lesssim_{(ii)} \left( \zeta^{-1} \sqrt{d\log(1/\delta)/n_j} + 1/n_j \right)^{1/2} \lesssim \sqrt{d\log(1/\delta)/n_j}.$$

with probability at least $1 - \delta$. Where (i) is by matrix Hoeffding's inequality and a simple union bound, (ii) is by $\bar{\Sigma}_j^{\mathrm{ra}}(\widehat{\theta}) \succeq \zeta I \succeq \eta^{-2} I$ and $n_j^{\mathrm{ra}} \asymp n_j$. Therefore, we have with probability at least $1 - \delta$,

$$\left( \sqrt{d\log(1/\delta)} + \sqrt{n_{\tau,j}} h^\beta \right) \mathbb{E}_{x \sim Q_j}[\overline{\mathrm{Err}}_{j,1}(x)] \lesssim d \log^{3/2}(1/\delta) \cdot n_j^{-1/2} + h^\beta \sqrt{d\log(1/\delta)}.$$

**Bounding $\mathbb{E}_{x\sim Q_j}[\overline{\mathrm{Err}}_{j,2}(x)]$.** By Jensen's inequality, for every $j$ with $n_j > 0$,

$$\mathbb{E}_{x\sim Q_j}[\overline{\mathrm{Err}}_{j,2}(x)]$$

$$\lesssim (1+\sqrt{n_j}h^\beta)\cdot\sqrt{n_j^{-1}\mathbb{E}_{x\sim Q_j}[\|U_j(x,\widehat{\theta}_j)\|^2_{\bar{\Lambda}_j^{-1}(\widehat{\theta}_j)}] + \mathbb{E}_{x\sim Q_j}[\|U_j(x,\widehat{\theta}_j)\|^2_{\Lambda_j^{-1}(\widehat{\theta}_j)}]}$$

$$= (1+\sqrt{n_j}h^\beta)\cdot\Big(n_j^{-1}\underbrace{\langle\bar{\Lambda}_j^{-1}(\widehat{\theta}_j),\bar{\Lambda}_j(\widehat{\theta}_j)\rangle}_{=\mathcal{O}(1)} + \underbrace{\big\langle\bar{\Lambda}_j(\widehat{\theta}_j)-\frac{1}{n_j^{\mathsf{ra}}}\Lambda_j^{\mathsf{ra}}(\widehat{\theta}_j),\Lambda_j^{-1}(\widehat{\theta}_j)\big\rangle}_{=_{(\mathrm{i})}\mathcal{O}(\sqrt{d\log(1/\delta)/n_j})} + \frac{1}{n_j^{\mathsf{ra}}}\underbrace{\langle\Lambda_j^{\mathsf{ra}}(\widehat{\theta}_j),\Lambda_j^{-1}(\widehat{\theta}_j)\rangle}_{=_{(\mathrm{ii})}\mathcal{O}(1)}\Big)^{1/2}$$

$$\lesssim_{(\mathrm{iii})} \sqrt{d\log(1/\delta)/n_j} + h^\beta\sqrt{d\log(1/\delta)}$$

with probability at least $1-\delta$. Where (i) is by matrix's Hoeffding's inequality and a simple union bound, (ii) is by

$$\langle\Lambda_j^{\mathsf{ra}}(\widehat{\theta}_j),\Lambda_j^{-1}(\widehat{\theta}_j)\rangle = \sum_{t\in\mathcal{T}_j^{\mathsf{ra}}} U_j(x_t,\widehat{\theta}_j)^\top\Lambda_j^{-1}(\widehat{\theta})U_j(x_t,\widehat{\theta}_j)$$

$$\leq \sum_{t\in\mathcal{T}_j^{\mathsf{ra}}} U_j(x_t,\widehat{\theta}_j)^\top\Lambda_j^{\mathsf{ra};-1}(\widehat{\theta}_j)U_j(x_t,\widehat{\theta}_j) = \mathcal{O}(1),$$

(iii) is by $c\sqrt{n_j}I_j \preceq \Lambda_j(\widehat{\theta}_j)$.

Now we arrive at the same bound as in $\overline{\mathrm{Err}}_{j,1}$:

$$d^{7/2}\log^{3/2}(1/\delta)\mathbb{E}_{x\sim Q_j}[\overline{\mathrm{Err}}_{j,2}(x)] \lesssim d^4\log^2(1/\delta)\cdot n_j^{-1/2} + d^4\log^2(1/\delta)\cdot h^\beta,$$

thus the same argument leads to the desired result.

Combining above bounds, we have the desired result. $\qquad\square$

## G  PROOF OF THEOREM 13

Let $n_{\tau,j} := |\mathcal{T}_{\tau,j}|$. According to the algorithm design, the regret splits into the contribution from uniform sampling and the rounding term:

$$\mathrm{Regret}_T(\pi) \lesssim \sum_{\tau=0}^{\lceil\log_2 T\rceil} N_\tau\mathbb{E}_\pi\Big[\mathbb{E}_{\mathsf{c}\sim P_\mathcal{C},p\sim\pi^{(\tau)}(\mathsf{c})}[r(\mathsf{c}^\top\theta_*,p^\star(\mathsf{c}^\top\theta_*))-r(\mathsf{c}^\top\theta_*,p)]\Big]$$

$$+ \sum_{\tau=0}^{\lceil\log_2 T\rceil}\mathbb{E}\Big[\sum_{j=1}^{M_\tau}\mathbf{1}\{x^\top\bar{\theta}_0\in I_j\}\sqrt{n_{\tau,j}}\Big].$$

The first sum is the uniform sampling regret. The second sum is the rounding regret.

**Bounding the rounding regret.** By Cauchy-Schwarz inequality,

$$\sum_{\tau=0}^{\lceil\log_2 T\rceil}\mathbb{E}\Big[\sum_{j=1}^{M_\tau}\mathbf{1}\{x^\top\bar{\theta}_0\in I_j\}\sqrt{n_{\tau,j}}\Big] \leq \sum_{\tau=0}^{\lceil\log_2 T\rceil}\sqrt{N_\tau} \leq N_0\frac{\sqrt{2T}-1}{\sqrt{2}-1} \lesssim \sqrt{T}.$$

**Bounding the sampling regret.** For any $\tau\geq 0$, recall that

$$\mathrm{Err}_{\tau,j}(x) := \big(\sqrt{d\log(1/\delta)}+\sqrt{n_{\tau,j}}N_\tau^{-\frac{\beta}{2\beta+1}}\big)$$

$$\times\big(\|\boldsymbol{v}_{\tau,j}(x,\widehat{\theta}_{\tau,j})\|_{(\Sigma_{\tau,j}(\widehat{\theta}_{\tau,j})+\zeta I)^{-1}} + \|U_j(x,\widehat{\theta}_{\tau,j})\|_{\Lambda_{\tau,j}^{-1}(\widehat{\theta}_{\tau,j})}\big).$$

For any $\tau\geq 0$, we define the event

$$\Omega_1^\tau := \Big\{|\widehat{g}^\tau(x)-g(x^\top\theta_0)| \leq \sum_{j\in[M_\tau]}\mathbf{1}\{x^\top\bar{\theta}_0\in I_j\}\mathrm{Err}_j(x) + N_\tau^{-\frac{\beta}{2\beta+1}},$$

$$p^\star(\mathsf{c}^\top\theta_*) \in \mathrm{Supp}(\pi^{(\tau)}(\mathsf{c})), \ \forall \mathsf{c} \in \mathcal{C}, p \in [0, p_{\max}], \text{ and } x = (\mathsf{c}^\top, p)^\top\}.$$

It follows from Proposition 8 that $\mathbb{P}\big(\bigcap_{\tau=0}^{\lceil \log_2 T \rceil} \Omega_1^\tau\big) \geq 1 - \delta T \log_2 T$.

On the other hand, by the definition of the rounding samples, for any unit vector $u \in \mathbb{R}^{\lfloor \beta \rfloor + 1}$,

$$n_{\tau,j} u^\top H \bar{\Lambda}_{\tau,j}(\widehat{\theta}_{\tau,j}) H u \geq \sum_{k \in [\lfloor \sqrt{n_{\tau,j}}/(\lfloor \beta \rfloor + 1)\rfloor]} u^\top \tilde{Z}_k^\top \tilde{Z}_k u$$

$$\gtrsim \left\lfloor \frac{\sqrt{n_{\tau,j}}}{\lfloor \beta \rfloor + 1} \right\rfloor \cdot \min_{k \in [\lfloor \sqrt{n_{\tau,j}}/(\lfloor \beta \rfloor + 1)\rfloor]} \sigma_{\min}^2(\tilde{Z}_k) \gtrsim \sqrt{n_{\tau,j}}, \qquad (9)$$

where $\tilde{Z}_k$ is a $(\lfloor \beta \rfloor + 1)$ dimensional Vandermonde matrix with $\Theta(1)$ separation and in the penultimate step we have used (Gautschi, 1963, Theorem 1) to derive that $\sigma_{\min}^2(\tilde{Z}_k) \gtrsim 1$. Similarly, we have

$$u^\top H \Lambda_{\tau,j}(\widehat{\theta}_{\tau,j}) H u \gtrsim \sqrt{n_{\tau,j}}. \qquad (10)$$

So when $\zeta \asymp \eta^{-2} = \Omega(d^7 \log^{7/2}(1/\delta)\sqrt{T})$, Theorem 9 is applicable and with $P_{\tau,j}$ being the distribution of $x = (\mathsf{c}^\top, p)^\top$ such that $\mathsf{c} \sim P_\mathcal{C}, p \sim \pi^{(\tau)}(\mathsf{c})$ at $\tau$-th epoch condition on $x$ such that $x^\top \overline{\theta}_0 \in I_j$, we have $\mathbb{P}\big(\bigcap_{\tau=0}^{\lceil \log_2 T \rceil} \Omega_2^\tau\big) \geq 1 - \delta T \log_2 T$, where for $\tau \geq 0$,

$$\Omega_2^\tau := \Big\{ \mathbb{E}_{x \sim P_{\tau,j}}[\mathrm{Err}_{\tau,j}(x)] \lesssim d^4 \log^2(1/\delta)(n_{\tau,j}^{-1/2} + N_\tau^{-\frac{\beta}{2\beta+1}}), \ \forall j \in [M_\tau] \Big\}.$$

Let $P_\tau$ be the distribution of $x = (\mathsf{c}^\top, p)^\top$ such that $\mathsf{c} \sim P_\mathcal{C}, p \sim \pi^{(\tau)}(\mathsf{c})$ at $\tau$-th epoch. The Chernoff's bound yields that $\mathbb{P}\big(\bigcap_{\tau=0}^{\lceil \log_2 T \rceil} \Omega_3^\tau\big) \geq 1 - \delta T \log_2 T$, where for $\tau \geq 0$,

$$\Omega_3^\tau := \Big\{ n_{\tau,j} + 1 \gtrsim \max\big\{ \mathbb{E}[n_{\tau,j}] - \sqrt{\mathbb{E}[n_{\tau,j}]\log(1/\delta)}, 0 \big\} + 1, \ \forall j \in [M_\tau] \Big\}.$$

Therefore, on the event $\bigcap_{\tau=0}^{\lceil \log_2 T \rceil}(\Omega_2^\tau \cap \Omega_3^\tau)$, we can derive that

$$\mathbb{E}_{x \sim P_\tau}\Bigg[ \sum_{j \in [M_\tau]} \mathbf{1}\{x^\top \overline{\theta}_0 \in I_j\} \cdot \mathrm{Err}_{\tau,j}(x) \Bigg] = \sum_{j \in [M_\tau]} P_\tau(x^\top \overline{\theta}_0 \in I_j) \mathbb{E}_{x \sim P_{\tau,j}}[\mathrm{Err}_{\tau,j}(x)]$$

$$\lesssim d^4 \log^2(1/\delta) \sum_{j \in [M_\tau]} P_\tau(x^\top \overline{\theta}_0 \in I_j)(n_{\tau,j}^{-1/2} + N_\tau^{-\frac{\beta}{2\beta+1}})$$

$$\lesssim d^4 \log^2(1/\delta) \cdot \Bigg( \sqrt{\sum_{j \in [M_\tau]} \frac{P_\tau(x^\top \overline{\theta}_0 \in I_j)}{n_{\tau,j}+1}} + N_\tau^{-\frac{\beta}{2\beta+1}} \Bigg)$$

$$= d^4 \log^2(1/\delta) \cdot \Bigg( \sqrt{\frac{1}{N_\tau} \sum_{j \in [M_\tau]} \frac{\mathbb{E}[n_{\tau,j}]}{n_{\tau,j}+1}} + N_\tau^{-\frac{\beta}{2\beta+1}} \Bigg)$$

$$\overset{(i)}{\lesssim} d^4 \log^{5/2}(1/\delta) \cdot \Bigg( \sqrt{\frac{M_\tau}{N_\tau}} + N_\tau^{-\frac{\beta}{2\beta+1}} \Bigg), \qquad (11)$$

where in (i), we have used the elementary inequality

$$\frac{a}{\max\{a - \sqrt{ac}, 0\} + 1} \lesssim c + 1, \quad \forall a, c > 0.$$

Write $\Omega := \bigcap_{\tau=0}^{\lceil \log_2 T \rceil}(\Omega_1^\tau \cap \Omega_2^\tau \cap \Omega_3^\tau)$. By Proposition 11, for $\tau \geq 1$,

$$\mathbb{E}_\pi \Big[ \mathbb{E}_{\mathsf{c} \sim P_\mathcal{C}, p \sim \pi^{(\tau)}(\mathsf{c})}[r(\mathsf{c}^\top\theta_*, p^\star(\mathsf{c}^\top\theta_*)) - r(\mathsf{c}^\top\theta_*, p)] \Big]$$

$$= \mathbb{E}_\pi \Big[ \mathbb{E}_{\mathsf{c} \sim P_\mathcal{C}, p \sim \pi^{(\tau)}(\mathsf{c})}[r(\mathsf{c}^\top\theta_*, p^\star(\mathsf{c}^\top\theta_*)) - r(\mathsf{c}^\top\theta_*, p)] \cdot \mathbf{1}\{\Omega\} \Big]$$

$$+ \mathbb{E}_\pi \Big[ \mathbb{E}_{\mathsf{c} \sim P_\mathcal{C}, p \sim \pi^{(\tau)}(\mathsf{c})}[r(\mathsf{c}^\top\theta_*, p^\star(\mathsf{c}^\top\theta_*)) - r(\mathsf{c}^\top\theta_*, p)] \cdot \mathbf{1}\{\Omega^c\} \Big]$$

$$\leq \frac{1}{4}\mathbb{E}_\pi\Big[\mathbb{E}_{\mathsf{c}\sim P_\mathcal{C}, p\sim\pi^{(\tau-1)}(\mathsf{c})}[r(\mathsf{c}^\top\theta_*, p^\star(\mathsf{c}^\top\theta_*)) - r(\mathsf{c}^\top\theta_*, p)]\cdot\mathbf{1}\{\Omega\}\Big]$$

$$+ \mathbb{E}_\pi\Big[\mathbb{E}_{x\sim P_\tau}\Big[\sum_{j\in[M_\tau]}\mathbf{1}\{x^\top\bar\theta_0\in I_j\}\cdot\mathrm{Err}_{\tau,j}(x)\Big]\cdot\mathbf{1}\{\Omega\}\Big] + p_{\max}\mathbb{P}(\Omega^c)$$

$$\leq \frac{1}{4}\mathbb{E}_\pi\Big[\mathbb{E}_{\mathsf{c}\sim P_\mathcal{C}, p\sim\pi^{(\tau-1)}(\mathsf{c})}[r(\mathsf{c}^\top\theta_*, p^\star(\mathsf{c}^\top\theta_*)) - r(\mathsf{c}^\top\theta_*, p)]\Big]$$

$$+ \mathcal{O}\bigg(d^4\log^{5/2}(1/\delta)\cdot\bigg(\sqrt{\frac{M_\tau}{N_\tau}} + N_\tau^{-\frac{\beta}{2\beta+1}}\bigg) + \delta T\log_2 T\bigg).$$

By choosing $\delta$ sufficiently small, i.e., $\delta = T^{-10}$, we arrive at

$$\mathbb{E}_\pi\Big[\mathbb{E}_{\mathsf{c}\sim P_\mathcal{C}, p\sim\pi^{(\tau)}(\mathsf{c})}[r(\mathsf{c}^\top\theta_*, p^\star(\mathsf{c}^\top\theta_*)) - r(\mathsf{c}^\top\theta_*, p)]\Big]$$

$$\leq \frac{1}{4}\mathbb{E}_\pi\Big[\mathbb{E}_{\mathsf{c}\sim P_\mathcal{C}, p\sim\pi^{(\tau-1)}(\mathsf{c})}[r(\mathsf{c}^\top\theta_*, p^\star(\mathsf{c}^\top\theta_*)) - r(\mathsf{c}^\top\theta_*, p)]\Big] + \mathcal{O}\bigg(d^4\log^{5/2}(T)\cdot N_\tau^{-\frac{\beta}{2\beta+1}}\bigg).$$

Iterating the above bound, we have

$$\sum_{\tau=0}^{\lceil\log_2 T\rceil} N_\tau\mathbb{E}_\pi\Big[\mathbb{E}_{\mathsf{c}\sim P_\mathcal{C}, p\sim\pi^{(\tau)}(\mathsf{c})}[r(\mathsf{c}^\top\theta_*, p^\star(\mathsf{c}^\top\theta_*)) - r(\mathsf{c}^\top\theta_*, p)]\Big]$$

$$\lesssim N_0 + d^4\log^{5/2}(T)\cdot T^{\frac{\beta+1}{2\beta+1}} \lesssim d^4\log^{5/2}(T)\cdot T^{\frac{\beta+1}{2\beta+1}}.$$

Combining the rounding regret and the sampling regret, we have if $\eta^{-2} = T^{\frac{\beta+1}{2\beta+1}} = \Omega(d^7\log^{7/2}(T)\sqrt{T}) \implies T^{\frac{1}{4\beta+2}} = \Omega(d^7\log^{7/2}(T))$,

$$\mathrm{Regret}(T) \lesssim d^4\log^{5/2}(T)\cdot T^{\frac{\beta+1}{2\beta+1}}.$$

Adding the burn-in time term completes the proof. $\qquad\square$

# H  PROOFS OF LEMMAS IN APPENDIX E

## H.1  PRELIMINARY NOTATIONS

For each $j\in[M]$, we denote $D_j = (D_{j0},\dots,D_{j\lfloor\beta\rfloor})^\top \in \mathbb{R}^{\lfloor\beta\rfloor+1}$ with $D_{js} = \frac{g^{(s)}(\bar x_j^\top\theta_0)}{s!}$ for $s\in\{0,1,\dots,\lfloor\beta\rfloor\}$, under which the local polynomial expansion of $g$ at $\bar x_j^\top\theta$ up to $\lfloor\beta\rfloor$ order can be written as $D_j^\top U_j(x,\theta)$.

## H.2  PROOF OF LEMMA 16

The claim follows from Proposition 6. With $\eta = \mathcal{O}(n^{-\frac{\beta+1}{4\beta+2}})$ we have $\eta^2 = \mathcal{O}(h^{(\beta+1)}) = \mathcal{O}(h^\beta)$.
$\qquad\square$

## H.3  PROOF OF LEMMA 17

Noticing that

$$\mathcal{L}_j(\theta) = \sum_{t\in\mathcal{T}_j}(y_t - \widehat g_j(x_t\mid\theta))^2 = \sum_{t\in\mathcal{T}_j}(g(x_t^\top\theta_0) - \widehat g_j(x_t\mid\theta) + \varepsilon_t)^2$$

$$= \underbrace{\sum_{t\in\mathcal{T}_j}\varepsilon_t^2}_{\text{independent of }\theta} + \underbrace{2\sum_{t\in\mathcal{T}_j}\varepsilon_t\big[\widehat g_j(x_t\mid\theta) - g(x_t^\top\theta_0)\big]}_{:=\mathcal{E}_1(\theta)} + \underbrace{\sum_{t\in\mathcal{T}_j}\big[g(x_t^\top\theta_0) - \widehat g_j(x_t\mid\theta)\big]^2}_{:=\mathcal{E}_2(\theta)}.$$

**Lemma 19** (Bounds on $\mathcal{E}_1$). *With probability at least $1 - \mathcal{O}(\delta)$, we have*

$$\mathcal{E}_1(\theta) = \mathcal{O}\bigg(\sqrt{\log(1/\delta)}\cdot\bigg(\sqrt{\boldsymbol\delta_j(\theta)^\top\Sigma_j(\theta)\boldsymbol\delta_j(\theta) + n_j h^{2\beta}} + 1\bigg)\bigg),$$

**Lemma 20** (Bounds on $\mathcal{E}_2$). *With probability at least $1 - \mathcal{O}(n_j \delta)$, we have*

$$\mathcal{E}_2(\theta) = \Theta\Big(\boldsymbol{\delta}_j(\theta)^\top \Sigma_j(\theta)\boldsymbol{\delta}_j(\theta)\Big) + \mathcal{O}\big(\log(1/\delta) + n_j h^{2\beta}\big).$$

The proofs for the above lemmas are defered to Section H.3.1. Combining the bounds for $\mathcal{E}_1(\theta), \mathcal{E}_2(\theta)$, we get the desired result. $\qquad\square$

### H.3.1 PROOFS OF LEMMAS 19 AND 20

*Proof of Lemma 19.* By Lemma 16,

$$\mathcal{E}_1(\theta) = \underbrace{2\sum_{t\in\mathcal{T}_j} \varepsilon_t \boldsymbol{v}_j(x_t,\theta)^\top \boldsymbol{\delta}_j(\theta) + \sum_{t\in\mathcal{T}_j} \varepsilon_t \cdot \mathcal{O}(h^\beta) + \sum_{t\in\mathcal{T}_j} \varepsilon_t \cdot \sqrt{n_j}\|U_j(x_t,\theta)\|_{\Lambda_j^{-1}(\theta)} \cdot \mathcal{O}(h^\beta)}_{:=\mathcal{E}_{11}(\theta)}$$

$$+ \underbrace{2\sum_{t\in\mathcal{T}_j} \varepsilon_t U_j(x_t,\theta)^\top \sum_{t'\in\mathcal{T}_j} \varepsilon_{t'} \Lambda_j^{-1}(\theta)U_j(x_{t'},\theta)}_{:=\mathcal{E}_{12}(\theta)}.$$

*Term $\mathcal{E}_{11}(\theta)$:* Noticing that condition on $\{x_t\}_{t=1}^n$, $\{\varepsilon\}_{t=1}^n$ are mutually independent and zero-mean random variables. By Hoeffding's inequality, with probability at least $1 - \mathcal{O}(\delta)$,

$$\mathcal{E}_{11}(\theta) \lesssim \sqrt{\log(1/\delta)} \cdot \left(\sqrt{\boldsymbol{\delta}_j(\theta)^\top \Sigma_j(\theta)\boldsymbol{\delta}_j(\theta)} + \sqrt{n_j}h^\beta + \Big(\sum_{t\in\mathcal{T}_j} n_j U_j(x_t,\theta)^\top \Lambda_j^{-1}(\theta)U_j(x_t,\theta)\Big)^{1/2} h^\beta\right).$$

Using the fact that

$$\sum_{t\in\mathcal{T}_j} U_j(x_t,\theta)^\top \Lambda_j^{-1}(\theta)U_j(x_t,\theta) = \mathrm{tr}\left(\Lambda_j^{-1}(\theta)\sum_{t\in\mathcal{T}_j} U_j(x_t,\theta)U_j(x_t,\theta)^\top\right) = \lfloor\beta\rfloor + 1, \qquad (12)$$

we arrive at

$$\mathcal{E}_{11}(\theta) \lesssim \sqrt{\log(1/\delta)} \cdot \left(\sqrt{\boldsymbol{\delta}_j(\theta)^\top \Sigma_j(\theta)\boldsymbol{\delta}_j(\theta)} + \sqrt{n_j}h^\beta\right).$$

*Term $\mathcal{E}_{12}(\theta)$:* For $j \in [M]$, let

$$\boldsymbol{\varepsilon}_j := (\varepsilon_t)_{t\in\mathcal{T}_j}^\top \in \mathbb{R}^{n_j}, \boldsymbol{C}_j := \big(U_j(x_t,\theta)^\top \Lambda_j^{-1}(\theta)U_j(x_{t'},\theta)\big)_{t,t'\in\mathcal{T}_j} \in \mathbb{R}^{n_j\times n_j}.$$

Then $\mathcal{E}_{12}(\theta)$ can be rewritten as

$$\mathcal{E}_{12}(\theta) = \boldsymbol{\varepsilon}_j^\top \boldsymbol{C}_j \boldsymbol{\varepsilon}_j$$

Applying the standard Hanson-Wright inequality leads to

$$\mathbb{P}\Big(\big|\mathcal{E}_{12}(\theta) - \mathbb{E}\mathcal{E}_{12}(\theta)\big| > u\Big) \le 2\exp\left(-c\min\left\{\frac{u}{\|\boldsymbol{C}_j\|_2}, \frac{u^2}{\|\boldsymbol{C}_j\|_F^2}\right\}\right)$$

for some absolute constant $c > 0$. On the other hand, using the facts that

$$\max_{j\in[M]}\|\boldsymbol{C}_j\|_2 \le \max_{j\in[M]}\|\boldsymbol{C}_j\|_F = \max_{j\in[M]}\sqrt{\sum_{t,t'\in\mathcal{T}_j}\big[U_j(x_t,\theta)^\top \Lambda_j^{-1}(\theta)U_j(x_{t'},\theta)\big]^2}$$

$$= \left(\sum_{t\in\mathcal{T}_j} U_j(x_t,\theta)^\top \Lambda_j^{-1}(\theta)\underbrace{\sum_{t'\in\mathcal{T}_j} U_j(x_{t'},\theta)U_j(x_{t'},\theta)^\top}_{=\Lambda_j(\theta)}\Lambda_j^{-1}(\theta)U_j(x_t,\theta)\right)^{1/2}$$

$$= \left(\sum_{t\in\mathcal{T}_j} U_j(x_t,\theta)^\top \Lambda_j^{-1}(\theta)U_j(x_t,\theta)\right)^{1/2} \overset{(12)}{=} \sqrt{\lfloor\beta\rfloor + 1}$$

and $\|C_j\|_F^2 = \mathcal{O}(1)$, we may then select $u \gtrsim \sqrt{\log(1/\delta)} + \log(1/\delta)$ to obtain that with probability at least $1 - \mathcal{O}(\delta)$,

$$\left| \mathcal{E}_{12}(\theta) - \mathbb{E}\mathcal{E}_{12}(\theta) \right| \lesssim \sqrt{\log(1/\delta)}.$$

Finally, as

$$\mathbb{E}\mathcal{E}_{12}(\theta) = \sum_{t \in \mathcal{T}_j} \mathbb{E}[\varepsilon_t^2] U_j(x_t, \theta)^\top \Lambda_j^{-1}(\theta) U_j(x_t, \theta) \leq \max_{t \in \mathcal{T}_j} \mathbb{E}[\varepsilon_t^2] \cdot \langle \Lambda_j^{-1}, \Lambda_j \rangle = \mathcal{O}(1),$$

we have with probability at least $1 - \mathcal{O}(\delta)$

$$\mathcal{E}_{12}(\theta) \lesssim \sqrt{\log(1/\delta)}$$

This completes the proof of Lemma 19. $\qquad \square$

*Proof of Lemma 20.* It follows from Lemma 16 and the elementary inequality $\frac{1}{2}a^2 - 4b^2 \leq (a+b)^2 \leq 2a^2 + 2b^2$ that

$$\mathcal{E}_2(\theta) = \Theta\Big( \boldsymbol{\delta}_j(\theta)^\top \Sigma_j(\theta) \boldsymbol{\delta}_j(\theta) \Big) + \mathcal{O}\Big( \underbrace{\sum_{t \in \mathcal{T}_j} \Big[ U_j(x_t, \theta)^\top \sum_{t' \in \mathcal{T}_j} \varepsilon_{t'} \Lambda_j^{-1}(\theta) U_j(x_{t'}, \theta) \Big]^2}_{:=\mathcal{E}_{21}(\theta)} \Big)$$

$$+ \mathcal{O}\Big( \underbrace{\sum_{t \in \mathcal{T}_j} h^{2\beta} + \sum_{t \in \mathcal{T}_j} n_j h^{2\beta} U_j(x_t, \theta)^\top \Lambda_j^{-1}(\theta) U_j(x_t, \theta)}_{:=\mathcal{E}_{22}(\theta)} \Big).$$

*Term $\mathcal{E}_{21}(\theta)$:* By Hoeffding's inequality, we have with probability at least $1 - \mathcal{O}(n\delta)$,

$$\mathcal{E}_{21}(\theta) \lesssim \log(1/\delta) \sum_{t \in \mathcal{T}_j} U_j(x_t, \theta)^\top \Lambda_j^{-1}(\theta) U_j(x_t, \theta) \overset{(12)}{\lesssim} \log(1/\delta).$$

*Term $\mathcal{E}_{22}(\theta)$:* It can be directly bounded that

$$\mathcal{E}_{22}(\theta) \lesssim n_j h^{2\beta} + n_j h^{2\beta} \sum_{t \in \mathcal{T}_j} U_j(x_t, \theta)^\top \Lambda_j^{-1}(\theta) U_j(x_t, \theta) \overset{(12)}{\lesssim} n_j h^{2\beta}.$$

This completes the proof of Lemma 20. $\qquad \square$

# I    PROOF OF LEMMA IN APPENDIX F

## I.1    PROOF OF LEMMA 18

(i) Recall that $H = \mathrm{diag}(1, h, \dots, h^\ell) \in \mathbb{R}^{(\ell+1) \times (\ell+1)}$ and $h = n^{-\frac{1}{2\beta+1}}$. Note that

$$\bar{\boldsymbol{v}}_j(x, \theta) - \boldsymbol{v}_j(\boldsymbol{x}, \theta)$$

$$= \underbrace{(HU_j(x, \theta))^\top \Big[ (H\Lambda_j(\theta)H)^{-1} - (n_j^{\mathrm{ra}} H\bar{\Lambda}_j(\theta)H)^{-1} \Big] \sum_{t \in \mathcal{T}_j} HU_j(x_t, \theta) \boldsymbol{X}_j(x_t, \theta)}_{:=\mathcal{R}_1}$$

$$+ \underbrace{(HU_j(x, \theta))^\top (H\bar{\Lambda}_j(\theta)H)^{-1} \Big[ \frac{1}{n_j^{\mathrm{ra}}} \sum_{t \in \mathcal{T}_j} HU_j(x_t, \theta) \boldsymbol{X}_j(x_t, \theta) - HV_j(\theta) \Big]}_{:=\mathcal{R}_2}$$

*Term $\mathcal{R}_2$:* For any unit vector $w$, let $Y_t := (HU_j(x, \theta))^\top (H\bar{\Lambda}_j(\theta)H)^{-1} HU_j(x_t, \theta) \boldsymbol{X}_j(x_t, \theta) w$. Then we have

$$\mathcal{R}_2 w = \frac{1}{n_j^{\mathrm{ra}}} \sum_{t \in \mathcal{T}_j^{\mathrm{ra}}} \Big( Y_t - \mathbb{E}_{x_t \sim Q_j}[Y_t] \Big).$$

As $\lambda_{\min}(n_j^{\mathsf{ra}} H \bar{\Lambda}_j(\theta) H) \wedge \lambda_{\min}(H \Lambda_j(\theta) H) \gtrsim \sqrt{n_j}$, we have $|Y_t| \lesssim n_j^{1/4} \|U_j(x, \theta)\|_{\bar{\Lambda}_j^{-1}(\theta)}$. Using further

$$\mathbb{E}_{x_t \sim Q_j}[Y_t^2] = \mathbb{E}_{x_t \sim Q_j}[(\boldsymbol{X}_j(x_t, \theta) w)^2 U_j(x, \theta)^\top \bar{\Lambda}_j^{-1}(\theta) U_j(x_t, \theta) U_j(x_t, \theta)^\top \bar{\Lambda}_j^{-1}(\theta) U_j(x_t, \theta)]$$

$$\lesssim U_j(x, \theta)^\top \bar{\Lambda}_j^{-1}(\theta) \left( \mathbb{E}_{z \sim Q_j}[U_j(z, \theta) U_j(z, \theta)^\top] + \frac{1}{n_j^{\mathsf{ra}}} \sum_{t \in \mathcal{T}_j^{\mathsf{ro}}} U_j(x_t, \theta) U_j(x_t, \theta)^\top \right) \bar{\Lambda}_j^{-1}(\theta) U_j(x, \theta)$$

$$= U_j(x, \theta)^\top \bar{\Lambda}_j^{-1}(\theta) U_j(x, \theta)$$

together with matrix Bernstein's inequality, we can obtain that with probability at least $1 - \mathcal{O}(\delta)$,

$$\mathcal{R}_2 w \lesssim \frac{\log(1/\delta)}{\sqrt{n_j}} \|U_j(x, \theta)\|_{\bar{\Lambda}_j^{-1}(\theta)}.$$

Now taking union bound over $w$ in the unit ball, we can get with probability at least $1 - \mathcal{O}(\delta)$,

$$\|\mathcal{R}_2\| \lesssim \frac{d \log(1/\delta)}{\sqrt{n_j}} \|U_j(x, \theta)\|_{\bar{\Lambda}_j^{-1}(\theta)}.$$

*Term $\mathcal{R}_1$:* We first decompose the term as follows:

$$\mathcal{R}_1 = \underbrace{(H U_j(x, \theta))^\top \left[ (H \Lambda_j(\theta) H)^{-1} - (n_j^{\mathsf{ra}} H \bar{\Lambda}_j(\theta) H)^{-1} \right] \sum_{t \in \mathcal{T}_j^{\mathsf{ra}}} H U_j(x_t, \theta) \boldsymbol{X}_j(x_t, \theta)}_{\mathcal{R}_{11}}$$

$$+ \underbrace{(H U_j(x, \theta))^\top \left[ (H \Lambda_j(\theta) H)^{-1} - (n_j^{\mathsf{ra}} H \bar{\Lambda}_j(\theta) H)^{-1} \right] \sum_{t \in \mathcal{T}_j^{\mathsf{ro}}} H U_j(x_t, \theta) \boldsymbol{X}_j(x_t, \theta)}_{\mathcal{R}_{12}}$$

For $\mathcal{R}_{12}$, for any unit vector $w$, by Bernstein's inequality, we have with probability at least $1 - \mathcal{O}(\delta)$,

$$\mathcal{R}_{12} w = \frac{1}{n_j^{\mathsf{ra}}} U_j(x, \theta)^\top \bar{\Lambda}_j^{-1}(\theta) \left[ n_j^{\mathsf{ra}} \bar{\Lambda}_j(\theta) - \Lambda_j(\theta) \right] \Lambda_j^{-1}(\theta) \sum_{t \in \mathcal{T}_j^{\mathsf{ro}}} U_j(x_t, \theta) \boldsymbol{X}_j(x_t, \theta) w$$

$$\lesssim \frac{\log(1/\delta)}{n_j^{1/4}} \|U_j(x, \theta)\|_{\bar{\Lambda}_j^{-1}(\theta)},$$

where we have also used the condition that $\lambda_{\min}(n_j^{\mathsf{ra}} H \bar{\Lambda}_j(\theta) H) \wedge \lambda_{\min}(H \Lambda_j(\theta) H) \gtrsim \sqrt{n_j}$.

For $\mathcal{R}_{11}$, we have for any unit vector $w$, with $\bar{V}_j(\theta) := \mathbb{E}_{z \sim Q_j} U_j(z, \theta) \boldsymbol{X}_j(z, \theta)$,

$$\mathcal{R}_{11} w = \underbrace{(H U_j(x, \theta))^\top \left[ (H \Lambda_j(\theta) H)^{-1} - (n_j^{\mathsf{ra}} H \bar{\Lambda}_j(\theta) H)^{-1} \right] \sum_{t \in \mathcal{T}_j^{\mathsf{ra}}} (H U_j(x_t, \theta) \boldsymbol{X}_j(x_t, \theta) - H \bar{V}_j(\theta)) w}_{=: \mathcal{R}_{111}}$$

$$+ \underbrace{(H U_j(x, \theta))^\top \left[ (H \Lambda_j(\theta) H)^{-1} - (n_j^{\mathsf{ra}} H \bar{\Lambda}_j(\theta) H)^{-1} \right] \sum_{t \in \mathcal{T}_j^{\mathsf{ra}}} H \bar{V}_j(\theta) w}_{=: \mathcal{R}_{112}}.$$

As $\lambda_{\min}(n_j^{\mathsf{ra}} H \bar{\Lambda}_j(\theta) H) \wedge \lambda_{\min}(H \Lambda_j(\theta) H) \gtrsim \sqrt{n_j}$,

$$|\mathcal{R}_{111}| \leq n_j^{-1/2} \cdot \|U_j(x, \theta)\|_{\Lambda_j^{-1}(\theta)} \cdot \|n_j^{\mathsf{ra}} H \bar{\Lambda}_j(\theta) H - H \Lambda_j(\theta) H\| \cdot \|\sum_{t \in \mathcal{T}_j^{\mathsf{ra}}} Y_j'(x_t, \theta) - \mathbb{E}_{x_t \sim Q_j} Y_j'(x_t, \theta)\|,$$

where $Y_j'(x_t, \theta) := (n_j^{\mathsf{ra}} H \bar{\Lambda}_j(\theta) H)^{-1/2} \sum_{t \in \mathcal{T}_j^{\mathsf{ra}}} H U_j(x_t, \theta) \boldsymbol{X}_j(x_t, \theta) w$. Using

$$\mathbb{E}_{x_t \sim Q_j} \|Y_j'(x_t, \theta)\|^2 = \mathbb{E}_{x_t \sim Q_j} \left[ w^\top \boldsymbol{X}(x_t, \theta)^\top (H U_j(x_t, \theta))^\top (n_j^{\mathsf{ra}} H \bar{\Lambda}_j(\theta) H)^{-1} (H U_j(x_t, \theta)) \boldsymbol{X}(x_t, \theta) w \right]$$

$$\lesssim \mathbb{E}_{x_t \sim Q_j}[(HU_j(x_t, \theta))^\top (n_j^{\mathsf{ra}} H\bar{\Lambda}_j(\theta)H)^{-1}(HU_j(x_t, \theta))]$$

$$\lesssim \frac{1}{n_j}\mathbb{E}_{z \sim Q_j}[\|U_j(z, \theta)\|^2_{\bar{\Lambda}_j^{-1}(\theta)}] \lesssim \frac{1}{n_j}$$

and $\|Y_j'(x_t, \theta)\| \lesssim \sigma_{\min}^{-1/2}(n_j^{\mathsf{ra}} H\bar{\Lambda}_j(\theta)H) \lesssim n_j^{-1/4}$ together with the Bernstein's inequality, we have with probability at least $1 - \mathcal{O}(\delta)$,

$$\|\sum_{t \in \mathcal{T}_j^{\mathsf{ra}}} Y_j'(x_t, \theta) - \mathbb{E}_{x_t} Y_j'(x_t, \theta)\| = \mathcal{O}\Big(\sqrt{\log(1/\delta)} + n_j^{-1/4}\log(1/\delta)\Big).$$

Moreover, by the matrix Bernstein's inequality, we have with probability at least $1 - \mathcal{O}(\delta)$ that

$$\|n_j^{\mathsf{ra}} H\bar{\Lambda}_j(\theta)H - H\Lambda_j(\theta)H\|_2 \lesssim \sqrt{n_j \log(1/\delta)}.$$

Combining the estimates in the above displays, we have with probability at least $1 - \mathcal{O}(\delta)$ that

$$|\mathcal{R}_{111}| \lesssim \log(1/\delta)\|U_j(x, \theta)\|_{\Lambda_j^{-1}(\theta)}.$$

Next, we consider the bound for $\mathcal{R}_{112}$. Note that

$$\mathcal{R}_{112} = (HU_j(x, \theta))^\top (H\Lambda_j(\theta)H)^{-1}\Big[n_j^{\mathsf{ra}} H\bar{\Lambda}_j(\theta)H - H\Lambda_j(\theta)H\Big](H\bar{\Lambda}_j(\theta)H)^{-1}H\bar{V}_j(\theta)w$$

$$= -(HU_j(x, \theta))^\top (H\Lambda_j(\theta)H)^{-1}\Big[\sum_{t \in \mathcal{T}_j^{\mathsf{ra}}}\big(1 - \mathbb{E}_{x_t \sim Q_j}\big)[HU_j(x_t, \theta)(HU_j(x_t, \theta))^\top]\Big](H\bar{\Lambda}_j(\theta)H)^{-1}H\bar{V}_j(\theta)w.$$

Let $\mathsf{Z}_t := \mathbb{E}_{z \sim Q_j}[\boldsymbol{X}_j(z, \theta)w \cdot HU_j(x_t, \theta)(HU_j(x_t, \theta))^\top (H\bar{\Lambda}_j(\theta)H)^{-1}HU_j(z, \theta)]$, the above term can be rewritten as

$$\mathcal{R}_{112} = -(HU_j(x, \theta))^\top (H\Lambda_j(\theta)H)^{-1}\sum_{t \in \mathcal{T}_j^{\mathsf{ra}}}(\mathsf{Z}_t - \mathbb{E}_{x_t \sim Q_j}\mathsf{Z}_t)$$

$$= \underbrace{-(HU_j(x, \theta))^\top (n_j^{\mathsf{ra}} H\bar{\Lambda}_j(\theta)H)^{-1}\sum_{t \in \mathcal{T}_j^{\mathsf{ra}}}(\mathsf{Z}_t - \mathbb{E}_{x_t \sim Q_j}\mathsf{Z}_t)}_{:= \mathcal{R}_{1121}}$$

$$+ \underbrace{(HU_j(x, \theta))^\top ((n_j^{\mathsf{ra}} H\bar{\Lambda}_j(\theta)H)^{-1} - (H\Lambda_j(\theta)H)^{-1})\sum_{t \in \mathcal{T}_j^{\mathsf{ra}}}(\mathsf{Z}_t - \mathbb{E}_{x_t \sim Q_j}\mathsf{Z}_t)}_{:= \mathcal{R}_{1122}}.$$

For $\mathcal{R}_{1121}$, note that

$$|\tilde{\mathsf{Z}}_t| := |(HU_j(x, \theta))^\top (n_j^{\mathsf{ra}} H\bar{\Lambda}_j(\theta)H)^{-1}\mathsf{Z}_t| \lesssim \frac{1}{\sqrt{n_j}}\|U_j(x, \theta)\|_{\bar{\Lambda}_j(\theta)}$$

and

$$\mathbb{E}[\tilde{\mathsf{Z}}_t^2] = \mathbb{E}_{x_t}[(\mathbb{E}_z[\boldsymbol{X}_j(z, \theta)w \cdot U_j(x, \theta)^\top (n_j^{\mathsf{ra}} \bar{\Lambda}_j(\theta))^{-1}U_j(x_t, \theta)U_j(x_t, \theta)^\top \bar{\Lambda}_j^{-1}(\theta)U_j(z, \theta)|z^\top \bar{\theta} \in I_j])^2]$$

$$\lesssim_{(i)} \mathbb{E}_{x_t}[(U_j(x, \theta)^\top (n_j^{\mathsf{ra}} \bar{\Lambda}_j)^{-1}U_j(x_t, \theta)U_j(x_t, \theta)^\top \bar{\Lambda}_j^{-1}(\theta)U_j(x_t, \theta)U_j(x_t, \theta)^\top (n_j^{\mathsf{ra}} \bar{\Lambda}_j)^{-1}U_j(x, \theta)]$$

$$\leq n_j^{-2}\|U_j(x, \theta)\|^2_{\bar{\Lambda}_j^{-1}(\theta)} \cdot \mathbb{E}_{x_t}[(U_j(x_t, \theta)^\top \bar{\Lambda}_j^{-1}(\theta)U_j(x_t, \theta))^2]$$

$$\lesssim_{(ii)} n_j^{-2}\|U_j(x, \theta)\|^2_{\bar{\Lambda}_j^{-1}(\theta)} \cdot \mathbb{E}_{x_t}[\|U_j(x_t, \theta)\|^4_{\bar{\Lambda}_j^{-1}(\theta)}] \lesssim_{(iii)} n_j^{-3/2}\|U_j(x, \theta)\|^2_{\bar{\Lambda}_j^{-1}(\theta)},$$

$$(13)$$

where (i) is by Jensen's inequality and $|\boldsymbol{X}_j(z, \theta)w| = \mathcal{O}(1)$; (ii) is by $\mathbb{E}_z[\|U_j(z, \theta)\|^2_{\bar{\Lambda}_j^{-1}(\theta)}] = \mathcal{O}(1)$, (iii) is by

$$\mathbb{E}_{x_t}[\|U_j(x_t, \theta)\|^4_{\bar{\Lambda}_j^{-1}(\theta)}] \lesssim \max_{z:z^\top \bar{\theta} \in I_j}\|U_j(z, \theta)\|^2_{\bar{\Lambda}_j^{-1}(\theta)} \overset{\lambda_{\min}(H\bar{\Lambda}_j(\theta)H) \gtrsim n_j^{-1/2}}{\lesssim} \sqrt{n_j}.$$

Then we may use Bernstein's inequality to obtain that with probability at least $1 - \mathcal{O}(\delta)$,

$$\mathcal{R}_{1121} \lesssim \sqrt{\log(1/\delta)} n_j^{-1/4} \|U_j(x,\theta)\|_{\bar{\Lambda}_j^{-1}(\theta)} + \log(1/\delta) n_j^{-1/2} \|U_j(x,\theta)\|_{\bar{\Lambda}_j^{-1}(\theta)}.$$

For $\mathcal{R}_{1122}$, note that

$$|\mathcal{R}_{1122}| \leq \|U_j(x,\theta)\|_{\Lambda_j^{-1}(\theta)} \cdot \underbrace{\|(H\Lambda_j(\theta)H)^{-1/2}(H\Lambda_j(\theta)H - n_j^{\text{ra}} H\bar{\Lambda}_j(\theta)H)(n_j^{\text{ra}} H\bar{\Lambda}_j(\theta)H)^{-1/2}\|}_{\text{with probability at least } 1 - \mathcal{O}(\delta), \quad \leq \sqrt{\log(1/\delta)}}$$

$$\times \left\| (n_j^{\text{ra}} H\bar{\Lambda}_j(\theta)H)^{-1/2} \sum_{t \in \mathcal{T}_j^{\text{ra}}} (Z_t - \mathbb{E}_{x_t} Z_t) \right\|.$$

It can be easily bound that $|(n_j^{\text{ra}} H\bar{\Lambda}_j(\theta)H)^{-1/2} Z_t| \lesssim 1$ and by the same reason as in (i) of (13),

$$\mathbb{E}[\|(n_j^{\text{ra}} \bar{\Lambda}_j(\theta))^{-1/2} Z_t\|_2^2]$$

$$\lesssim \frac{1}{n_j} \mathbb{E}[U_j^\top(z,\theta) \bar{\Lambda}_j^{-1}(\theta) U_j(x_t,\theta) U_j(x_t,\theta)^\top \bar{\Lambda}_j^{-1}(\theta) U_j(x_t,\theta) U_j(x_t,\theta)^\top \bar{\Lambda}_j^{-1}(\theta) U_j(z,\theta)]$$

$$= \frac{1}{n_j} \mathbb{E}[\text{tr}\left( U_j(x_t,\theta) U_j(x_t,\theta)^\top \bar{\Lambda}_j^{-1}(\theta) U_j(x_t,\theta) U_j(x_t,\theta)^\top \bar{\Lambda}_j^{-1}(\theta) \right)]$$

$$\lesssim \frac{1}{n_j} \mathbb{E}[\|U_j(x_t,\theta)\|_{\bar{\Lambda}_j^{-1}(\theta)}^4] \lesssim n_j^{-1/2}.$$

Then we may use Bernstein's inequality to obtain that with probability at least $1 - \mathcal{O}(\delta)$,

$$\mathcal{R}_{1122} \lesssim (n_j^{1/4} \log(1/\delta) + \log^{3/2}(1/\delta)) \cdot \|U_j(x,\theta)\|_{\Lambda_j^{-1}(\theta)}.$$

Combining all of the above estimates, for any unit vector $w$, we have with probability at least $1 - \mathcal{O}(\delta)$,

$$|\mathcal{R}_1 w| \lesssim \log^{3/2}(1/\delta) \cdot (n_j^{-1/4} \|U_j(x,\theta)\|_{\bar{\Lambda}_j^{-1}(\theta)} + n_j^{1/4} \|U_j(x,\theta)\|_{\Lambda_j^{-1}(\theta)}).$$

Standard $\varepsilon$-net argument then leads to, with probability at least $1 - \mathcal{O}(\delta)$,

$$\|\mathcal{R}_1\|_2 \lesssim d^{3/2} \log^{3/2}(1/\delta) \cdot (n_j^{-1/4} \|U_j(x,\theta)\|_{\bar{\Lambda}_j^{-1}(\theta)} + n_j^{1/4} \|U_j(x,\theta)\|_{\Lambda_j^{-1}(\theta)}).$$

Therefore, with probability at least $1 - \mathcal{O}(\delta)$,

$$\|\bar{\boldsymbol{v}}_j(x,\theta) - \boldsymbol{v}_j(x,\theta)\|_2 \lesssim d^{3/2} \log^{3/2}(1/\delta) \cdot (n_j^{-1/4} \|U_j(x,\theta)\|_{\bar{\Lambda}_j^{-1}(\theta)} + n_j^{1/4} \|U_j(x,\theta)\|_{\Lambda_j^{-1}(\theta)}).$$

The claim in (i) follows by further taking union bounds on $x$ and $\theta$.

(ii). With the bound in (i), we have with probability at least $1 - \mathcal{O}(\delta)$,

$$\boldsymbol{v}_j(x,\theta)^\top (\Sigma_j(\theta) + \zeta I)^{-1} \boldsymbol{v}_j(x,\theta) = \|\boldsymbol{v}_j(x,\theta) - \bar{\boldsymbol{v}}_j(x,\theta)\|_{(\Sigma_j(\theta)+\zeta I)^{-1}}^2$$

$$+ \bar{\boldsymbol{v}}_j(x,\theta)^\top (\Sigma_j(\theta) + \zeta I)^{-1} \bar{\boldsymbol{v}}_j(x,\theta) + \left(\boldsymbol{v}_j(x,\theta) - \bar{\boldsymbol{v}}_j(x,\theta)\right)^\top (\Sigma_j(\theta) + \zeta I)^{-1} \bar{\boldsymbol{v}}_j(x,\theta)$$

$$\overset{(i)}{\lesssim} \|\boldsymbol{v}_j(x,\theta) - \bar{\boldsymbol{v}}_j(x,\theta)\|_{(\Sigma_j(\theta)+\zeta I)^{-1}}^2 + \bar{\boldsymbol{v}}_j(x,\theta)^\top (\Sigma_j(\theta) + \zeta I)^{-1} \bar{\boldsymbol{v}}_j(x,\theta)$$

$$\lesssim d^7 \log^3(1/\delta) \cdot (n_j^{-1/2} \zeta^{-1} \|U_j(x,\theta)\|_{\bar{\Lambda}_j^{-1}(\theta)}^2 + n_j^{1/2} \zeta^{-1} \|U_j(x,\theta)\|_{\Lambda_j^{-1}(\theta)}^2)$$

$$+ \bar{\boldsymbol{v}}_j(x,\theta)^\top (\Sigma_j(\theta) + \zeta I)^{-1} \bar{\boldsymbol{v}}_j(x,\theta)$$

where in (i) we have used $ab \lesssim a^2 + b^2$. It remains to replace $\Sigma_j(\theta) + \zeta I$ by $\bar{\Sigma}_j(\theta) + \zeta I$. Note by the bound in (i), it holds with probability at least $1 - \mathcal{O}(\delta)$ that

$$\left\| \sum_{t \in \mathcal{T}_j} (\boldsymbol{v}_j(x_t,\theta) - \bar{\boldsymbol{v}}_j(x_t,\theta))(\boldsymbol{v}_j(x_t,\theta) - \bar{\boldsymbol{v}}_j(x_t,\theta))^\top \right\|_2$$

$$\lesssim d^7 \log^3(1/\delta) \cdot \sum_{t \in \mathcal{T}_j} (n_j^{-1/2} \|U_j(x_t,\theta)\|_{\bar{\Lambda}_j^{-1}(\theta)}^2 + n_j^{1/2} \|U_j(x_t,\theta)\|_{\Lambda_j^{-1}(\theta)}^2)$$

$$\lesssim d^7 \log^3(1/\delta) \cdot \left( \sum_{t \in \mathcal{T}_j} n_j^{-1/2} \|U_j(x_t, \theta)\|^2_{\bar{\Lambda}_j^{-1}(\theta)} + n_j^{1/2} \right)$$

$$= d^7 \log^3(1/\delta) \cdot \left( n_j^{-1/2} \sum_{t \in \mathcal{T}_j} (1 - \mathbb{E}_{x_t}) \|U_j(x_t, \theta)\|^2_{\bar{\Lambda}_j^{-1}(\theta)} + n_j^{-1/2} \sum_{t \in \mathcal{T}_j} \mathbb{E}_{x_t}[\|U_j(x_t, \theta)\|^2_{\bar{\Lambda}_j^{-1}(\theta)}] + n_j^{1/2} \right)$$

$$\lesssim d^7 \log^3(1/\delta) \cdot \left( n_j^{-1/2} \sum_{t \in \mathcal{T}_j} (1 - \mathbb{E}_{x_t}) \|U_j(x_t, \theta)\|^2_{\bar{\Lambda}_j^{-1}(\theta)} + n_j^{1/2} \right).$$

As $\max_{z:z^\top \bar{\theta} \in I_j} \|U_j(z, \theta)\|^2_{\bar{\Lambda}_j^{-1}(\theta)} \lesssim \sqrt{n_j}$ and $\mathbb{E}_{x_t}[\|U_j(x_t, \theta)\|^2_{\bar{\Lambda}_j^{-1}(\theta)}] \lesssim 1$, we may apply Hoeffding's inequality to obtain that with probability at least $1 - \mathcal{O}(\delta)$,

$$\left\| \sum_{t \in \mathcal{T}_j} (\boldsymbol{v}_j(x_t, \theta) - \bar{\boldsymbol{v}}_j(x_t, \theta))(\boldsymbol{v}_j(x_t, \theta) - \bar{\boldsymbol{v}}_j(x_t, \theta))^\top \right\| \lesssim d^7 \log^{7/2}(1/\delta) \cdot n_j^{1/2}.$$

Then using the fact that $bb^\top \preceq 2aa^\top + 2(a - b)(a - b)^\top$ holds for any vectors $a, b$, we have

$$\bar{\Sigma}_j(\theta) + \zeta I = \sum_{t \in \mathcal{T}_j} \bar{\boldsymbol{v}}_j(x_t, \theta) \bar{\boldsymbol{v}}_j(x_t, \theta)^\top + \zeta I$$

$$\preceq 2 \sum_{t \in \mathcal{T}_j} \boldsymbol{v}_j(x_t, \theta) \boldsymbol{v}_j(x_t, \theta)^\top + 2 \sum_{t \in \mathcal{T}_j} (\boldsymbol{v}_j(x_t, \theta) - \bar{\boldsymbol{v}}_j(x_t, \theta))(\boldsymbol{v}_j(x_t, \theta) - \bar{\boldsymbol{v}}_j(x_t, \theta))^\top + \zeta I$$

$$\preceq 2 \sum_{t \in \mathcal{T}_j} \boldsymbol{v}_j(x_t, \theta) \boldsymbol{v}_j(x_t, \theta)^\top + \zeta I + \mathcal{O}\left( d^7 \log^{7/2}(1/\delta) \cdot \sqrt{n_j} \right) I.$$

By the choice of $\zeta$, we arrive at

$$\bar{\Sigma}_j(\theta) + \zeta I \preceq 2(\Sigma_j(\theta) + \zeta I).$$

This concludes the proof. □

## J  PROOF OF REMARK 7

In this section, we provide a detailed algorithm design and regret guarantee with the first term of right-hand-side in Proposition 6 is omitted. Throughout the analysis, we only use $\widehat{g}_t(\cdot \mid \bar{\theta}_0)$, thus we simplify the notation via

$$U_j(x) := U_j(x, \bar{\theta}_0), \quad \Lambda_j := \Lambda_j(\bar{\theta}_0).$$

Moreover for the quantities (e.g. $\Lambda_j, \mathcal{T}_j$) defined in Algorithm 2 when it is called at $t$-th step, we use the notation $\Lambda_{t,j}, \mathcal{T}_{t,j}$ to denote them..

**Initial Exploration.**  In the line 3-7, we first computing the rounded prices for every fine intervals $\{I_j\}_{j \in [M]}$, as we discussed in Section 6 and rigorously proved in Appendix G, we have this ensures that when computing $\Lambda_{t,j}$ over each $j \in [M]$ invertible and has the eigenvalue lower bound $\Omega(1/T)$ for all subsequent $t$. The total regret incurred in this phase is bounded by the total exploration steps, which is given by

$$\mathcal{O}(\sqrt{Th}/h) = \mathcal{O}(\sqrt{T/h}) = \mathcal{O}(T^{\frac{\beta+1}{2\beta+1}}),$$

thus it suffices to bound the regret incurred over line 8 to 13.

**UCB Phase.**  In the UCB phase, we first compute a confidence bound on $\widehat{g}_t$ based on Proposition 6: when the first term is omitted, we have the output $\widehat{g}_{t,j}(x \mid \bar{\theta}_0)$ satisfies

$$\widehat{g}_{t,j}(x \mid \bar{\theta}_0) - g(x^\top \theta_0) = \underbrace{U_j(x)^\top \sum_{s \in \mathcal{T}_{t,j}} \varepsilon_s \Lambda_{t,j}^{-1} U_j(x_s)}_{:=\mathcal{A}_j} + \mathcal{O}\left( T^{-\frac{\beta}{2\beta+1}} (1 + \sqrt{t_j} \|U_j(x)\|_{\Lambda_{t,j}^{-1}}) \right).$$

for $\varepsilon_s := y_s - g(x_s^\top \theta_0)$ and $t_j = |\mathcal{T}_{t,j}|$. Now for $\mathcal{A}_j$, we have the following self-normalized martingale concentration result:

---

**Algorithm 5** Piloted UCB Algorithm with Local Polynomial Regression

---

1: **Inputs:** pilot estimator $\bar{\theta}_0$ with $\|\bar{\theta}_0 - \theta_0\| \leq \eta$; smoothness $\beta \geq 1$, hyper-parameter $\alpha > 0$.
2: **Initialization:** fix the polynomial degree level $\ell = \lfloor \beta \rfloor, \mathcal{D} = \emptyset$, set $h = T^{-\frac{1}{2\beta+1}}$ partition $[-V, V]$ into $M = \lceil 1/h \rceil$ intervals $\{I_j\}_{j \in M}, t = 1$.
3: **for** $j = 1, \ldots, M$ **do**
4:     **for** $L = 1, \ldots, \lceil \sqrt{Th} \rceil$ **do**
5:         After observing $c_t$, selecting a price $p_t$ so that $x_t^\top \bar{\theta}_0$ is the $(L \bmod \ell)$-th $\ell$-equi-partition point of $I_j$. //Forced Exploration for every $I_j$.
6:         Add $x_t$ and the feedback $y_t$ to $\mathcal{D}$.
7:         $t \leftarrow t + 1$.
8: **while** $t < T$ **do**
9:     Compute $\widehat{g}_{t,j}(\cdot \mid \bar{\theta}_0)$ for $j \in [M]$ via Algorithm 2 with input $\mathcal{D}$ and precision $h = T^{-\frac{1}{2\beta+1}}$

$$\textbf{Glued Estimator:} \widehat{g}_t(x) := \sum_{j \in [M]} \mathbf{1}\{x^\top \bar{\theta}_0 \in I_j\} \widehat{g}_j(x \mid \bar{\theta}_0)$$

$$\textbf{Glued Confidence Bound:} \mathrm{CB}_t(x) := \sum_{j \in [M]} \mathbf{1}\{x^\top \bar{\theta}_0 \in I_j\} \mathrm{CB}_{t,j}(x),$$

    with $\mathrm{CB}_{t,j}$ defined as in (14).
10:     Computing $\widehat{g}_t^{\mathrm{UCB}}(x) := \widehat{g}_t(x) + \alpha \mathrm{CB}_t(x)$ pull the UCB price $p$ so that for observed $c_t$, and $z_t(p) := (c_t^\top, p)^\top$,

$$p_t = \mathrm{argmax}_{p \in [0, p_{\max}]} p \widehat{g}_t^{\mathrm{UCB}}(z_t(p)^\top \bar{\theta}_0).$$

11:     Observe the feedback $y_t$ and add $(z_t(p_t), y_t)$ to $\mathcal{D}$.

---

**Lemma 21** (Theorem 1 and 2 of Abbasi-Yadkori et al. (2011)). *For any $\delta > 0$, with probability at least $1 - \delta$, it holds that*

$$\|\sum_{s \in \mathcal{T}_{t,j}} \varepsilon_s \Lambda_{t,j}^{-1/2} U_j(x_s)\| \lesssim \sqrt{\log(T/\delta)}, \quad \forall t \in [T].$$

Thus we have with probability at least $1 - \delta$,

$$|\mathcal{A}_j| \lesssim \|U_j(x)\|_{\Lambda_{t,j}^{-1}} \sqrt{\log(TM/\delta)}, \quad \forall j \in [M].$$

And by $M = T^{-\frac{1}{2\beta+1}}$, we can give the confidence bound as

$$\mathrm{CB}_t(x) := \sum_{j \in [M]} \mathbf{1}\{x^\top \bar{\theta}_0 \in I_j\} \left( \|U_j(x)\|_{\Lambda_{t,j}^{-1}} \sqrt{\log(T)} + T^{-\frac{\beta}{2\beta+1}} \right), \quad (14)$$

it then holds that with probability at least $1 - \mathcal{O}(1/T)$,

$$|\widehat{g}_{t,j}(x) - g(x^\top \theta_0)| \leq \alpha \mathrm{CB}_t(x)$$

uniformly for all $t$ at UCB phase and $x$ for some large enough $\alpha$ depending only on $\beta$.

As a result, we have with probability at least $1 - \mathcal{O}(1/T)$, for $x_t := (c_t^\top, p_t)^\top$

$$\sum_t R(c_t, p^\star(c_t^\top \theta_*)) - R(c_t, p_t) \leq \sum_t \left( p_t \widehat{g}_t^{\mathrm{UCB}}(x_t) - p_t g(x_t^\top \theta_0) \right) \leq \alpha p_{\max} \sum_t \mathrm{CB}_t(x_t)$$

$$\leq \alpha p_{\max} \sum_t \sum_{j \in [M]} \mathbf{1}\{x^\top \bar{\theta}_0 \in I_j\} \left( \|U_j(x)\|_{\Lambda_{t,j}^{-1}} \sqrt{\log(T)} + T^{-\frac{\beta}{2\beta+1}} \right)$$

$$\leq \alpha p_{\max} \sqrt{\log T} \left( T^{\frac{\beta+1}{2\beta+1}} + \sum_{j \in [M]} \sum_{t \in \mathcal{T}_{T,j}} \|U_j(x)\|_{\Lambda_{t,j}^{-1}} \right)$$

$$\leq_{\text{(i)}} \alpha p_{\max} \sqrt{\log T} \left( T^{\frac{\beta+1}{2\beta+1}} + \log(T) \sum_{j \in [M]} \sqrt{T_j} \right) \leq_{\text{(ii)}} \alpha p_{\max} T^{\frac{\beta+1}{2\beta+1}} \log^{3/2} T.$$

Where in (i) we have used the elliptic potential lemma (see e.g. Lemma 11 of Abbasi-Yadkori et al. (2011)) and in (ii) we have used

$$\sum_j \sqrt{T_j} \leq \sqrt{TM} \leq T^{\frac{\beta+1}{2\beta+1}},$$

as desired.

**Remark 22** (Adversarial Context Setting and Adaptive Exploration.). *We note that throughout our algorithm and analysis, the only component that requires a stochastic context assumption is the initial construction of a pilot estimator with error $\mathcal{O}(\eta)$, which is needed to satisfy the conditions of Proposition 6. On the other hand, in $\beta = 1$ setting, Tullii et al. (2024) propose an adaptive exploration procedure for estimating $\bar{\theta}_0$ that works even under adversarial contexts. We believe their approach could also be incorporated here. However, since this discussion is intended solely to illustrate the difficulty created by the first term in Proposition 6—a term we deliberately omit in our analysis—we keep our algorithmic design simple and aligned with the main paper's structure for clarity.*

## K  NUMERICAL EXPERIMENTS

### K.1  IMPLEMENTATION DETAILS OF ALGORITHM

In this section, we discuss several details in implementing Algorithm 3. Note that in the description of the algorithm, we frequently use quantities $\{\widehat{g}_j(\cdot|\theta)\}_{\theta \in \Theta}$ and $\{\pi(\mathsf{c})\}_{\mathsf{c} \in \mathcal{C}}$ for continuous spaces $\Theta, \mathcal{C}$. However, from a computational perspective, maintaining these quantities would require keeping parameterized functions simultaneously for all possible values of $\theta$ or $\mathsf{c}$ over a continuous range, which requires a discretization over $\Theta, \mathcal{C}$, leads to computational inefficiency[6]. In the following, we provide details on how to efficiently bypass the operations that would seem to require maintaining these quantities.

### K.1.1  CONSTRAINED LEAST SQUARED SOLUTION IN EQUATION (2).

Note that the only procedure in the algorithm that requires querying $\widehat{g}(\cdot|\theta)$ for continuously varying $\theta$ is when solving (2). An important observation is that, given the data $\{(x_i, y_i)\}$ collected within each epoch, we can compute $\widehat{g}(x_i|\theta)$ for any $\theta$. This allows us to rewrite the objective function in (2) purely in terms of $\theta$, which can be evaluated directly using the collected data. As a result, solving (2) becomes feasible using standard black-box continuous constrained-optimization methods.

We also note that, as a function of $\theta$, the objective in (2) is generally non-convex even when $\beta = 2$ (as in the setting of Wang & Chen (2025) and in earlier statistical literature such as Härdle et al. (1993); Ichimura (1993); Horowitz & Härdle (1996)). Following the approach of Wang & Chen (2025), we apply a general interior-point method for this continuous optimization problem during the experiment. While this method is only guaranteed to return a local minimum, it already demonstrates good empirical performance in our implementation.

### K.1.2  SAMPLING OF THE CONTEXT-WISE PRICING POLICY INTERVAL.

Another subtle challenge in efficiently implementing Algorithm 3 is that the context-wise pricing policy interval must be queried at every round. Since this interval is defined separately for each context $c$ over a continuous space of dimension $d$, even an approximate tabulation would require storage that scales exponentially in $d$. In this section, we describe a "lazy update" approach that stores only the historical datasets and computes the pricing interval solely for those contexts $c_t$ that actually appear during online decision making. This procedure can be implemented efficiently by repeatedly calling the policy-improvement procedure (Algorithm 4) for $\mathcal{O}(\log T)$ iterations, thereby eliminating the exponential storage cost in $\exp(d)$.

---

[6]Note that both $\Theta, \mathcal{C}$ are $d$-dimensional, and find its minimum $\varepsilon$-covering requires $\Theta(\varepsilon^{-d})$ storage

Roughly speaking, we treat the policy-improvement procedure used at epoch $\ell$, denoted by $\mathcal{A}_\ell$, as an operator and store only the data required to evaluate $\mathcal{A}_\ell$ in future rounds. When a context $c$ arrives in epoch $\tau$, we *compose* the previously saved operators:

$$[\underline{\pi}^{(\tau-1)}(\mathsf{c}), \bar{\pi}^{(\tau-1)}(\mathsf{c})] \leftarrow (\mathcal{A}_{\tau-1} \circ \cdots \circ \mathcal{A}_1)(\mathsf{c}),$$

which exactly matches the interval that would have been obtained had we maintained and updated it epoch-by-epoch, yet without requiring any per-context storage.

**Data Required for Evaluating $\mathcal{A}_\ell$.** At the end of each epoch $\ell$, we store the dataset $\mathcal{V}_\ell$ that contains:

1. **The pilot estimator and per-bin constrained least-squares estimators:** $\bar{\theta}_0$ and $\{\widehat{\theta}_{\ell,j}\}_{j\in[M_\ell]}$.

2. **Bin-wise local polynomial design matrices under $\widehat{\theta}_{\ell,j}$:**
$$\left\{ \left(\Lambda_{\ell,j}(\widehat{\theta}_{\ell,j}), \sum_{t\in\mathcal{T}_{\ell,j}} y_t\, U_j(x_t, \widehat{\theta}_{\ell,j})\right)\right\}_{j\in[M_\ell]}.$$

This information is sufficient for computing each $\{\widehat{g}_{\ell,j}(x|\widehat{\theta}_{\ell,j})\}$ and the glued estimator $\widehat{g}_\ell(x)$ and its confidence bound $\mathrm{CB}_\ell(x)$ for any $x = (\mathsf{c}, p)$.

**Computing $\pi^{(\tau-1)}(\mathsf{c})$ for a given $\mathsf{c}$.** Suppose a context $c$ is observed at epoch $\tau$. We now describe how to compute $\pi^{(\tau-1)}(\mathsf{c})$ from the stored datasets $\cup_{\ell<\tau}\mathcal{V}_\ell$. For each step $\ell = 1, 2, \ldots, \tau - 1$, given the input context $c$ and the current policy interval $[\underline{\pi}^{(\ell-1)}(\mathsf{c}), \bar{\pi}^{(\ell-1)}(\mathsf{c})]$,[7] the algorithm evaluates the integrals appearing in $\widehat{r}(J_k)$ and $\Delta(J_k)$ for each sub-interval $J_k$ using numerical integration with discretization length $1/\varepsilon$ for $\varepsilon = 1/\sqrt{T}$.[8] This requires $\mathcal{O}(\sqrt{T})$ queries to $\mathrm{CB}_\ell(\cdot)$ and $\widehat{g}_\ell(\cdot)$, both of which are computable from $\mathcal{V}_\ell$.

## K.2 Experiment Setup

In this section, we present numerical simulations under several setups to illustrate the performance of our algorithm and to compare it with previous work Fan et al. (2024). In the following sections, we describe the setup and purpose of four different experiments, including:

1. Illustration of the effect of $\beta$ on regret: given an underlying smooth environment, whether using a larger $\beta$ parameter in the algorithm input leads to better regret.

2. Illustration of the effect of $d$ on regret, especially whether the $\mathrm{Poly}(d^\beta)$ term in the main theorem significantly influences the empirical results.

3. Comparison with the algorithms in Fan et al. (2024).

**$\beta$-smooth Tail Function Generation.** Before describing more details of setup in each setting, we first recall the noise sampling procedure proposed in Fan et al. (2024) for generating a $\beta_0$-smooth $g$, which we will frequently call in each setup: Given any smoothness factor $\beta_0$, we set the density function of $\xi_t$ as

$$f_\beta(z) \propto (1/4 - z^2)^{\beta/2} \cdot \mathbf{1}\{|z| \le 1/2\}. \tag{15}$$

It can be verified that $f_\beta(\cdot)$ is $(\beta - 1)$-smooth function, thus its corresponding CDF( and $g$) is $\beta$-smooth.

### K.2.1 Effect of $\beta$ on Regret

In Figure 4, we test our algorithm under a $d = 2$ environment with underlying smoothness $\beta_0 = 6$, with the underlying parameter $\theta_0 = (0.25, 0.25)$, coordinate-wise i.i.d. context distribution with the density function

$$f_m(x) \propto (2/3 - x^2)^{m+1} \cdot \mathbf{1}\left\{|x| \le \sqrt{2/3}\right\}. \tag{16}$$

---

[7]By initialization, $[\underline{\pi}^{(0)}, \bar{\pi}^{(0)}] = [0, p_{\max}]$.

[8]This contributes at most $\mathcal{O}(1/\sqrt{T})$ error to the calculation, which is dominated by the $\mathrm{CB}(\cdot)$ term.

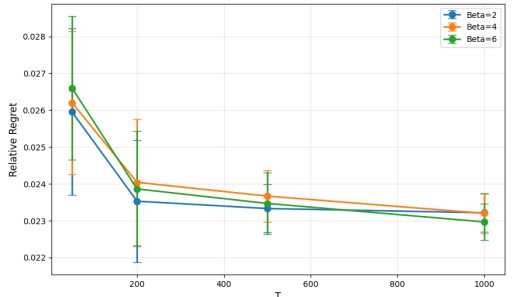 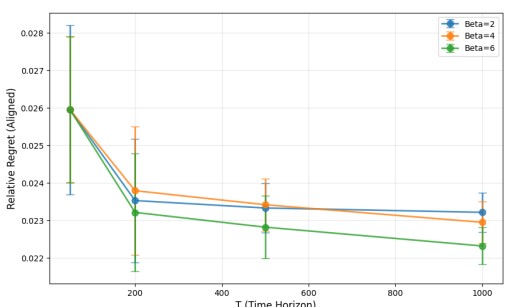

(a) Relative regret( the regret normalized by $T$) under $\beta_0 = 6, d = 2$ environment with algorithm parameters $\beta \in \{2, 4, 6\}, T \in \{50, 200, 500, 1000\}$.

(b) Same results as in Figure 4a with the starting point of each curve aligned to illustrate the regret decay rate.

Figure 4: Illustration of $\beta$ effect in regret.

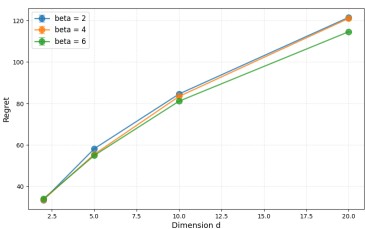

Figure 5: Regret under $\beta_0 = 6, T = 1000$ environment, with changing $d \in \{2, 5, 10, 20\}$ and algorithm parameters $\beta \in \{2, 4, 6\}$.

We test the LPSP algorithm under this environment with input smoothness parameters $\beta = 2, 4, 6$ and time horizons $T \in \{50, 200, 500, 1000\}$, and we report the relative regrets (regret divided by $T$) in Figure 1(a). To further compare the regret rates while reducing the influence of absolute constants, we additionally align the starting $y$-axis values in Figure 1(b).

From Figure 4a, we observe that larger $\beta$ values ($\beta = 4, 6$) do not necessarily lead to smaller regret compared with $\beta = 2$ when $T$ is relatively small, likely due to the $\beta$-dependent constants hidden in the regret bound. However, as $T$ increases, the performance of the larger-$\beta$ algorithms begins to match or outperform the $\beta = 2$ setting. Figure 4b provides more direct evidence of better long-run regret: after aligning the starting regrets for each $\beta$, so that only the decay rate matters, we see that larger $\beta$ generally leads to a sharper decay rate, consistent with our theoretical findings.

### K.2.2 EFFECT OF $d$ ON REGRET

In Figure 5, we report the regret of our algorithm for $\beta \in \{2, 4, 6\}$ with $T = 1000$ under different dimensions $d$. As in the previous setup, the demand noise is generated with smoothness $\beta_0 = 6$ using (15). The underlying parameter is chosen as $\theta = (1/\sqrt{d}, \ldots, 1/\sqrt{d}) \in \mathbb{R}^d$, and the context distribution follows (16) without additional normalization. Hence Assumption 4 is satisfied with $c_{\min} = d$, which implies an exploration length of order $\mathcal{O}\left(\sqrt{d}T^{\frac{\beta+1}{2\beta+1}}\right)$.

The figure illustrates that although the regret increases at least linearly in $d$, the choice of $\beta$ does not appear to affect the growth rate significantly. This suggests that the $\mathrm{Poly}(d^\beta)$ factor appearing in our second-order regret bound may be an artifact of the analysis rather than a fundamental barrier. The empirical trend also indicates the possibility of further improving the $d^4$ dependence in the leading-order term.

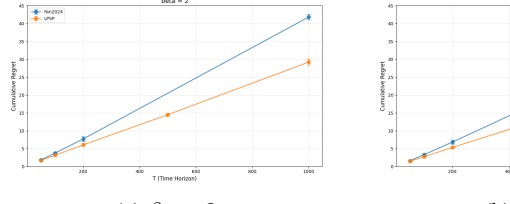 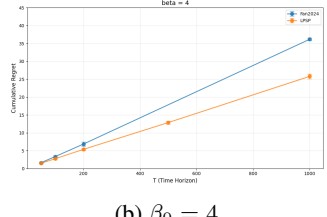 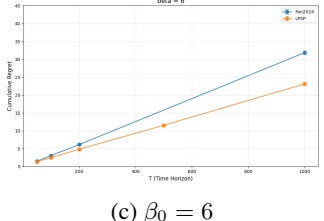

(a) $\beta_0 = 2$        (b) $\beta_0 = 4$        (c) $\beta_0 = 6$

Figure 6: Comparison with the explore-then-commit algorithm in Fan et al. (2024) under different smoothness parameters.

### K.2.3 COMPARISON TO FAN ET AL. (2024).

In Figure 6, we compare the cumulative regret for $T \in \{50, 200, 500, 1000\}$ of our algorithm with Fan et al. (2024) under different environments, with $d = 2$, $\theta_0$ and context distribution described same as in Section K.2.1, and noise distribution under different $\beta_0$ are generated as in (15).

While our algorithm achieves consistently smaller regret than Fan et al. (2024) in both experimental settings, we emphasize that this comparison is not fully fair. The primary message we aim to convey is simply that both algorithms are able to exploit the underlying smoothness: as the true smoothness parameter $\beta_0$ increases, the regret curves decrease accordingly.

The key subtlety lies in the computational scale of the two methods. The algorithm of Fan et al. (2024) is simple to implement and computationally lightweight, which enables them to run experiments with very large time horizons (e.g., up to $T \approx 12,000$ in their paper). In contrast, our method involves several computationally intensive steps—such as the constrained least-squares refinement and repeated distribution-shift corrections—as discussed in Section K.1. These components significantly increase runtime, which limits our experiments to relatively small horizons (up to $T = 1000$). This difference in feasible scale may disadvantage Fan et al. (2024) in our plots: in their original setup, the initial exploration length is fixed at $500$, whereas in our smaller-$T$ regime we can only afford an initial phase of roughly 20–100 rounds. Consequently, their algorithm may not reach its typical performance regime under the smaller horizons we are able to simulate. We also emphasize that the primary focus of this work is theoretical: our goal is to push the boundary of regret guarantees for semi-parametric pricing by showing that improved rates are achievable—albeit through a relatively complicate algorithm that may not yet be practical. Developing simpler, more efficient, and easy-to-implement algorithms that attain the same theoretical regret remains an important direction for future work.

