# OpenReview forum: "Semi-Parametric Contextual Pricing with General Smoothness"
_ICLR.cc/2026/Conference — ICLR 2026 Poster_

### Official Review · Reviewer_TBPD · 2025-10-25

**Soundness:** 3
**Presentation:** 2
**Contribution:** 2
**Rating:** 6
**Confidence:** 4

**Summary:**

The paper studies contextual dynamic pricing with binary purchase feedback under a semi-parametric single-index model: the expected demand takes the form $g\left(c_t^{\top} \theta-p_t\right)$, where $g$ is unknown and $\beta$ Hölder smooth, for some known $\beta \geq 1$. They design an algorithm based on local polynomial regression that achieves regret $\bar{O}\left(T^{(\beta+1) /(2 \beta+1)}\right)$ for all $\beta \geq 1$, matching previous works for $\beta=1,2$. Assumptions include bounded contexts, a strong unimodality/curvature condition around the revenue maximizer, and a context diversity condition only during the short exploration phase.

**Strengths:**

Overall, the paper's primary strength is that it achieves the semiparametric optimal regret rate uniformly for all $\beta \geq 1$, addressing a long-standing challenge since Fan et al. (2024). Two additional notable contributions are:

(1) the demonstration that the use of local polynomial regression effectively shortens the exploration phase, achieved without imposing a boundedness assumption on the covariance of the context, and

(2) the establishment of joint convergence of $(\theta, g)$, which I believe is of independent interest to the statistical literature.

**Weaknesses:**

The main limitation is that a similar result has already been achieved by [1]. Their approach is closely related, and their parameter $\mathrm{\beta}$ can also lie in $(0,1)$, not only on $[1,\infty)$. Additionally, $\beta$ might be unknown. In this latter case, they propose an algorithm that maintains the same lower bound $\Omega(T^{\frac{\beta+1}{2\beta+1}})$. It would strengthen the paper to clarify how your contribution differs from or improves upon this prior work. A good position on this would raise my score.

## References
[1] Ye, Zeqi, and Hansheng Jiang. "Smoothness-adaptive dynamic pricing with nonparametric demand learning." International Conference on Artificial Intelligence and Statistics. PMLR, 2024.

**Questions:**

1. I find the content of the paper really interesting, but it could be arranged better; for example, by finding some space for the related works and conclusions, and by retaining only the essential material in the main text. In particular, I think that Algorithm 4 in Appendix C is important enough to be at least mentioned or summarized in the main part, since it plays a role in the exploitation phase.

2. It would be helpful to include an explicit remark in the main text explaining how the boundedness assumption on the covariance of the context can be avoided. If I am not mistaken, the key point lies in Equation (12), which essentially shows that the ``transformed'' $U$-based covariance matrix is controlled (specifically, equal to $[\beta] + 1$).

3. I am not sure about Footnote 3. It would be clearer if you specified which sub-results hold under the adversarial case and which do not. For instance, Theorem 9 (and subsequent results) hold under the stochastic setting (i.i.d. and stationary covariates). The sentence ``we keep our discussion in the stochastic setting mainly for clarity'' makes the scope somewhat confusing.

4. The usage of the term "uniformly" is somewhat confusing. For example, in Proposition 9, the uniformity over all $x$ appears to relate to $\sup_x \|\hat{g} - g\|$, but the upper bound itself depends on $x$. This made me think that you might instead be referring to the intersection of events over all $x$ as in (3). If that is the case, perhaps the phrase "simultaneously for all $x$'' would be less ambiguous. The same clarification applies to other results, such as Lemma 16.

5. In Proposition 8, do you mean that the sequence $\{\xi_i\}$ (instead of $y_i$) is mutually independent given $x$?

6. In Theorem 9, you use the notation $\mathcal{T}_j^{ro}$, but I could not find where it was defined.

7. Line 147: $\theta_0$

---

> ### Author Response · Authors · 2025-11-22
> **Response to Reviewer TBPD**
>
> Thank you for your valuable feedback and suggestions! Please find our point-to-point response below:
>
> **1. Regarding the Overlap with [1] (W1).**
>
> Thank you for raising this question! We agree with the reviewer that both our work and [1] achieve the $\tilde{O}\left(T^{\frac{\beta+1}{2\beta+1}}\right)$ regret. However, **the most significant difference is that [1] considers the non-contextual setting, whereas we study a semi-parametric contextual setting.** In fact, as mentioned in our response to Reviewer wC3A (Q1) and dJDh (Q2) , [1] can be viewed as a $\beta$-adaptive follow-up to [2]. As discussed in Remark 7, our Proposition 6 can already unify the non-contextual $O\left(T^{(\beta+1)/(2\beta+1)}\right)$ result of [2] for general $\beta$. And all remaining results, including our main algorithmic design, focus on addressing the challenges arising from the contextual formulation.
>
>
> **2. Position of the Initialization Phase(Q1).**
>
> Thank you for this valuable suggestion! We have moved the initialization Algorithm to the main context in the revised version to improve clarity.
>
> **3. Remark on Lower Bound Condition(Q2).**
>
> Thank you for the suggestion! Yes, your understanding is right! We further clarify the point here for completeness. First, the previous work [3] already removes the covariance lower-bound assumption in the case $\beta = 2$ by adopting a **linear time forced sampling** procedure (corresponding to our Figure 2). In our setting, we can similarly remove this lower-bound condition because we extend their design to general $\beta$. However, an important modification in our analysis is that we show that only **sub-linear time forced sampling** is sufficient to eliminate the covariance lower-bound requirement. The key to achieving this improvement is to apply self-normalizing arguments such as (12) in several key steps, as pointed out by the reviewer. We have added a remark in the revised version to help readers better understand this technical contribution.
>
> **4. Footnote on adversarial context(Q3).**
>
> Thank you for raising this point! We apologize for the confusion. To clarify, the only result that may potentially extend to the adversarial setting is the discussion in Appendix J. We have added an additional paragraph in Appendix J for clarity and have revised the corresponding footnote to avoid further confusion.
>
>
> **5. Meaning of Uniformity (Q4).**
>
> Thank you for raising this point! By `uniformly holds', we mean the event
>     $$\left|\widehat{g}_j\left(x \lvert \widehat{\theta}_j\right)-g\left(x^{\top} \theta_0\right)\right| \lesssim \operatorname{Err}_j(x)+n^{-\frac{\beta}{2 \beta+1}},$$
>     holds **simultaneously** for all $x$. A counterpart of this statement is ``for any $x$, with probability $1-\delta$'',
>     $$\left|\widehat{g}_j\left(x \lvert \widehat{\theta}_j\right)-g\left(x^{\top} \theta_0\right)\right| \lesssim \operatorname{Err}_j(x)+n^{-\frac{\beta}{2 \beta+1}},$$
>     under which we needs to use additional union bound argument over $x$. We have revised our presentation in Proposition 8 for clarity.
>
> **6. Mutually independence under $x$ (Q5).**
>
> Thank you for raising this point! We would like to clarify that there is a subtle but important distinction between the mutual independence of $y$ and the mutual independence of $\xi$ conditional on $x$. Since the randomness of $y$ given $x$ depends not only on $\xi$ but also on the sampled policy $p$, mutual independence of $y$ effectively requires independence with respect to both $\xi$ and $p$. This condition is ensured by our epoch-based design: within each epoch, observing $x$ does not trigger any policy update, so the policy $p$ remains fixed throughout the epoch. As a result, the desired conditional independence holds.
>
>
> **7. Definition of $\mathcal{T}_j^{\text{ro}}$ (Q6).**
>
>  Thank you for raising this point! Actually, by writing $\mathcal{T}_j = \mathcal{T}^{\text{ra}}_j \cup \mathcal{T}^{\text{ro}}_j $ in Theorem~9, we mean $\mathcal{T}_j$ allows **any such** decomposition with $\mathcal{T}^{\text{ra}}_j, \mathcal{T}^{\text{ro}}_j$ satisfying the statements followed in Theorem 9, and this then automatically becomes the definition of $\mathcal{T}^{\text{ra}}_j, \mathcal{T}^{\text{ro}}_j$. We have updated the description in this part in the revised version to improve clarity.
>
>
> **8. $\theta$ typo (Q7).**
>
> Thank you so much for your careful reading and pointing out this typo! We have corrected it in the revised version.
>
>
> **References**
>
> [1] Ye Z, Jiang H. Smoothness-adaptive dynamic pricing with nonparametric demand learning. InInternational Conference on Artificial Intelligence and Statistics 2024 Apr 18 (pp. 1675-1683). PMLR.
>
> [2] Wang, Yining, Boxiao Chen, and David Simchi-Levi. "Multimodal dynamic pricing." Management Science 67.10 (2021): 6136-6152.
>
> [3] Wang, Yining, and Boxiao Chen. "Tight Regret Bounds in Contextual Pricing with Semi-parametric Demand Learning." Available at SSRN 5133677 (2025).

---

> > ### Comment · Reviewer_TBPD · 2025-11-26
> >
> > I thank the authors for their detailed responses and have raised my score, as all my concerns have been addressed, and I now find the paper’s contribution more than sufficient.

---

### Official Review · Reviewer_dJDh · 2025-10-27

**Soundness:** 2
**Presentation:** 2
**Contribution:** 3
**Rating:** 6
**Confidence:** 4

**Summary:**

This paper studies the contextual dynamic pricing problem under a semi-parametric reward model. The authors propose a unified algorithm that achieves a regret bound of  $\tilde{\mathcal{O}} (T^{(\beta+1)/(2\beta+1)})$ for general smoothness levels $\beta \ge 1$. The general setting with arbitrary $\beta \ge 1$ has previously been explored only by Fan et al. (2024), whose method attains a much slower rate of  $\tilde{\mathcal{O}} (T^{(2\beta+1)/(4\beta-1)} )$. The proposed rate therefore improves upon prior work and, in particular, matches the minimax-optimal rates for the special cases $\beta = 1$ and $\beta = 2$ established in the literature.

The paper’s main theoretical contribution lies in an improved analysis that bypasses the dependence issue in the joint estimation procedure and relaxes the eigenvalue lower bound condition, thereby significantly shortening the required exploration period.

**Strengths:**

This paper makes a strong theoretical contribution to the study of semi-parametric dynamic pricing, extending sharp regret guarantees to general smoothness levels $\beta \ge 1$.
The theoretical analysis is ambitious and technically nontrivial, especially the part dealing with the dependence structure and the weaker eigenvalue lower bound condition.

**Weaknesses:**

1. The paper includes no numerical experiments, either simulations or real data. Adding such experiments and comparing the results with existing methods would help demonstrate the practical advantages of the proposed algorithm, especially for smoother settings with $\beta > 2$.

2. The authors emphasize in the Contributions section that they address a key challenge: the dependence caused by reusing the same samples $\mathcal{T}_j$ to compute both $\widehat{\theta}_j$ and $\widehat{g}_j(\cdot \mid \cdot)$. However, the main text does not clearly explain how this dependence is theoretically handled.

3. Section 5 is hard to follow. For instance, Proposition 11 is not sufficiently motivated or explained and only becomes understandable after reading Section 6.

**Questions:**

1. What is the performance of the proposed method under adversarial settings as studied in recent works such as Fan et al. (2024), Wang et al. (2021) and Tullii et al. (2024).

2. The theoretical results assume that the smoothness parameter $\beta$ is known.  Can the algorithm be adapted automatically to unknown $\beta$?

3. Could the authors provide simulation results or comparisons with existing algorithms to clarify the empirical performance of the proposed method?

4. Please clarify how the proposed analysis bypasses the dependence issue between $\widehat{\theta}_j$ and $\widehat{g}_j(\cdot \mid \cdot)$.


**Minor Comments**

1. Table 1: Wang et al. (2021)  $\to$  Wang \& Chen (2025).

2. Line 102: $\tilde{\mathcal{O}}(T^{1/3}) \to \tilde{\mathcal{O}}(T^{2/3})$.

3. Line 147: $theta0 \to \theta_0$.

4. Algorithm 2, line 9: $p_t \sim \pi^{(0)}(c_t) \to p_t \sim \pi^{(\tau-1)}(c_t)$.

---

> ### Author Response · Authors · 2025-11-22
> **Response to Reviewer dJDh (1/2)**
>
> Thank you for your valuable feedback and suggestions! Please find our point-to-point response below:
>
> **1. Adversarial Setting (Q1).**
>
> Thank you for your insightful question. First, our main result essentially requires the stochastic context assumption due to both the pilot estimation phase in [1] and the stationary subroutine in [2]; this stochastic context requirement is also essential in [1,2]. Despite this limitation, Proposition 6, which establishes the estimation error for local polynomial regression, applies to an arbitrary context sequence and therefore has the potential to imply certain results in adversarial-context settings. This is particularly related to the works [3] for the non-contextual setting and [4] for the Lipschitz setting mentioned by the reviewer, as noted in Remark 7 and discussed in detail in Appendix J. However, we emphasize that leveraging this idea would lead to another algorithm that is independent of our main algorithm(the LPSP algorithm); the discussion is included only to illustrate the flexibility of Proposition 6 and its theoretical connections to prior work.
>
> **2. Adaptivity to $\beta$ (Q2).**
>
> Thank you for your insightful question! We believe that, as shown in previous works [5,6], an algorithm that adapts to $\beta$ can be designed under the *self-similarity condition* imposed in these works. In particular, as we responded to Reviewer 1, while such a generalization is possible, both the design and analysis under our setting still require substantial effort, and we believe this would merit an independent paper, much like the progression from the non-adaptive method in [2] to the adaptive version in [5] in the non-contextual setting.
>
> **3. Numerical Simulations (Q3).**
>
> Thank you for your valuable suggestion! We have included a new experiment section discussing the detailed implementation and numerical results of our algorithms. We would briefly include some results here related W1 and Q3:
>
> - **Regarding the effect of $\beta$.** We have used the construction in [1] for exact $\beta_0= 6$ order smooth environment, and test our algorithm with $\beta$ levels $2,4,6.$ It can be found that, while initially larger $\beta$ incurs larger error due to requirement in larger burn-in samples and $\beta$ dependent constant, then we can observe larger $\beta$ leads to faster drop on the relative regret(normalized by $T$) as $T$ increases, indicates larger $\beta$ enough smaller long-run regret.
>
> - **Regarding the comparison to other approaches.** Following the setup of [1], we have compared our algorithm and their algorithm under the environment constructed with underlying smoothness $\beta = 2,4$. While it shown that our result is better, we would acknowledge that it is really hard to make a total fair comparison. The key subtlety lies in the computational scale of the two methods. The algorithm of [1] is simple to implement and computationally lightweight, which enables them to run experiments with very large time horizons (e.g., up to T ≈12,000 in their paper). In contrast, our method involves several computationally intensive steps—such as the constrained least-squares refinement and repeated distribution-shift corrections—as discussed in Section L.1. These components significantly increase runtime, which limits our experiments to relatively small horizons (up to T = 1000). This difference in feasible scale may disadvantage [1] in our plots: for example, in [1]'s original setup, the initial exploration length is fixed at 500, whereas in our smaller-T regime we can only afford an initial phase of roughly 20–100 rounds. Consequently, their algorithm may not reach its typical performance regime under the smaller horizons we are able to simulate.  As our work mainly focus on theoretical bounds, we would leave the development of simpler, more efficient, and easy-to-implement algorithms that attain the same theoretical regret remains an important direction for future work.
>
> **References**
>
> [1] Fan, Jianqing, Yongyi Guo, and Mengxin Yu. "Policy optimization using semiparametric models for dynamic pricing." Journal of the American Statistical Association 119.545 (2024): 552-564.
>
> [2] Wang, Yining, and Boxiao Chen. "Tight Regret Bounds in Contextual Pricing with Semi-parametric Demand Learning." Available at SSRN 5133677 (2025).
>
> [3] Wang, Yining, Boxiao Chen, and David Simchi-Levi. "Multimodal dynamic pricing." Management Science 67.10 (2021): 6136-6152.
>
> [4] Tullii, Matilde, et al. "Improved algorithms for contextual dynamic pricing." Advances in Neural Information Processing Systems 37 (2024): 126088-126117.
>
> [5] Ye Z, Jiang H. Smoothness-adaptive dynamic pricing with nonparametric demand learning. InInternational Conference on Artificial Intelligence and Statistics 2024 Apr 18 (pp. 1675-1683). PMLR.
>
> [6] Gong X, Zhang J. Parameter-Adaptive Dynamic Pricing. arXiv preprint arXiv:2503.00929. 2025 Mar 2.

---

> ### Author Response · Authors · 2025-11-22
> **Response to Reviewer dJDh (2/2)**
>
> **3. Clarifying Dependency Issue(Q4\& W2).**
>
> Thank you for your suggestion!  At a high level, both our analysis and Wang's analysis require taking a union bound over $\theta$; that is, after fixing a $\theta$, we establish concentration for $\hat{g}(\cdot \lvert \theta)$ separately, and then take a union bound over an $\epsilon$-net discretization. However, even for a fixed $\theta$, analyzing the least-squares estimator remains subtle (we apologize for didn't mention this point in previous discussion below Proposition 8, we have now added it to the discussion in the revised version), since we must deal with the summation $\sum_i \varepsilon_i \hat{g}(x_i \lvert \theta)$, where each $\hat{g}(x_i \lvert \theta)$ depends simultaneously on all $\varepsilon_i$. Wang attempted to handle this dependency by defining an augmented leave-one-out estimator for $\hat{g}$, but as we discuss in Appendix K, this dependency issue still persists. In our approach, we observe that, due to the local polynomial regression form, $\hat{g}$, the above summation becomes a quadratic form of $\mathbf{\varepsilon}$, whose concentration can be handled using the standard Hanson-Wright inequality. We have now added a discussion below Proposition 8 in the revised version for clarity.
>
> **4. Other writing problems**
>
> We also thank the reviewer for the careful reading and for pointing out these typos. We have updated them accordingly in the revised version.

---

> > ### Comment · Reviewer_dJDh · 2025-11-26
> >
> > I appreciate the authors’ detailed response. My questions have been fully addressed. I have also read the rebuttals to the other reviewers’ comments. Since I had already provided a positive score, I will keep it unchanged.

---

### Official Review · Reviewer_tEP9 · 2025-10-31

**Soundness:** 2
**Presentation:** 3
**Contribution:** 2
**Rating:** 4
**Confidence:** 3

**Summary:**

This paper investigates the dynamic pricing problem, where a seller determines the price based on users’ contextual features (such as user information, time, and region) and learns the optimal pricing strategy from the observed purchasing behavior. The authors propose a semi-parametric model that combines a linear component with an unknown smooth function component to describe the variation of user demand with respect to price. They further develop a pricing algorithm that achieves a unified optimal regret bound across different smoothness levels β. Specifically, the study provides a unified treatment of contextual pricing problems under varying smoothness conditions β, overcoming the limitations of prior works restricted to β=1 and β=2. In algorithmic design, the method refines parameter estimation through local polynomial regression and constrained least squares, thereby improving convergence rates. The paper also reexamines the work of Wang & Chen (2025), identifies gaps in their proof, and provides stricter dependence control and theoretical analysis. Overall, the work demonstrates clear novelty in both theoretical depth and algorithmic design. By establishing a unified analytical framework across different smoothness levels β, it fills the theoretical gap in prior studies that only addressed specific values of β without a general analysis.

**Strengths:**

Relevance: The paper focuses on a contextual dynamic pricing problem of significant research value. By introducing semi-parametric estimation into dynamic pricing, it contributes to balancing model interpretability and flexibility.

Theoretical Work: The paper presents a unified upper bound on regret that continuously covers the entire range from β=1 to infinity. The theoretical analysis is rigorous, and the derivations are logically structured and well-organized.

**Weaknesses:**

Assumptions: The practical validity of assumptions is limited. (a) The strong uni-modality assumption (Assumption 3) may not hold in real pricing scenarios. The authors are advised to discuss the economic meaning, applicability, and necessity of this assumption in the main text. (b) The initial diversity assumption (Assumption 4) depends on the distribution of contexts, which may be difficult to guarantee in real-world deployments. The paper should discuss the potential effects when this assumption is violated.

Experiments: The experimental section is weak. (a) There is a lack of comparative baselines; the proposed method should be evaluated against representative approaches. (b) Purely theoretical results are insufficient to demonstrate robustness. It is recommended to include empirical comparisons under different smoothness levels β to strengthen the persuasiveness of the framework.

Validation: (a) While the research topic is practically meaningful, the paper mainly discusses theoretical pricing models, leaving a noticeable gap from real-world platform settings. (b) The absence of empirical or simulation-based validation makes it difficult for readers to assess the practical usability and robustness of the proposed algorithm on real data.

**Questions:**

See Weaknesses.

**Details Of Ethics Concerns:**

No.

---

> ### Author Response · Authors · 2025-11-22
> **Response to Reviewer tEP9**
>
> Thank you for raising points regarding empirical and practical aspects of our work! Please find our responses below:
>
> **1. Regarding Assumption 3 (Strong unimodality)**
>
> While this unimodality assumption is found to be natural in several specific problems, including modeling segment-level pricing (as discussed after Assumption 3 in [1]), we agree with the reviewer that this is a relatively strong assumption. We believe that future developments in the line of semi-parametric pricing will eventually match our regret upper bound without relying on this condition—much like what was ultimately achieved in the non-contextual setting by [2].
>
> In our analysis and algorithm design, the only part requiring Assumption 3 is the “distribution-shift subroutine’’ we call from [3] —the most important benchmark of our work that also makes this assumption. From a technical point of view, we believe we have already moved a bit forward from [3] in getting rid of Assumption 3. More precisely, in the forced-exploration analysis, we remove the need for strong unimodality in their argument via sharper analysis. We hope this provides a foundation on which future work can further relax or eliminate this assumption entirely.
>
>
> **2. Regarding Assumption 4 (Diverse context)**
>
> Thank you for raising this point. While we agree that it should be appealing to develop an algorithm fully get rid of diverse context assumption (as [4] do in $beta = 1$ setting), all existing works [3,5,6] in contextual semi-parametric pricing setting with $\beta > 1$ requires such kind of assumption up to our knowledge. Moreover, we would note that Assumption 4 we made here is already the most weak one compared with other works: it is only used to obtain an $\eta$-accurate initialization; the main online algorithm and regret analysis no longer rely on it once such an initialization is available.  In comparison:
> - [3] require the distribution have density lower bound during the initialization period.
> - [4,5] requires this assumption holds during the whole learning period.
>
>
> **3. Regarding Empirical Study**
>
> Thank you for raising this concern. We have added an experiment section in Appendix L to discuss the implementation and provide numerical results, including
>
> - testing the algorithm under different $\beta$ to illustrate its effect on regret.
> - testing the algorithm under different $d$ to see how the regret grows in $d$.
> - the comparison to explore-then-commit algorithm in [5].
>
> However, we want to emphasize that all of the above empirical studies are mainly for verifying or demonstrating our theoretical findings, instead of arguing that the proposed algorithm robustly achieves superior performance on real-data platforms. In fact, as a theoretical work aiming to achieve improved regrets under general $\beta$ settings, our algorithm design involves complicated sub-routines for theoretical purposes that are not scalable to large $d$ or $T$, as we discussed at the end of Appendix L.
>
> Despite this, we believe our work provides an important understanding of pricing problems by pushing the limit on the best possible regret one can achieve, and the procedures we introduced for theoretical purpose may also motivate future development of more efficient and practically implementations, though that is beyond the scope of our current work.
>
>
>
> **References**
>
> [1] Chen, Ningyuan, and Guillermo Gallego. "Nonparametric pricing analytics with customer covariates." Operations Research 69.3 (2021): 974-984.
>
> [2] Wang, Yining, Boxiao Chen, and David Simchi-Levi. "Multimodal dynamic pricing." Management Science 67.10 (2021): 6136-6152.
>
> [3] Wang, Yining, and Boxiao Chen. "Tight Regret Bounds in Contextual Pricing with Semi-parametric Demand Learning." Available at SSRN 5133677 (2025).
>
> [4] Tullii, Matilde, et al. "Improved algorithms for contextual dynamic pricing." Advances in Neural Information Processing Systems 37 (2024): 126088-126117.
>
> [5] Fan, Jianqing, Yongyi Guo, and Mengxin Yu. "Policy optimization using semiparametric models for dynamic pricing." Journal of the American Statistical Association 119.545 (2024): 552-564.
>
> [6] Luo, Yiyun, Will Wei Sun, and Yufeng Liu. "Contextual dynamic pricing with unknown noise: Explore-then-ucb strategy and improved regrets." Advances in Neural Information Processing Systems 35 (2022): 37445-37457.

---

### Official Review · Reviewer_a8PW · 2025-11-01

**Soundness:** 3
**Presentation:** 2
**Contribution:** 3
**Rating:** 6
**Confidence:** 3

**Summary:**

This paper considers the contextual pricing problem with binary purchase feedback, where a buyer’s valuation is modeled as a linear function of the context plus an arbitrary noise term whose CDF is $\beta$-Hölder continuous. Depending on the smoothness parameter $\beta$ of the CDF, this semi-parametric contextual pricing problem exhibits varying levels of difficulty, resulting in different regret exponents as functions of $\beta$.

With an improved confidence bound analysis, the authors present a universal algorithm for semi-parametric contextual pricing with general $\beta$, achieving a regret bound of $\widetilde{O}(T^{\frac{\beta+1}{2\beta+1}})$, which constitutes a uniform improvement over prior universal algorithms (previously $\widetilde{O}(T^{\frac{2\beta+1}{4\beta-1}})$ by Fan et. al.) and matches the optimal rates for specific cases such as $\beta = 1, 2$.

**Strengths:**

- Their upper bound is a uniform (i.e., over all $\beta$) improvement over the previous universal bound by Fan et al. (2024) (although there is a difference in assumptions between this work and Fan et al.). In particular, the new bound matches the optimal rates for $\beta = 1, 2$, where the previous bound had a gap.

- The technical contribution through an improved confidence bound analysis is solid. While the framework builds upon the analysis of Wang & Chen, the extension of the analysis to general $\beta$, especially $\beta \ge 2$, is non-trivial due to intricate dependency issues overlooked in Wang & Chen. The authors resolve this by directly using the analytical form of the local polynomial regression estimator and the Hanson–Wright inequality.

**Weaknesses:**

- While the bound matches the optimal rate in specific cases, there is no uniform (over all $\beta$) lower bound for this setting. Although a non-contextual lower bound of $\widetilde{\Omega}(T^{\frac{\beta+1}{2\beta+1}})$ exists and matches the claimed rate of this paper, that result does not rely on a strong unimodality assumption.

- The regret bound’s dependence on the feature dimension $d$ is quite large, $\mathrm{poly}(d^\beta, \log T)$ (Theorem 13), leading to exponential dependence on $\beta$. It is not clear how this dependency compares with previous works on semi-parametric contextual pricing. Moreover, the origin of this term is not well explained in either the main text or the proof of Theorem 13. The proof suggests that it stems from the burn-in time, but it is unclear where this burn-in time is formally established. It would be helpful for the authors to discuss the origin of this term and potential ways to mitigate this dependency.

- It would also be useful to provide numerical simulations supporting that this dependence is likely an artifact of the analysis rather than a fundamental limitation of the algorithm design.

**Questions:**

As mentioned in the weaknesses: where exactly does the $\mathrm{poly}(d^\beta, \log T)$ term come from? If it originates from the burn-in time, where is that proved? Or is it considered a standard knowledge in this line of work?

---

> ### Author Response · Authors · 2025-11-22
> **Response to Reviewer a8PW**
>
> Thank you for your valuable feedback and suggestions! Please find our point-to-point response below:
>
> **1. Origin of the $d^\beta$ dependency (Q1 \& W2).**
>
> We appreciate the the reviewer's careful reading and for raising this point. The polynomial dependence on $d^\beta$ is indeed a technical artifact of the matrix regularization that we introduce for the self-normalized analysis. We would explain the source of this this dependency in a two-step procedure:
>
>
> **Step1: Necessity on set $\zeta \asymp \eta^{-2} \asymp T^{\frac{\beta+1}{2\beta+1}}$:**
>
> Throughout the paper, $\zeta$ is tied to the pilot accuracy $\eta$ via $\zeta \asymp \eta^{-2}$. The reason is that the vector $\mathbf{\delta}\_j(\theta)$ is linear in $\theta - \theta_0$, so for all $\\| \theta - \theta\_0 \\| \leq \eta$ we have $\\| \mathbf{\delta}\_j(\theta) \\| \leq \eta$. in the proof of Proposition 8 (Eq. (8) in Appendix E), we explicitly use the condition $\zeta\cdot \\| \mathbf{\delta}\_j(\hat {\theta}) \\|^2 = \mathcal{O}(1)$ to control the regularized quadratic form $\mathbf{\delta}_j(\widehat {\theta})^\top (\Sigma_j(\widehat{\theta}) + \zeta I ) \mathbf{\delta}_j(\widehat {\theta})$. This naturally leads to the choice $\zeta \asymp \eta^{-2}$. Below we discuss where does the $\mathrm{Poly}(d^\beta, \log  T)$ term come from and possible mitigation.
>
> **Step2: A burn in requirement from $\zeta \gtrsim d^7\sqrt{T}\log T.$**
>
> Under the relation $\zeta \asymp \eta^{-2}$, Lemma 18(ii) shows that, in order to compare the empirical covariance $\Sigma_j(\theta)$ with its population analogue $\bar{\Sigma}_j(\theta)$ via  $\bar{\Sigma}_j(\theta) + \zeta I \preceq   2({\Sigma}_j(\theta) + \zeta I)$,
>         we additionally need $\zeta$ to dominate the covariance perturbation such that $\zeta \gtrsim d^7 \log^{7/2}(1/\delta) \sqrt{n_j}$, which is the final step in the proof of Lemma 18. Moreover, in the regret analysis we fix the pilot level as in Section 3, $\eta = T^{-\frac{\beta + 1}{4\beta + 2}}$. Combining, we need
>         \begin{align*}
>             T^{\frac{\beta +1}{2\beta +1}} \gtrsim d^7 \log(T) \sqrt{T} \quad \Longleftrightarrow  \quad T^{\frac{1}{4\beta + 2}} \gtrsim  d^7 \log(T).
>         \end{align*}
>         This defines a burn-in horizon $T_0(d,\beta)$: for $T \geq T_0(d,\beta) $ the assumptions of Proposition 8, Lemma 18, and Theorem 9 hold and we obtain
>         \begin{align*}
>             \mathrm{Regret}(T) \lesssim  d^4\log^{5/2} (T)  T^{\frac{\beta + 1}{2\beta + 1}}.
>         \end{align*}
>         For  $T < T_0(d,\beta) $, we simply use the trivial bound $  \mathrm{Regret}(T)  \leq T_0(d,\beta)$.  Since $T_0(d,\beta)$ is polynomial in $d^\beta$ (up to logarithmic factors), this contribution is summarized in Theorem 13 as the $\mathrm{Poly}(d^\beta, \log  T)$ term.
>
>
> **2. Possible way to mitigate the $d^\beta$ term (W2).**
>
> One simple way to eliminate the $\mathrm{Poly}(d^\beta, \log T)$ term is to strengthen the pilot stage and choose a larger regularization level so that $\eta^{-2} \asymp \zeta \gtrsim d^7 \log^{7/2}(T)\sqrt{T} $ holds for all relevant horizons. For example, we may increase the exploration length in Algorithm 3 so that $\eta^{-2}$ scales like $d^{7}T^{\frac{\beta +1}{2\beta +1}} $  instead of $T^{\frac{\beta +1}{2\beta +1}}$. Then the condition of Lemma 18(ii) is automatically satisfied and the burn-in disappears, so the regret bound no longer contains the $\mathrm{Poly}(d^\beta,\log  T)$ term. The price we pay is that the regret incurred by the pilot phase becomes dominant and leads to a substantially worse polynomial dependence on $d$ (roughly $d^{10}$) whereas our current calibration keeps the leading term close to $\tilde{\mathcal{O}}(d^4 T^{\frac{\beta +1}{2\beta +1}})$. We believe that a more careful analysis can remove the $\mathrm{Poly}(d^\beta, \log T)$ term without worsening the polynomial dependence on $d$ in the leading term, and we leave this improvement as an interesting direction for future work.
>
>
> **3. Numerical result regarding $d$-dependency (W3).**
>
> Thank you for this valuable suggestion! We have added Appendix L to include several numerical results of our algorithm. In particular, in Figure 5, we show that although the regret increases with $d$, the empirical growth rate is nearly linear and far milder than the dependence stated in our theorem. Moreover, the growth rate in $d$ does not appear to be strongly influenced by $\beta$. These observations suggest that both the $O(d^4)$ leading-order term and the $\mathrm{Poly}(d^\beta, \log T)$ factor in our stated bound are largely artifacts of the analysis rather than fundamental performance limitations. We have also added the related discussion below Theorem 13 in the revised version for clarity.

---

> > ### Comment · Reviewer_a8PW · 2025-11-26
> > **Thank you for your answers and revisions**
> >
> > Thank you for all the changes. All my questions and concerns have been addressed. I’m especially glad to see that the $d$-dependency is largely an artifact of the analysis and does not appear in practice, which in turn motivates refining the analysis further (though that may not yield a major contribution). The paper is now of much higher quality, and I believe it is a solid piece of work worthy of presentation. Accordingly, I have raised my score.

---

### Official Review · Reviewer_wC3A · 2025-11-01

**Soundness:** 2
**Presentation:** 3
**Contribution:** 2
**Rating:** 6
**Confidence:** 3

**Summary:**

This paper studies a contextual dynamic pricing problem, where the demand function is semiparametric (as single-index model) and Holder smooth. The key demand assumptions are: (1) strong uni-modality (2) positive definite covariate matrix (3) monotonicity of the link fucntion $g(\cdot)$. This paper claims the algorithm enjoys an optimal regret rate for general smoothness level $\beta$, which directly generalizes previous result of Wang and Chen (2025) with $\beta=2$.

I am generally positive about this paper in terms of the problem setup and claim contributions, but did not check the key technical proofs line by line, and therefore have limited assessment on the correctness, e.g., specific critique of prior proofs (Appendix K).

**Strengths:**

- The studied contextual pricing model is fundamental and the regret bound result is strong.
- The authors clearly articulates the difference compared to the prior work Wang and Chen (2025). Therefore, if correct, the result provides a solid contribution and fills the gap.
- The writing is good overall with clear explanation on the algorithm design and proof ideas.

**Weaknesses:**

- The numerical feasibility of the proposed algorithm is not very clear and no numerical study is shown.
   - The LPSP algorithm (Algorithm 2) relies on key smoothness parameter.
   - In particular, Algorithm 1 produces an estimate $\hat{g}(\cdot | \theta)$ for $\theta$ lying in a continuous range. How is this facilitated in computation? Note that Algorithm 1 is called routinely in Algorithm 2.
- The strong uni-modality assumption is still relatively strong, even if given the fact that several prior works also invoked it.

**Questions:**

- The algorithm relies on the knowledge of $\beta$. There is recently work on achieving smoothness-adaptive dynamic pricing in the non-contextual and linear-contextual case [1][2]. Can their approaches be potentially adopted here?
- Can the authors comment more on the regret dependence on the context dimension $d$? Currently the dependence is $d^4$. One would expect the performances might deteriorate quickly as $d$ increases.
- I understand that this work is mainly of theoretical nature. However, more detailed discussion on the numerical implementation or even including numerical simulations would strengthen the work.



[1] Ye Z, Jiang H. Smoothness-adaptive dynamic pricing with nonparametric demand learning. InInternational Conference on Artificial Intelligence and Statistics 2024 Apr 18 (pp. 1675-1683). PMLR.

[2] Gong X, Zhang J. Parameter-Adaptive Dynamic Pricing. arXiv preprint arXiv:2503.00929. 2025 Mar 2.

---

> ### Author Response · Authors · 2025-11-22
> **Response to Reviewer wC3A (1/2)**
>
> Thank you for your valuable feedback and suggestions! Please find our point-to-point response below:
>
> **1. Adaptivity to $\beta$ (Q1):**
>
> Thank you for pointing out this insightful question and the important related works.
> Yes, it might work under the self-similarity condition used in their works. However, one potential gaps on applying their frameworks directly to our setting is the **difference in problem formulation**:  [1] focuses only the non-contextual pricing setting without  $\mathsf{c}_t$ information, [2] has extended the study to contextual pricing setting but assumes a different additive demand model than ours. Thus extending their idea to our setting may require substantial work, and we believe this is worth a separate paper(indeed, both insightful works [1,2] are built upon previous non-adaptive results, e.g. [3,4], and we hope our work can serve as a non-adaptive backbone for later adaptive versions). We have added a discussion regarding this possible direction in the revised version.
>
> **2. Regarding the numerical implementation and results(W1, Q3):**
>
> Thank you for your valuable suggestion! We have added a new section in the appendix that discusses several subtle challenges in implementing the algorithm and presents the related experimental results. Below, we would briefly include the discussion related to W1 **regarding the storage of $\hat{g}(\cdot\lvert \theta)$ for a continuous range of $\theta$.**
>
> We agree with the reviewer that simultaneously maintaining $\{\hat{g}(\cdot \lvert \theta)\}\_{\theta}$ for $\theta$ lying in a continuous range can be subtle.  Fortunately, throughout the algorithm we only need to query $\hat{g}(\cdot \mid \theta)$ for continuously varying $\theta$ when solving the constrained LSE problem (2). After that step, we only need to maintain $\{\hat{g}(\cdot \lvert \hat{\theta}\_{\tau,j})\}$ for finitely many $j$.
> An important observation is that, given the data $\\{(x\_i, y\_i)\\}$ collected within each epoch, we can compute $\hat{g}(x_i \lvert \theta)$ for any $\theta$. This allows us to rewrite the objective function in (2) as a function of $\theta$, which can be evaluated directly from the collected data. As a result, solving (2) becomes feasible using standard black-box continuous constrained-optimization methods.
>
> We also note that, as a function of $\theta$, the objective in (2) is generally non-convex even when $\beta = 2$ (as discussed in [5] and in earlier statistical literature [6,7]). Following the approach in [5], we apply a general interior-point method for this continuous optimization problem, which is only ensured to return a local minimum but already has good empirical performance.
>
> We have added an experimental section in Appendix L of the revised version to discuss additional implementation details and to provide several numerical results that demonstrate our theoretical findings.
>
>
>
>
>
>
> **References**
>
> [1] Ye Z, Jiang H. Smoothness-adaptive dynamic pricing with nonparametric demand learning. InInternational Conference on Artificial Intelligence and Statistics 2024 Apr 18 (pp. 1675-1683). PMLR.
>
> [2] Gong X, Zhang J. Parameter-Adaptive Dynamic Pricing. arXiv preprint arXiv:2503.00929. 2025 Mar 2.
>
> [3] Bu, Jinzhi, David Simchi-Levi, and Chonghuan Wang. "Context-based dynamic pricing with separable demand models." Management Science (2025).
>
> [4] Wang, Yining, Boxiao Chen, and David Simchi-Levi. "Multimodal dynamic pricing." Management Science 67.10 (2021): 6136-6152.
>
> [5] Wang, Yining, and Boxiao Chen. "Tight Regret Bounds in Contextual Pricing with Semi-parametric Demand Learning." Available at SSRN 5133677 (2025)
>
> [6] Horowitz, Joel L., and Wolfgang Härdle. "Direct semiparametric estimation of single-index models with discrete covariates." Journal of the American Statistical Association 91.436 (1996): 1632-1640.
>
>
> [7] Härdle, Wolfgang, Peter Hall, and Hidehiko Ichimura. "Optimal smoothing in single-index models." The annals of Statistics (1993): 157-178.

---

> ### Author Response · Authors · 2025-11-22
> **Response to Reviewer wC3A (2/2)**
>
> **3. Comments on $d$ dependency(Q2).**
>
> The factor $d^{4}$ is inherited from the semi-parametric confidence bound in Theorem 9. There, for each bin $j$, we obtain a self-normalized bound of the form
> \begin{align*}
>      \mathbb{E}_{x\sim Q_j}[ \mathrm{Err}_j(x)] &\lesssim  d^{4}\log^{2}(1/\delta)  \big(n_j^{-\frac{1}{2}} +n^{-\frac{\beta}{2\beta +1}}  \big),
> \end{align*}
> where $\mathrm{Err}_j(x)$ controls the deviation $\big|\tilde g_j(x\mid\hat\theta_j)-g(x^\top\theta_0)\big|$ and $n_j$ is the local sample size.
>
> The exponent $4$ over $d$ comes from mainly from the union bound arguments in the analysis involving $\theta$, including:
>
> - an $\varepsilon$-net argument over the $d$-dimensional parameter ball $\{\theta:\|\theta-\theta_0\|\leq\eta\}$ in Lemma 18 (previously Lemma 17), which introduces a polynomial factor in $d$
> - a similar  $\varepsilon$-net argument with matrix concentration inequalities (matrix Bernstein) applied to the Gram matrices $\Lambda_j(\theta)$ and covariance matrices $\Sigma_j(\theta).$
>
> These steps are designed to keep the dependence explicit but are not optimized with respect to $d$; improving the dimension exponents in these concentration and covering arguments would directly translate into a smaller power of $d$ in front of $T^{\frac{\beta+1}{2\beta+1}}.$
>
>
> For comparison, the existing semi-parametric contextual pricing literature already exhibits nontrivial polynomial $d$-dependence: [1] states a regret bound scaling as $\tilde O(d^{3/2})$ for $\beta = 2$, and [2] states a $\tilde O(d^{5/2})$ dependence for general $\beta$. Our current $d^{4}$ factor is therefore somewhat more conservative but remains within a similar low-degree polynomial regime. We believe the current dependency is due to artificial terms involved in our analysis (please also see our numerical results in Appendix L, where empirically the algorithm performance grows only linearly in $d$). Improving this dimensional dependence is an interesting direction for future work.
>
>
>
> **References**
>
> [1] Wang, Yining, and Boxiao Chen. "Tight Regret Bounds in Contextual Pricing with Semi-parametric Demand Learning." Available at SSRN 5133677 (2025).
>
> [2] Fan, Jianqing, Yongyi Guo, and Mengxin Yu. "Policy optimization using semiparametric models for dynamic pricing." Journal of the American Statistical Association 119.545 (2024): 552-564.

---

> ### Comment · Reviewer_wC3A · 2025-11-23
> **Thanks. But Wrong Reference**
>
> I would like to thank the authors for the detailed response. I especially appreciate the clarification on the numerical feasibility of the implementation. A complete code package for reproduction would be highly welcome. I also recommend adding a brief discussion on the $d$ dependence in the revised version.
>
> However, I must also critically point out that the added references [1] (Ye & Jiang) [2] (Gong & Zhang)  contain incorrect (likely hallucinated) author first names. Please correct these citations.

---

> > ### Author Response · Authors · 2025-11-23
> >
> > Thank you for your feedback and additional suggestions! We apologize for the incorrect first names in two of the references and have updated the revised pdf with the corrected citations. Moreover
> >
> > - **Regarding the discussion in $d$:** we have added a new discussion below Theorem 13 in the revised version.
> > - **Regarding the code of implementation:** we have uploaded a new zip file including a Jupyter notebook that reproduces all the figures in the experiment section.
> >
> > We hope above updates can address the raised concerns. Thank you again for your helpful feedback!

---

### Meta-Review · Area_Chair_4hLq · 2026-01-09

**Summary:**

This paper investigates the dynamic pricing problem. The authors propose a semi-parametric model that combines a linear component with an unknown smooth function component to describe the variation of user demand with respect to price. For this setting they develop a pricing algorithm that achieves a unified optimal regret bound across different smoothness levels β.

The reviewers agree that this paper makes a strong theoretical contribution to the study of semi-parametric dynamic pricing. Concerns of reviewers include:
1. No numerical experiments.
2. The regret bound’s dependence on the feature dimension is quite large.

Overall, this is a solid theory paper.

**Reviewer Concerns:**

Most concerns are addressed.

**Reviewer Scores:**

6,6,4,6,6

At least 2 of the reviewers is likely to increase their scores.

---

### Decision · Program_Chairs · 2026-01-26

Accept (Poster)